# Fast Mixing Steady-State Control in Markov Decision Processes

**Federico Corso** [1]   **Marco Mussi** [1]   **Alberto Maria Metelli** [1]

## Abstract

Stability is a property of fundamental importance in real-world systems. Although it has been widely studied and well understood in *control theory* (CT) for deterministic systems, it is largely overlooked in stochastic systems such as *Markov decision processes* (MDPs). In this paper, we aim to translate the steady-state control problem, well established in CT, where the goal is to synthesize a controller with prescribed asymptotic stability properties, into the MDP framework. To this end, we propose the novel *fast-mixing steady-state* (FMSS) problem. Given an ergodic MDP and a target steady-state distribution, the objective is to synthesize a Markovian policy that induces this distribution with the fastest possible convergence rate. Addressing this problem requires controlling the spectral properties of the induced *Markov chain* (MC) transition matrix, which generally leads to non-convex programs. Thus, we derive a tractable surrogate objective that leads to a convex program, whose properties we study in terms of approximation quality, feasibility, and computational complexity. We then move to the learning setting and propose an offline sample-based algorithm for FMSS (`FMSS-SV`), designed for tabular MDPs, in which the environment's transition model is estimated from data. We quantify the impact of transition model estimation errors on both the objective value and the learned policy, and provide a finite-sample complexity analysis.

## 1. Introduction

When addressing sequential decision-making problems under uncertainty, *reinforcement learning* (RL, Sutton & Barto, 2018) has emerged as a central paradigm for learning policies (i.e., controllers)[1] that optimize a scalar objective, i.e., the expected cumulative *reward*, leveraging the framework of the *Markov decision processes* (MDPs, Puterman, 1994). Research efforts have mainly focused on providing guarantees for the reward maximization under different structural assumptions (e.g., Azar et al., 2017; Buşoniu et al., 2018; Jin et al., 2020b; Wang et al., 2020; Du et al., 2021). This emphasis is largely driven by the intuition underlying the *reward hypothesis* (Sutton, 2004), suggesting that any goal-directed behavior can be expressed as the maximization of an appropriate scalar signal.

Although this hypothesis has been recently questioned (Silver et al., 2021; Bowling et al., 2023), it appears quite natural that reward maximization alone is insufficient in real-world systems (Buşoniu et al., 2018). Reliance on a scalar reward signal might prove inadequate when requirements such as safety (García & Fernández, 2015), risk-sensitivity (Mihatsch & Neuneier, 2002), or robustness (Morimoto & Doya, 2005) are concerned. Another pressing requirement, often disregarded by the RL community (Wang et al., 2020), is the *stability* of the deployed policy. This includes a range of more specific desiderata, e.g., guaranteeing that the system state remains within safe bounds or ensuring convergence to a desired equilibrium. For this reason, in such applications, it is often preferred to rely on the rigorous but "limited" *control theory* (CT, Åström & Murray, 2021) frameworks, which enable the design of controllers with guarantees of *asymptotic stability*. This property is fundamental when the predictability of the system and the resilience w.r.t. exogenous phenomena are major concerns (Slotine & Li, 1991; Lewis et al., 2012).

More technically, the most basic problem in CT is the *steady-state control* (or regulation), where the objective is to ensure that the system state converges to a given steady state (or equilibrium) and is maintained there, even in the presence of external disturbances (Åström & Murray, 2021). This problem has been extensively studied for *deterministic linear* systems (Åström & Murray, 2021), where the existence of an equilibrium is determined by *spectral* properties of the closed-loop system. Specifically, the location of its eigenvalues in the complex plane is responsible for both the asymptotic stability (i.e., the *property* of converging to a

---

[1]Politecnico di Milano, Milan, Italy.
Correspondence to: Federico Corso <federico.corso@polimi.it>.

*Proceedings of the 43$^{rd}$ International Conference on Machine Learning*, Seoul, South Korea. PMLR 306, 2026. Copyright 2026 by the author(s).

---

[1]We use the terms "policy" and "controller" interchangeably.

steady state) and the rate of convergence (i.e., the *performance* in terms of convergence speed).

Unfortunately, the mathematical framework of MDPs differs fundamentally from the *deterministic setting* of CT. Consequently, analyzing the stability of RL-based controllers using CT tools is often ill-posed, given the stochastic nature of MDPs. As argued by Gros & Zanon (2022), stability in MDPs is best analyzed within the broader context of *Markov chains* (MCs, Meyn & Tweedie, 2012). In this *stochastic setting*, the search for an equilibrium is replaced by the analysis of *steady-state* (or equilibrium) *distributions*,[2] under the assumption of *ergodicity* (Levin & Peres, 2017). Crucially, ergodicity is the *property* that guarantees both the existence and uniqueness of the equilibrium distribution and the asymptotic convergence of the process to it. In this sense, ergodicity of MCs represents the direct analogue of asymptotic stability in deterministic linear systems.

For finite MDPs, the problem of synthesizing a policy capable of inducing a desired target equilibrium distribution has been formally addressed by Akshay et al. (2013). This work established that the problem admits a polynomial-time (PTIME) solution via *linear programming* (LP). However, significant theoretical questions have been left open. First, when the aforementioned LP is feasible, the solution is rarely unique, raising the critical question of how to select an appropriate policy from this candidate set. Second, when the target equilibrium distribution is not *realizable* within the given MDP, there is no established approach for computing a valid policy. These challenges have been only marginally touched in the RL literature, where the choice of a policy with such characteristics emerges as an auxiliary element rather than as a primary objective, and has been addressed mostly heuristically (e.g., Tarbouriech & Lazaric, 2019; Mutti & Restelli, 2020). Furthermore, existing analyses frequently rely on restrictive structural assumptions, e.g., *reversibility* (Levin & Peres, 2017), which limit their applicability to realistic systems.

In this work, we posit that the selection criterion for an adequate policy should be governed by the *rate of convergence* to the desired target equilibrium distribution. This objective mirrors the *performance* index of the classical steady-state control problem. Formally, the convergence rate of the induced MC is dictated by the spectral properties of the MC transition matrix, such as the *second-largest eigenvalue modulus* (SLEM, Levin & Peres, 2017; Meyn, 2022), and by its departure from *normality* (Ipsen & Selee, 2011). Consequently, the CT problem of designing a high-performance controller for the steady-state control problem translates, in MDPs, to synthesizing a policy that not only realizes the tar-

---

[2]In the literature, several alternative terms are used, such as "stationary" or "invariant" distribution/measure. In this paper, we employ the terms "steady-state" or "equilibrium" distribution.

get equilibrium distribution but minimizes the time required to reach it from any initial condition. This convergence rate is intrinsically connected to the so-called *mixing* properties (e.g., mixing time) of the induced MC (Levin & Peres, 2017). In this paper, we propose and address the novel *fast-mixing steady-state* (FMSS) control problem in MDPs that aims to determine a policy that attains a desired target equilibrium distribution with the fastest rate of convergence possible.

The need for such "fast-mixing" policies also emerges from the theoretical analysis of RL algorithms. Indeed, mixing times impact the sample efficiency of policy evaluation in the undiscounted setting (Kearns & Singh, 2002). Furthermore, finite-time convergence analyses of stochastic approximation methods, such as *temporal difference*, explicitly depend on the mixing properties of the MC (Bhandari et al., 2018; Srikant & Ying, 2019). Finally, a fast-convergence behavior is of interest for *Markov chain Monte Carlo* (MCMC, Andrieu & Thoms, 2008) methods, where accelerating the convergence of a sampler to a target posterior impacts computational efficiency. In this field, finding an MC minimizing the SLEM, under graph-based constraints, a problem connected to our FMSS, has been solved efficiently but for reversible chains only (Boyd et al., 2004; 2009).

**Original Contributions.** The contributions of the paper are summarized as follows.

- In Section 3, we introduce the FMSS control problem. Motivated by the fact that directly optimizing the SLEM leads to non-convex problems, we formulate it as a convex program over the policy space that minimizes the *second-largest singular value* (SLSV) of the induced MC transition matrix. Furthermore, we also propose a projection approach, PROJ, to handle the case where the desired target distribution is not realizable in the considered MDP.

- In Section 4, we analyze the FMSS program, showing that, for a fixed realizable distribution, there exists an infinite number of policies that induce it, motivating the need to search for a fast-mixing one. We also discuss the extent to which the SLSV can be considered an appropriate surrogate for the SLEM. Finally, we analyze the complexity of solving the FMSS optimization problem.

- In Section 5, we propose FMSS through Singular Values (FMSS-SV) that solves the FMSS problem by estimating the environment transition model from a trajectory of interaction. We characterize the *estimation error* and conduct an *error propagation analysis* that highlights the error due to the finite number of samples in the SLSV and in the induced steady-state distribution.

Finally, we include in Appendix A some remarks on the FMSS problem by solving analytically some simple MDPs, while Appendix F empirically validates FMSS-SV on a grid-world. All proofs are deferred to the Appendices B, C and D.

## 2. Preliminaries

**Notation.** Vectors are intended as column vectors, and denoted in bold. Matrices are identified through bold uppercase letters. The orthogonal complement of a vector $\boldsymbol{v}$ is denoted as $\mathrm{range}(\boldsymbol{v})^{\perp}$. The symbol $\mathbf{1}_d$ denotes a column vector of all ones, while $\mathbf{I}_d$ represents the identity matrix, both of size $d$. Given a vector $\mathbf{x} \in \mathbb{R}^d$ with non-negative components, we denote the induced inner product as $\langle \mathbf{v}, \mathbf{w} \rangle_{\mathbf{x}} = \mathbf{v}^{\top} \mathrm{diag}(\mathbf{x}) \mathbf{w}$ for every pair of vectors $\mathbf{v}, \mathbf{w} \in \mathbb{R}^d$. If the subscript is omitted, it is implicitly assumed that $\mathbf{x} = \mathbf{1}$, i.e., the usual Euclidean inner product. The choice of the inner product allows to define the $\ell^p(\mathbf{x})$ norm of a vector $\mathbf{v} \in \mathbb{R}^d$, as $\|\mathbf{v}\|_{\ell^p(\mathbf{x})} := \left[ \sum_{i \in [\![d]\!]} |v_i|^p x_i \right]^{1/p}$ for $p \in [1, +\infty)$ and as $\max_{i \in [\![d]\!]} |v_i|$ when $p = +\infty$. When $\mathbf{x} = \mathbf{1}$ then we denote the norm simply as $\|\mathbf{v}\|_p$. For a matrix $\mathbf{A}$, define the operator norm with respect to the above inner product as $\|\mathbf{A}\|_{\ell^2(\mathbf{x})} := \max_{\|\mathbf{v}\|_{\ell^2(\mathbf{x})}=1} \|\mathbf{A}\mathbf{v}\|_{\ell^2(\mathbf{x})}$, when $\mathbf{x} = \mathbf{1}$ simply denote it as $\|\mathbf{A}\|_2$. The spectral norm corresponds to the largest singular value $\sigma_1(\mathbf{A})$ w.r.t. the given inner product. The simplex over a set $\mathcal{X}$ is denoted as $\Delta(\mathcal{X})$. Random variables are identified through uppercase letters while lowercase ones denote their values. Given two distributions $p, q \in \Delta(\mathcal{X})$ on the finite set $\mathcal{X}$, their *total variation* (TV) is defined as $\|p - q\|_{\mathrm{TV}} = \frac{1}{2} \sum_{x \in \mathcal{X}} |p(x) - q(x)| = \frac{1}{2} \|p - q\|_1$. Let $a, b \in \mathbb{N}$ with $a < b$, we introduce the symbols $[\![a, b]\!] = \{a, \ldots, b\}$ and $[\![b]\!] = [\![1, b]\!]$.

**Markov Decision Processes.** We consider the model of an infinite-horizon, homogeneous, and discrete-time finite *Markov decision process* (MDP) without reward function (Puterman, 1994). Formally, consider the tuple $\mathcal{M} = \langle \mathcal{S}, \mathcal{A}, \mathbf{P}, \boldsymbol{\mu}_0 \rangle$, where $\mathcal{S}$ is the state space, with cardinality $|\mathcal{S}|$; $\mathcal{A}$ is the action space, with cardinality $|\mathcal{A}|$; $\mathbf{P} \in \mathbb{R}^{|\mathcal{S}||\mathcal{A}| \times |\mathcal{S}|}$ is the transition kernel where $p(s'|s, a)$ is the probability of reaching next state $s' \in \mathcal{S}$ given that action $a \in \mathcal{A}$ is played in state $s \in \mathcal{S}$, $\boldsymbol{\mu}_0 \in \mathbb{R}^{|\mathcal{S}|}$ is the initial-state distribution where $\mu_0(s)$ is the probability that the MDP is initialized at state $s$. We denote with $\boldsymbol{\pi} \in \Pi^{\mathrm{MR}}$ a stationary Markovian randomized policy, where $\pi(a|s)$ is the probability of playing action $a$ in state $s$. The policy is conveniently represented as a matrix $\boldsymbol{\Pi} \in \mathbb{R}^{|\mathcal{S}| \times |\mathcal{S}||\mathcal{A}|}$ with entries $\boldsymbol{\Pi}(s, (s, a)) = \pi(a|s)$ for every state $s$ and action $a$ and the other entries equal to 0.[3] Given the transition kernel $\mathbf{P}$, an initial state-distribution $\boldsymbol{\mu}_0$, and a policy $\boldsymbol{\pi}$, we denote with $\mathbb{P}_{\boldsymbol{\mu}_0}^{\mathbf{P}, \boldsymbol{\pi}}$ the induced probability measure over all possible sample paths $(S_t, A_t)_{t \geq 0}$.

**Markov Chains.** Any policy $\boldsymbol{\pi} \in \Pi^{\mathrm{MR}}$ induces a *Markov chain* (MC) $(S_t)_{t \geq 0}$ over the state space $\mathcal{S}$ defined as the tuple $\mathcal{M}^{\boldsymbol{\pi}} := \langle \mathcal{S}, \mathbf{P}^{\boldsymbol{\pi}}, \boldsymbol{\mu}_0 \rangle$ (Levin & Peres, 2017) where $\mathbf{P}^{\boldsymbol{\pi}} = \boldsymbol{\Pi} \mathbf{P} \in \mathbb{R}^{|\mathcal{S}| \times |\mathcal{S}|}$, with entries $p^{\boldsymbol{\pi}}(s'|s) =$

$\sum_{a \in \mathcal{A}} \pi(a|s) p(s'|s, a)$, represent the state transition matrix. Given $\mathcal{M}^{\boldsymbol{\pi}}$, we can compute recursively the state-distribution at time step $t \geq 1$ as $\boldsymbol{\mu}_t^{\boldsymbol{\pi}} = (\mathbf{P}^{\boldsymbol{\pi}})^{\top} \boldsymbol{\mu}_{t-1}^{\boldsymbol{\pi}}$ with $\boldsymbol{\mu}_0^{\boldsymbol{\pi}} = \boldsymbol{\mu}_0$. Moreover, we denote as $\mathbf{P}^{\boldsymbol{\pi}, t} = (\mathbf{P}^{\boldsymbol{\pi}})^t$ the $t$-step transition matrix, with entries $p^{\boldsymbol{\pi}, t}(s'|s)$ corresponding to the probability of landing in $s'$ at time step $t$ starting from $s$. A MC is *irreducible* if for every states $s, s' \in \mathcal{S}$ there exists $t \geq 1$ such that $p^{\boldsymbol{\pi}, t}(s'|s) > 0$. Furthermore, a MC is *aperiodic* if every state $s \in \mathcal{S}$ has period equal to 1, i.e., $\gcd(\{t \geq 1 : p^{\boldsymbol{\pi}, t}(s|s) > 0\}) = 1$. When studying the limiting behavior of the sequence of distributions $(\boldsymbol{\mu}_t^{\boldsymbol{\pi}})_{t \geq 0}$, it is natural to assume the underlying MDP to be *ergodic*.[4]

**Assumption 2.1** (Ergodic MDP). *An MDP $\mathcal{M}$ is* ergodic *if for every deterministic Markovian policy $\boldsymbol{\pi} \in \Pi^{\mathrm{MD}}$ the induced MC $\mathcal{M}^{\boldsymbol{\pi}}$ is irreducible and aperiodic.*

Assumption 2.1 ensures that the sequence of $t$-step distributions $(\boldsymbol{\mu}_t^{\boldsymbol{\pi}})_{t \geq 0}$ converges to a limiting distribution $\boldsymbol{\eta}^{\boldsymbol{\pi}} = \lim_{t \to +\infty} \boldsymbol{\mu}_t^{\boldsymbol{\pi}}$, called *steady-state* or *equilibrium* distribution $\boldsymbol{\eta}^{\boldsymbol{\pi}}$ that satisfies the *invariance equation*: $\boldsymbol{\eta}^{\boldsymbol{\pi}} = (\mathbf{P}^{\boldsymbol{\pi}})^{\top} \boldsymbol{\eta}^{\boldsymbol{\pi}}$. Moreover, we denote its minimum entry as $\eta_{\min}^{\boldsymbol{\pi}} = \min_{s \in \mathcal{S}} \eta^{\boldsymbol{\pi}}(s) > 0$, that is guaranteed to be positive thanks to irreducibility. We use $\mathbf{D}_{\boldsymbol{\eta}^{\boldsymbol{\pi}}} := \mathrm{diag}(\boldsymbol{\eta}^{\boldsymbol{\pi}})$ to denote the diagonal matrix formed by the entries of $\boldsymbol{\eta}^{\boldsymbol{\pi}}$.

The convergence of the MC to its equilibrium distribution is characterized by the following fundamental result.

**Theorem 2.1** (Convergence Theorem – Levin & Peres 2017, Theorem 4.9). *Let $\mathbf{P}^{\boldsymbol{\pi}}$ be irreducible and aperiodic, and let $\boldsymbol{\eta}^{\boldsymbol{\pi}}$ denote its stationary distribution. Then, there exist constants $\alpha \in (0, 1)$ and $C > 0$ such that:*

$$d(t; \mathbf{P}^{\boldsymbol{\pi}}) := \max_{s \in \mathcal{S}} \|p^{\boldsymbol{\pi}, t}(\cdot|s) - \boldsymbol{\eta}^{\boldsymbol{\pi}}\|_{\mathrm{TV}} \leq C\alpha^t. \quad (1)$$

We call $d(t; \mathbf{P}^{\boldsymbol{\pi}})$ the "distance from equilibrium" which, indeed, quantifies the speed at which the MC converges to its steady-state distribution.

**Remark 2.1** (Relation with asymptotic stability in CT). *If we introduce the error vector $\boldsymbol{e}_t^{\boldsymbol{\pi}} := \boldsymbol{\eta}^{\boldsymbol{\pi}} - \boldsymbol{\mu}_t$, and we view the equation $\boldsymbol{e}_t^{\boldsymbol{\pi}} = (\mathbf{P}^{\boldsymbol{\pi}})^{\top} \boldsymbol{e}_{t-1}^{\boldsymbol{\pi}}$ as the dynamics of a discrete-time linear system with state $\boldsymbol{e}_t^{\boldsymbol{\pi}}$ and consider the origin as the desired equilibrium point, Equation (1) is the asymptotic stability condition (in the CT sense) expressed in TV divergence rather than in the usual Euclidean distance. In other words, with a slight abuse of terminology, we are concerned with the asymptotic stability (in the CT sense) of a discrete-time linear system defined on the subspace of zero-sum vectors $\boldsymbol{e}_t^{\boldsymbol{\pi}} \in \mathbb{R}^{|\mathcal{S}|}$.*

---

[3]From now on, we will omit the adjective "Markovian" when referring to a policy, as we consider only this type of policies.

[4]We highlight that in the literature of MDPs, differently from what is assumed here, the ergodicity assumption only requires irreducibility and, thus, convergence to a limiting distribution in the Cesaro average sense. Aperiodicity is not explicitly assumed.(See Puterman 1994, Chapter 8.3; Kallenberg 2022, Chapter 5.2)

*Table 1.* Comparison between reversible, normal, and general MCs. In this table, given a fixed stationary distribution $\overline{\boldsymbol{\eta}}$ over $\mathcal{S}$, $\|\mathbf{N}^{\boldsymbol{\pi}}\|_2$ represents the spectral norm of the nilpotent component in the Schur decomposition of the matrix $\widetilde{\mathbf{A}}_{\overline{\boldsymbol{\eta}}}^{\boldsymbol{\pi}}$ defined in Equation (3). For a derivation of the bound (see Ipsen & Selee, 2011, Theorem 7.11). Notice that $\|\mathbf{N}^{\boldsymbol{\pi}}\|_2 = 0$ when $\mathbf{P}^{\boldsymbol{\pi}}$ is normal.

| MC class | Property | $\nu(\mathbf{P}^{\boldsymbol{\pi}})$ cvx? | $\sigma_2(\mathbf{P}^{\boldsymbol{\pi}})$ cvx? | $\nu(\mathbf{P}^{\boldsymbol{\pi}})$ bounds | $d(t; \mathbf{P}^{\boldsymbol{\pi}})$ bounds |
|---|---|---|---|---|---|
| **Reversible** | $\mathbf{P}^{\boldsymbol{\pi}} = \widetilde{\mathbf{P}}^{\boldsymbol{\pi}}$ 
 self-adjoint in $\ell^2(\boldsymbol{\eta}^{\boldsymbol{\pi}})$ | ✓ | ✓ | $\{\sigma_2(\mathbf{P}^{\boldsymbol{\pi}})\}$ | $\left[\frac{1}{2}\nu(\mathbf{P}^{\boldsymbol{\pi}})^t, \frac{1}{2\sqrt{\eta_{\min}^{\boldsymbol{\pi}}}}\nu(\mathbf{P}^{\boldsymbol{\pi}})^t\right]$ |
| **Normal** | $\mathbf{P}^{\boldsymbol{\pi}}\widetilde{\mathbf{P}}^{\boldsymbol{\pi}} = \widetilde{\mathbf{P}}^{\boldsymbol{\pi}}\mathbf{P}^{\boldsymbol{\pi}}$ 
 normal in $\ell^2(\boldsymbol{\eta}^{\boldsymbol{\pi}})$ | ✓ | ✓ | $\{\sigma_2(\mathbf{P}^{\boldsymbol{\pi}})\}$ | $\left[\frac{1}{2}\nu(\mathbf{P}^{\boldsymbol{\pi}})^t, \frac{1}{2\sqrt{\eta_{\min}^{\boldsymbol{\pi}}}}\nu(\mathbf{P}^{\boldsymbol{\pi}})^t\right]$ |
| **General** | – | ✗ | ✓ | $[\sigma_2(\mathbf{P}^{\boldsymbol{\pi}}) - \|\mathbf{N}^{\boldsymbol{\pi}}\|_2, \sigma_2(\mathbf{P}^{\boldsymbol{\pi}})]$ | $\left[\frac{1}{2}\nu(\mathbf{P}^{\boldsymbol{\pi}})^t, \frac{1}{2\sqrt{\eta_{\min}^{\boldsymbol{\pi}}}}\sigma_2(\mathbf{P}^{\boldsymbol{\pi}})^t\right]$ |

$\ell^2(\boldsymbol{\eta}^{\boldsymbol{\pi}})$ **Inner Product Space.** The constants $C$ and $\alpha$ are made explicit by studying the matrix $\mathbf{P}^{\boldsymbol{\pi}}$ as an operator acting on the weighted *inner product space* (IPS) $\ell^2(\boldsymbol{\eta}^{\boldsymbol{\pi}}) := \left(\mathbb{R}^{|\mathcal{S}|}, \langle \cdot, \cdot \rangle_{\boldsymbol{\eta}^{\boldsymbol{\pi}}}\right)$ which implicitly define the $\ell^2(\boldsymbol{\eta}^{\boldsymbol{\pi}})$-norm over the space of real-valued functions $f : \mathcal{S} \to \mathbb{R}$ as $\|f\|_{\ell^2(\boldsymbol{\eta}^{\boldsymbol{\pi}})} := \sqrt{\langle f, f \rangle_{\boldsymbol{\eta}^{\boldsymbol{\pi}}}}$. Accordingly, the $\ell^2(\boldsymbol{\eta}^{\boldsymbol{\pi}})$-distance from equilibrium is defined as $d_{\ell^2(\boldsymbol{\eta}^{\boldsymbol{\pi}})}(t; \mathbf{P}^{\boldsymbol{\pi}}) := \sup_{\|f\|_{\ell^2(\boldsymbol{\eta}^{\boldsymbol{\pi}})} \leqslant 1} \|\mathbf{P}^{\boldsymbol{\pi},t}f - \boldsymbol{\eta}^{\boldsymbol{\pi}}(f)\|_{\ell^2(\boldsymbol{\eta}^{\boldsymbol{\pi}})}$, where $\boldsymbol{\eta}^{\boldsymbol{\pi}}(f)$ is a short-hand notation for $\langle f, \mathbf{1} \rangle_{\boldsymbol{\eta}^{\boldsymbol{\pi}}} \mathbf{1}$. Importantly, $d(t; \mathbf{P}^{\boldsymbol{\pi}}) \leqslant \frac{1}{2}d_{\ell^2(\boldsymbol{\eta}^{\boldsymbol{\pi}})}(t; \mathbf{P}^{\boldsymbol{\pi}})$ (see Levin & Peres, 2017, Section 4.7). This space is fundamental to assess convergence to equilibrium of *reversible* MCs (Levin & Peres, 2017). Under the reversibility assumption $\mathbf{P}^{\boldsymbol{\pi}}$ is *self-adjoint* in $\ell^2(\boldsymbol{\eta}^{\boldsymbol{\pi}})$, that is $\langle \mathbf{P}^{\boldsymbol{\pi}}f, g \rangle_{\boldsymbol{\eta}^{\boldsymbol{\pi}}} = \langle f, \widetilde{\mathbf{P}}^{\boldsymbol{\pi}}g \rangle_{\boldsymbol{\eta}^{\boldsymbol{\pi}}} = \langle f, \mathbf{P}^{\boldsymbol{\pi}}g \rangle_{\boldsymbol{\eta}^{\boldsymbol{\pi}}}$. Here $\widetilde{\mathbf{P}}^{\boldsymbol{\pi}}$, i.e., the *adjoint* operator, is known as *time reversal* and is defined as $\widetilde{\mathbf{P}}^{\boldsymbol{\pi}} := \mathbf{D}_{\boldsymbol{\eta}^{\boldsymbol{\pi}}}^{-1}(\mathbf{P}^{\boldsymbol{\pi}})^{\top}\mathbf{D}_{\boldsymbol{\eta}^{\boldsymbol{\pi}}}$. Alternatively, this property may be expressed as $\mathbf{D}_{\boldsymbol{\eta}^{\boldsymbol{\pi}}}\mathbf{P}^{\boldsymbol{\pi}} = (\mathbf{P}^{\boldsymbol{\pi}})^{\top}\mathbf{D}_{\boldsymbol{\eta}^{\boldsymbol{\pi}}}$ (Seabrook & Wiskott, 2023), i.e., the *detailed balance* equations. Reversible chains satisfy the *spectral theorem* (Axler, 2024, Theorem 7.29), and admit a real-valued spectrum.

**Spectrum and SLEM.** We denote the spectrum of $\mathbf{P}^{\boldsymbol{\pi}}$, i.e., the set of distinct eigenvalues of $\mathbf{P}^{\boldsymbol{\pi}}$, as $\Lambda(\mathbf{P}^{\boldsymbol{\pi}}) = \{\lambda_1^{\boldsymbol{\pi}}, \lambda_2^{\boldsymbol{\pi}}, \ldots, \lambda_k^{\boldsymbol{\pi}}\}$, with $k \leqslant |\mathcal{S}|$.[5] Being $\mathbf{P}^{\boldsymbol{\pi}}$ a row-stochastic matrix its spectral radius satisfies $\rho(\mathbf{P}^{\boldsymbol{\pi}}) := \max_{i \in [\![1,k]\!]} |\lambda_i^{\boldsymbol{\pi}}| = 1$. Ergodicity ensures that 1 is a simple eigenvalue and also that $|\lambda_i^{\boldsymbol{\pi}}| < 1$ for every $i \in [\![2, k]\!]$. The right eigenspace associated with the eigenvalue 1 is spanned by the eigenvector $\mathbf{1}$ and is the space of constant-valued functions $f \in \ell^2(\boldsymbol{\eta}^{\boldsymbol{\pi}})$, while the left eigenvector is the equilibrium distribution $\boldsymbol{\eta}^{\boldsymbol{\pi}}$. We arrange the spectrum of $\mathbf{P}^{\boldsymbol{\pi}}$ in decreasing order of modulus as $1 = \lambda_1^{\boldsymbol{\pi}} > |\lambda_2^{\boldsymbol{\pi}}| \geqslant \cdots \geqslant |\lambda_k^{\boldsymbol{\pi}}|$ and denote with $\nu(\mathbf{P}^{\boldsymbol{\pi}}) := |\lambda_2^{\boldsymbol{\pi}}|$ the *second-largest eigenvalue modulus* (SLEM).

**Reversible MC.** If the MC $\mathbf{P}^{\boldsymbol{\pi}}$ is reversible, we have that $d(t; \mathbf{P}^{\boldsymbol{\pi}}) \leqslant \frac{1}{2\sqrt{\eta_{\min}^{\boldsymbol{\pi}}}}\nu(\mathbf{P}^{\boldsymbol{\pi}})^t$ (Levin & Peres, 2017, Lemma 12.18 (i)). Thus, convergence to equilibrium, for a given $\boldsymbol{\eta}^{\boldsymbol{\pi}}$ is completely controlled by the SLEM. This is confirmed

by the lower bound $d(t; \mathbf{P}^{\boldsymbol{\pi}}) \geqslant \frac{1}{2}\nu(\mathbf{P}^{\boldsymbol{\pi}})^t$ (Montenegro & Tetali, 2006, Theorem 4.9).[6] Note that, under reversibility, the SLEM $\nu(\mathbf{P}^{\boldsymbol{\pi}})$ is a *convex* function of the $\mathbf{P}^{\boldsymbol{\pi}}$ entries.

**Non-reversible MC.** When the MC is non-reversible, $\mathbf{P}^{\boldsymbol{\pi}}$ is no longer self-adjoint in $\ell^2(\boldsymbol{\eta}^{\boldsymbol{\pi}})$ and convergence analysis is notoriously more challenging (Chatterjee, 2025). Nevertheless, it is known that asymptotic convergence is characterized by the SLEM, as $\lim_{t \to +\infty} d(t; \mathbf{P}^{\boldsymbol{\pi}})^{1/t} = \nu(\mathbf{P}^{\boldsymbol{\pi}})$ (see Meyn, 2022, Theorem 6.2), independently of reversibility. Unfortunately, the SLEM $\nu(\mathbf{P}^{\boldsymbol{\pi}})$ is a *non-convex* function of the entries for general MCs. The most notable surrogate for $\nu(\mathbf{P}^{\boldsymbol{\pi}})$ appears to be the *second-largest singular value* (SLSV) defined as (Ipsen & Selee, 2011):

$$\sigma_2(\mathbf{P}^{\boldsymbol{\pi}}) := \|\mathbf{P}^{\boldsymbol{\pi}}\mathbf{Q}^{\perp \mathbf{1}}\|_{\ell^2(\boldsymbol{\eta}^{\boldsymbol{\pi}})}, \tag{2}$$

where $\mathbf{Q}^{\perp \mathbf{1}} := \mathbf{I}_{|\mathcal{S}|} - \mathbf{1}(\boldsymbol{\eta}^{\boldsymbol{\pi}})^{\top}$ is the orthogonal projection matrix. First, recall that in general, $\sigma_2(\mathbf{P}^{\boldsymbol{\pi}}) \geqslant \nu(\mathbf{P}^{\boldsymbol{\pi}})$. Moreover, this quantity satisfies $\sigma_2(\mathbf{P}^{\boldsymbol{\pi}}) = \nu(\mathbf{P}^{\boldsymbol{\pi}})$ whenever $\mathbf{P}^{\boldsymbol{\pi}}$ is either reversible or non-reversible yet *normal* (i.e., $\mathbf{P}^{\boldsymbol{\pi}}\widetilde{\mathbf{P}}^{\boldsymbol{\pi}} = \widetilde{\mathbf{P}}^{\boldsymbol{\pi}}\mathbf{P}^{\boldsymbol{\pi}}$). In fact, it is possible to show that the precision of the singular value in approximating $\nu(\mathbf{P}^{\boldsymbol{\pi}})$ degrades with the departure of $\mathbf{P}^{\boldsymbol{\pi}}$ from normality (Ipsen & Selee, 2011). Secondly, one of the most widely accepted, albeit only approximate, procedures to characterize the convergence of general MCs is through the SLEM of the *multiplicative reversibilization*, defined as $\mathbf{M}(\mathbf{P}^{\boldsymbol{\pi}}) := \mathbf{P}^{\boldsymbol{\pi}}\widetilde{\mathbf{P}}^{\boldsymbol{\pi}}$ that is reversible and converges to $\boldsymbol{\eta}^{\boldsymbol{\pi}}$. Then a well-known result states that $d(t; \mathbf{P}^{\boldsymbol{\pi}}) \leqslant \frac{1}{2\sqrt{\eta_{\min}^{\boldsymbol{\pi}}}}\nu(\mathbf{M}(\mathbf{P}^{\boldsymbol{\pi}}))^{t/2}$ (Fill, 1991, Theorem 2.1). Notably, $\sigma_2(\mathbf{P}^{\boldsymbol{\pi}}) = \sqrt{\nu(\mathbf{M}(\mathbf{P}^{\boldsymbol{\pi}}))}$. These properties are summarized in Table 1.

## 3. Problem Formulation

In this section, we introduce the *fast-mixing steady-state* (FMSS) control problem for MDPs, where the goal is to determine a policy $\boldsymbol{\pi} \in \Pi^{\mathrm{MR}}$ that makes the induced MC $\mathbf{P}^{\boldsymbol{\pi}}$ converge to a *target equilibrium distribution* $\overline{\boldsymbol{\eta}} \in \Delta(\mathcal{S})$ at the fastest possible rate allowed by the structure of the

---

[5] For general MCs, the spectrum is not necessarily real valued.

[6] This lower bound holds for general MCs, even not reversible.

MDP. In principle, as discussed in Section 2, one should minimize the distance to equilibrium $d(t; \mathbf{P}^{\boldsymbol{\pi}})$; however, this quantity is not easily manageable due to the need for choosing $t$ and the undesirable non-linear dependence on the policy entries $\boldsymbol{\pi}$ (see Equation 1). Similarly, minimizing the SLEM $\nu(\mathbf{P}^{\boldsymbol{\pi}})$ is challenging for non-normal MCs, as the objective is non-convex and may fail to control $d(t; \mathbf{P}^{\boldsymbol{\pi}})$ in the finite-time regime. Thus, we propose to formulate the FMSS control problem as the minimization of the second-largest singular value (SLSV) of $\mathbf{P}^{\boldsymbol{\pi}}$, denoted by $\sigma_2(\mathbf{P}^{\boldsymbol{\pi}})$.

## 3.1. Fast-Mixing Steady-State Control

It is convenient, for analytical purposes, to work in the standard Euclidean IPS $\ell^2$. For this reason, we introduce the isometry $T \colon \ell^2(\boldsymbol{\eta}^{\boldsymbol{\pi}}) \to \ell^2(\mathbb{R}^{|\mathcal{S}|})$, defined as $T\boldsymbol{f} = \mathbf{D}_{\boldsymbol{\eta}^{\boldsymbol{\pi}}}^{1/2}\boldsymbol{f}$ for every $\boldsymbol{f} \in \ell^2(\boldsymbol{\eta}^{\boldsymbol{\pi}})$. $T = \mathbf{D}_{\boldsymbol{\eta}^{\boldsymbol{\pi}}}^{1/2}$ is symmetric and, thanks to ergodicity, invertible. To show the isometric properties of this mapping notice that the scalar product fulfills the identify $\langle \boldsymbol{f}, \boldsymbol{g} \rangle_{\boldsymbol{\eta}^{\boldsymbol{\pi}}} = \langle T\boldsymbol{f}, T\boldsymbol{g} \rangle$ for every $\boldsymbol{f}, \boldsymbol{g} \in \ell^2(\boldsymbol{\eta}^{\boldsymbol{\pi}})$. This implies that $\|\boldsymbol{f}\|_{\ell^2(\boldsymbol{\eta}^{\boldsymbol{\pi}})} = \|\mathbf{D}_{\boldsymbol{\eta}^{\boldsymbol{\pi}}}^{1/2}\boldsymbol{f}\|_2$. Thus, the $\ell^2$-representation of operator $\mathbf{P}^{\boldsymbol{\pi}}$ is given by $\mathbf{A}_{\boldsymbol{\eta}^{\boldsymbol{\pi}}}^{\boldsymbol{\pi}} := \mathbf{D}_{\boldsymbol{\eta}^{\boldsymbol{\pi}}}^{1/2}\mathbf{P}^{\boldsymbol{\pi}}\mathbf{D}_{\boldsymbol{\eta}^{\boldsymbol{\pi}}}^{-1/2}$. Similarly, the representation of the orthogonal projector $\mathbf{Q}^{\perp \mathbf{1}} = \mathbf{I}_{|\mathcal{S}|} - \mathbf{1}\boldsymbol{\pi}^{\top}$ is denoted by the matrix $\boldsymbol{\Xi}^{\perp \sqrt{\boldsymbol{\eta}^{\boldsymbol{\pi}}}} = \mathbf{I}_{|\mathcal{S}|} - \sqrt{\boldsymbol{\eta}^{\boldsymbol{\pi}}}\sqrt{\boldsymbol{\eta}^{\boldsymbol{\pi}}}^{\top}$, implying that:

$$\|\mathbf{A}_{\boldsymbol{\eta}}^{\boldsymbol{\pi}}\boldsymbol{\Xi}^{\perp \sqrt{\boldsymbol{\eta}^{\boldsymbol{\pi}}}}\|_2 = \|\mathbf{P}^{\boldsymbol{\pi}}\mathbf{Q}^{\perp \mathbf{1}}\|_{\ell^2(\boldsymbol{\eta}^{\boldsymbol{\pi}})} = \sigma_2(\mathbf{P}^{\boldsymbol{\pi}}).$$

Now, for any given target stationary distribution $\overline{\boldsymbol{\eta}} \in \Delta(\mathcal{S})$ and any policy $\boldsymbol{\pi} \in \Pi^{\mathrm{MR}}$, we define the matrix:

$$\widetilde{\mathbf{A}}_{\overline{\boldsymbol{\eta}}}^{\boldsymbol{\pi}} := \mathbf{A}_{\overline{\boldsymbol{\eta}}}^{\boldsymbol{\pi}}\boldsymbol{\Xi}^{\perp \sqrt{\overline{\boldsymbol{\eta}}}} = \mathbf{D}_{\overline{\boldsymbol{\eta}}}^{1/2}\mathbf{P}^{\boldsymbol{\pi}}\mathbf{D}_{\overline{\boldsymbol{\eta}}}^{-1/2} - \sqrt{\overline{\boldsymbol{\eta}}}\sqrt{\overline{\boldsymbol{\eta}}}^{\top}. \quad (3)$$

We now formalize the FMSS($\overline{\boldsymbol{\eta}}$,$\mathbf{P}$) control problem as the following optimization problem:

$$\begin{aligned} &\underset{\boldsymbol{\pi} \in \Pi^{\mathrm{MR}}}{\text{minimize}} && \left\|\widetilde{\mathbf{A}}_{\overline{\boldsymbol{\eta}}}^{\boldsymbol{\pi}}\right\|_2 \\ &\text{subject to} && \overline{\boldsymbol{\eta}} = (\mathbf{P}^{\boldsymbol{\pi}})^{\top}\overline{\boldsymbol{\eta}} \end{aligned} \quad (4)$$

The linear equality constraint is enforcing that the target distribution $\overline{\boldsymbol{\eta}}$ is indeed an equilibrium distribution of the MDP. Intuitively, solving Program 4 amounts to identifying the policy $\boldsymbol{\pi} \in \Pi^{\mathrm{MR}}$ that induces the MC $\mathbf{P}^{\boldsymbol{\pi}}$ such that its multiplicative reversibilization, $\mathbf{M}(\mathbf{P}^{\boldsymbol{\pi}})$, achieves the minimal SLEM $\nu(\mathbf{M}(\mathbf{P}^{\boldsymbol{\pi}})) = \|\widetilde{\mathbf{A}}_{\overline{\boldsymbol{\eta}}}^{\boldsymbol{\pi}}\|_2^2$. Notice that the policy search is restricted to the space of Markovian randomized policies $\boldsymbol{\pi} \in \Pi^{\mathrm{MR}}$ since it has been shown that for the class of ergodic MDPs they are sufficient to induce any feasible target distribution over the state space (Akshay et al., 2013). Secondly, we emphasize how, differently from previous works (Boyd & Vandenberghe, 2004; Tarbouriech & Lazaric, 2019; Mutti & Restelli, 2020), by not imposing

a reversibility constraint of the form $\mathbf{D}_{\overline{\boldsymbol{\eta}}}\mathbf{P}^{\boldsymbol{\pi}} = (\mathbf{P}^{\boldsymbol{\pi}})^{\top}\mathbf{D}_{\overline{\boldsymbol{\eta}}}$, we implicitly allow for the selection of policies that induce non-reversible MCs, an important generalization.

## 3.2. Feasibility of the Target Distribution

In general, given a desired target $\overline{\boldsymbol{\eta}} \in \Delta(\mathcal{S})$, FMSS($\overline{\boldsymbol{\eta}}$,$\mathbf{P}$) might be infeasible. More formally, given an MDP $\mathcal{M}$, we define the set of realizable steady-state distributions as:

$$\mathcal{N}_{\mathbf{P}} := \left\{\boldsymbol{\eta} \in \Delta(\mathcal{S}) \mid \exists \mathbf{x} \in \mathcal{X}_{\mathbf{P}} : \boldsymbol{\eta} = \mathbf{K}\mathbf{x}\right\}. \quad (5)$$

where $\mathbf{K} = \mathbf{1}_{|\mathcal{A}|}^{\top} \otimes \mathbf{I}_{|\mathcal{S}|} \in \mathbb{R}^{|\mathcal{S}| \times |\mathcal{S}||\mathcal{A}|}$ is the action-marginalization matrix, defined for every $s, s' \in \mathcal{S}$ and $a \in \mathcal{A}$ as $K(s, (s', a)) = \mathbf{1}\{s' = s\}$, so that $(\mathbf{K}\mathbf{x})(s) = \sum_{a \in \mathcal{A}} x(s, a)$. In Equation (5), $\mathcal{X}_{\mathbf{P}}$ represents the set of state-action distributions $\mathbf{x}$ whose marginal over $\mathcal{S}$ is an equilibrium distribution in $\mathcal{M}$ for some policy $\boldsymbol{\pi}$:

$$\mathcal{X}_{\mathbf{P}} := \left\{\mathbf{x} \in \Delta(\mathcal{S} \times \mathcal{A}) \mid \mathbf{K}\mathbf{x} = \mathbf{P}^{\top}\mathbf{x}\right\}. \quad (6)$$

If $\overline{\boldsymbol{\eta}} \notin \mathcal{N}_{\mathbf{P}}$, then, Program 4 is not feasible since $\overline{\boldsymbol{\eta}}$ cannot be realized by any policy. We circumvent this issue by replacing it with a surrogate distribution $\widetilde{\boldsymbol{\eta}} \in \Delta(\mathcal{S})$ defined as the "closest" to the target $\overline{\boldsymbol{\eta}}$ according to an appropriate divergence $\mathcal{D}(\cdot\|\cdot)$. Formally, a candidate $\widetilde{\boldsymbol{\eta}}$ is obtained from the following projection problem PROJ($\overline{\boldsymbol{\eta}}$,$\mathbf{P}$):

$$\underset{\mathbf{x} \in \mathcal{X}_{\mathbf{P}}}{\text{minimize}} \quad \mathcal{D}(\overline{\boldsymbol{\eta}}\|\mathbf{K}\mathbf{x}). \quad (7)$$

Denoting with $\widetilde{\mathbf{x}}$ the solution to Program 7, we set in Program 4 as $\overline{\boldsymbol{\eta}} \leftarrow \widetilde{\boldsymbol{\eta}} = \mathbf{K}\widetilde{\mathbf{x}}$. In the above, $\mathcal{D}(\cdot\|\cdot)$ denotes any divergence convex in its second argument and well defined also for $\overline{\boldsymbol{\eta}} \geqslant 0$. Some examples are the standard $\ell^2$ norm squared or the KL-divergence (moment-projection).

# 4. Analysis of the Optimization Problem

In this section, we discuss the properties of Program 4 with specific reference to the feasible set of policies (Section 4.1), the objective function (Section 4.2), and computational complexity (Section 4.3).

## 4.1. Analysis of the Feasible Policy Space

For ease of exposition, we work with $\widetilde{\boldsymbol{\eta}} \in \mathcal{N}_{\mathbf{P}}$ so that feasibility is ensured. Let us denote the feasible set of policies as:

$$\begin{aligned} \Pi_{\widetilde{\boldsymbol{\eta}}} := \Big\{\boldsymbol{\pi} \in \Pi^{\mathrm{MR}} \mid \\ \widetilde{\eta}(s) = \sum_{s' \in \mathcal{S}, a' \in \mathcal{A}} \widetilde{\eta}(s')\pi(a'|s')p(s|s', a'), \forall s \in \mathcal{S}\Big\}. \end{aligned} \quad (8)$$

Equation (8) represents the set of Markovian randomized policies that satisfy the equilibrium constraints $\widetilde{\boldsymbol{\eta}} =$

$(\mathbf{\Pi P})^\top \widetilde{\boldsymbol{\eta}}$. Importantly, $\Pi_{\widehat{\boldsymbol{\eta}}}$ is a convex polytope, as it is derived from the intersection of the product of state-wise simplexes with an affine subspace. Given this definition, the following result holds.

**Proposition 4.1.** *Let $\mathcal{M}$ be an ergodic MDP with $|\mathcal{S}| \geqslant 1$ and $|\mathcal{A}| \geqslant 2$, let $\widetilde{\boldsymbol{\eta}} \in \mathcal{N}_{\mathbf{P}}$ be a feasible target distribution and let $\dim(\cdot)$ denote the dimension of a polytope. Define:*

$$\mathcal{X}_{\widetilde{\boldsymbol{\eta}}} := \left\{ \mathbf{x} \in \mathbb{R}^{|\mathcal{S}||\mathcal{A}|} \,\middle|\, \mathbf{K}\mathbf{x} = \widetilde{\boldsymbol{\eta}},\ \mathbf{P}^\top \mathbf{x} = \widetilde{\boldsymbol{\eta}},\ \mathbf{x} \geqslant 0 \right\}.$$

*Then, $\Pi_{\widetilde{\boldsymbol{\eta}}}$ and $\mathcal{X}_{\widetilde{\boldsymbol{\eta}}}$ are affinely isomorphic through $x(s, a) = \widetilde{\eta}(s)\pi(a|s)$, and, if $\Pi_{\widetilde{\boldsymbol{\eta}}}$ admits at least a $\boldsymbol{\pi}$ such that $\pi(a|s) > 0$ for all pairs $(s, a)$, then:*

$$\dim(\Pi_{\widetilde{\boldsymbol{\eta}}}) = \dim(\mathcal{X}_{\widetilde{\boldsymbol{\eta}}}) \geqslant 1.$$

*Otherwise, the feasible set $\Pi_{\widetilde{\boldsymbol{\eta}}}$ lies on the boundary of the policy simplex $\Pi^{\mathrm{MR}}$ and $\dim(\Pi_{\widetilde{\boldsymbol{\eta}}}) \geqslant 0$.*

Proposition 4.1 ensures that, whenever a feasible target distribution $\widetilde{\boldsymbol{\eta}} \in \mathcal{N}_{\mathbf{P}}$ is realized by a full-support policy, there exists an infinite number of policies realizing such a distribution. If no such full-support policy exists, the feasible set is confined to a lower-dimensional face of the policy simplex and can even reduce to a single policy. This further motivates our problem of choosing, within $\Pi_{\widetilde{\boldsymbol{\eta}}}$ whenever multiple feasible policies exist, the policy yielding the fastest convergence to the steady-state distribution.

### 4.2. Analysis of the Objective Function

Being $\|\widetilde{\mathbf{A}}_{\widehat{\boldsymbol{\eta}}}^{\boldsymbol{\pi}}\|_2$ the spectral norm of a linear function in $\mathbf{\Pi}$, it is convex over $\Pi_{\widehat{\boldsymbol{\eta}}}$ (see Boyd & Vandenberghe, 2004, Example 3.11). Thus, Program 4 is a convex optimization problem and it can be cast as a *semidefinite program* (Boyd & Vandenberghe, 2004). A key advantage of this formulation is that possible additional constraints on the policy, such as a minimum exploration constraint (i.e., $\pi(a|s) \geqslant \xi$ with $\xi \in [0, 1/|\mathcal{A}|]$), can be incorporated without sacrificing convexity. The theoretical guarantees of this formulation depend on the spectral properties of the transition kernel. When the induced transition matrix $\mathbf{P}^{\boldsymbol{\pi}}$ is normal for all $\boldsymbol{\pi} \in \Pi^{\mathrm{MR}}$, we have that $\|\widetilde{\mathbf{A}}_{\widehat{\boldsymbol{\eta}}}^{\boldsymbol{\pi}}\|_2 = \nu(\mathbf{P}^{\boldsymbol{\pi}})$ and, thus, the solution to Program 4 yields the minimum-SLEM policy among the ones inducing the target distribution $\widetilde{\boldsymbol{\eta}}$ in $\mathcal{M}$. Conversely, when $\mathbf{P}^{\boldsymbol{\pi}}$ is non-normal, we have that $\|\widetilde{\mathbf{A}}_{\widehat{\boldsymbol{\eta}}}^{\boldsymbol{\pi}}\|_2 = \sigma_2(\mathbf{P}^{\boldsymbol{\pi}}) \geqslant \nu(\mathbf{P}^{\boldsymbol{\pi}})$ and, thus, the resulting policy is the minimum-SLSV policy among the ones inducing the target distribution $\widetilde{\boldsymbol{\eta}}$ in $\mathcal{M}$. In other words, the solution is globally optimal w.r.t. the *upper bound* $\sigma_2(\mathbf{P}^{\boldsymbol{\pi}})$ on the SLEM $\nu(\mathbf{P}^{\boldsymbol{\pi}})$ as established in (Fill, 1991, Theorem 2.1). To tighten this bound and better approximate the true fastest policy, one might intuitively introduce a regularization term penalizing departures from normality. However, standard non-normality measures—such as the squared Frobenius

norm of the commutator, that is $\|(\widetilde{\mathbf{A}}_{\widehat{\boldsymbol{\eta}}}^{\boldsymbol{\pi}})^\top \widetilde{\mathbf{A}}_{\widehat{\boldsymbol{\eta}}}^{\boldsymbol{\pi}} - \widetilde{\mathbf{A}}_{\widehat{\boldsymbol{\eta}}}^{\boldsymbol{\pi}} (\widetilde{\mathbf{A}}_{\widehat{\boldsymbol{\eta}}}^{\boldsymbol{\pi}})^\top\|_F^2$, or Henrici's departure from normality (see Henrici, 1962), are non-convex with respect to $\boldsymbol{\pi}$. Incorporating such a penalty would therefore destroy the convexity of Program 4, negating its primary computational advantage. Consequently, as discussed in Section 2, accounting for the gap between $\sigma_2(\mathbf{P}^{\boldsymbol{\pi}})$ and the SLEM, $\nu(\mathbf{P}^{\boldsymbol{\pi}})$, for general non-reversible chains still remains non-trivial.

### 4.3. Computational Complexity

For tabular MDPs, solving Program 4 via *interior point methods* entails a space complexity of $O(|\mathcal{S}|^2 |\mathcal{A}|^2)$ and a time complexity of $O(|\mathcal{S}|^3 |\mathcal{A}|^3)$ (Den Hertog, 2012). While first-order methods can be employed to scale to larger problems, they typically exhibit a trade-off in solution accuracy.

### 4.4. Empirical Validation

A detailed empirical validation of Program 4, along with an analysis of its performance compared to the global optimum of the FMSS control problem for some specific MDP instances, is provided in Appendix A. The numerical results presented there support the theoretical properties discussed in Sections 2 and 4. Specifically, these experiments provide a visual way to see the difficulties caused by non-normality, which is the main obstacle to finding exact solutions to the FMSS control problem in general settings as briefly mentioned in Table 1. We refer the reader to Appendix A for a full description of the MDP structures and the resulting spectral behavior.

## 5. Learning Setting

In this section, we introduce and analyze `FMSS through Singular Values` (`FMSS-SV`, Algorithm 1), an algorithm to solve the FMSS control problem for tabular MDPs using a single trajectory collected *offline* by running a behavioral policy $\boldsymbol{\pi}^b \in \Pi^{\mathrm{MR}}$. Specifically, from Line 3 to Line 12, `FMSS-SV` leverages the collected trajectory $\tau = (S_t, A_t, S_{t+1})_{t \in [\![0, T-1]\!]}$ to obtain an estimate $\widehat{\mathbf{P}}$ of the true transition model $\mathbf{P}$. Importantly, the algorithm forces ergodicity of $\widehat{\mathbf{P}}$ by performing a mixture with a uniform next-state distribution, regulated by the parameter $\nu$ at Line 10. Then, in Line 14, given the target steady-state distribution $\overline{\boldsymbol{\eta}}$, `FMSS-SV` uses the estimated transition model $\widehat{\mathbf{P}}$ to solve PROJ$(\overline{\boldsymbol{\eta}}, \widehat{\mathbf{P}})$, i.e., projecting $\overline{\boldsymbol{\eta}}$ onto the set of realizable distributions $\mathcal{N}_{\widehat{\mathbf{P}}}$, obtaining $\widehat{\widetilde{\boldsymbol{\eta}}}$. Finally, in Line 15, the FMSS$(\widehat{\widetilde{\boldsymbol{\eta}}}, \widehat{\mathbf{P}})$ problem is solved making use of the estimated projected distribution $\widehat{\widetilde{\boldsymbol{\eta}}}$ and transition model $\widehat{\mathbf{P}}$ to obtain the candidate fast-mixing policy $\widehat{\boldsymbol{\pi}}^*$.

**Algorithm 1:** `FMSS through Singular Values.`

1. **Input:** Target steady-state distribution $\overline{\eta} \in \Delta(\mathcal{S})$, trajectory $\tau_b = (S_t, A_t, S_{t+1})_{t \in [\![0, T-1]\!]}$, parameter $\upsilon \in [0, \varepsilon/(2\sqrt{|\mathcal{S}||\mathcal{A}|})]$.
2. **Output:** Candidate fast-mixing policy $\widehat{\boldsymbol{\pi}}^* \in \Pi^{\mathrm{MR}}$ realizing $\widehat{\widehat{\overline{\eta}}} \simeq \overline{\eta}$.
3. **for** $(s, a) \in \mathcal{S} \times \mathcal{A}$ **do**
4.     **for** $s' \in \mathcal{S}$ **do**
5.         Compute the counts: $N^b(s, a, s') \leftarrow \sum_{t=0}^{T-1} \mathbf{1}\{(S_t, A_t, S_{t+1}) = (s, a, s')\}$
6.         $N^b(s, a) \leftarrow \sum_{s' \in \mathcal{S}} N^b(s, a, s')$
7.         Estimate the transition model:
8.         **if** $N^b(s, a) > 0$ **then** $\breve{p}(s'|s, a) \leftarrow \frac{N^b(s, a, s')}{N^b(s, a)}$ ;
9.         **else** $\breve{p}(s'|s, a) \leftarrow \frac{1}{|\mathcal{S}|}$ ;
10.         Regularize to ensure ergodicity: $\widehat{p}(s'|s, a) \leftarrow (1 - \upsilon)\breve{p}(s'|s, a) + \frac{\upsilon}{|\mathcal{S}|}$ ;
11.     **end**
12. **end**
13. Find a feasible candidate target steady-state distribution:
14. $\widehat{\widehat{\mathbf{x}}} \in \arg\min_{\mathbf{x} \in \mathcal{X}_{\widehat{\mathbf{P}}}} \mathcal{D}(\overline{\eta} \| \mathbf{K}\mathbf{x})$ and set $\widehat{\widehat{\overline{\eta}}} = \mathbf{K}\widehat{\widehat{\mathbf{x}}}$
15. **return** $\widehat{\boldsymbol{\pi}}^* \in \arg\min_{\boldsymbol{\pi} \in \Pi^{\mathrm{MR}}} \|\widehat{\widehat{\mathbf{A}}}_{\widehat{\widehat{\overline{\eta}}}}^{\boldsymbol{\pi}}\|_2$ s.t. $\widehat{\widehat{\overline{\eta}}} = (\widehat{\mathbf{P}}^{\boldsymbol{\pi}})^\top \widehat{\widehat{\overline{\eta}}}$

## 5.1. Sample Complexity Analysis

In this part of the paper, we conduct a sample complexity analysis for the estimation of the transition model. The proofs of the results presented in this section are provided in Appendix C.

We consider the availability of a trajectory $\tau_b = (S_t, A_t, S_{t+1})_{t \in [\![0, T-1]\!]}$ of length $T \in \mathbb{N}$ collected by running a behavioral policy $\boldsymbol{\pi}^b \in \Pi^{\mathrm{MR}}$ in the environment. To ensure an appropriate estimation of the transition model, we enforce that all actions are played with non-zero probability.

**Assumption 5.1** (Full-exploration—Meyn 2022, Section 9). *The behavioral policy $\boldsymbol{\pi}^b$ satisfies $\pi^b(a|s) \geqslant \xi$, with $\xi \in (0, 1/|\mathcal{A}|]$ for every $(s, a) \in \mathcal{S} \times \mathcal{A}$.*

Assumption 5.1 ensures that the joint process $(S_t, A_t)_{t \geqslant 0}$, where $A_t \sim \pi^b(\cdot|S_t)$, is an ergodic MC over $\mathcal{S} \times \mathcal{A}$. This condition guarantees that the reachable set of state-action pairs under $\boldsymbol{\pi}^b$ coincides with the whole $\mathcal{S} \times \mathcal{A}$, allowing sample averages to converge to their expectation and enabling the application of concentration inequalities.

The following result provides a polynomial upper bound to the sample complexity of Algorithm 1.

**Theorem 5.1** (Sample Complexity Guarantees). *Let $\mathcal{M}$ be a tabular and ergodic MDP (Assumption 2.1) and let $\boldsymbol{\pi}^b \in \Pi^{\mathrm{MR}}$ be a policy satisfying Assumption 5.1 inducing the steady-state distribution $\boldsymbol{\eta}^b$. Let $\tau^b = (S_t, A_t, S_{t+1})_{t \in [\![0, T-1]\!]}$ be a trajectory of length $T \in \mathbb{N}$*

*collected through $\boldsymbol{\pi}^b$. Let $\eta_{\min}^b = \min_{s \in \mathcal{S}} \eta^{\boldsymbol{\pi}^b}(s) > 0$ and let $t_{\mathrm{mix}}^b = \min\{t \geqslant 0 | d(t; \mathbf{P}^{\boldsymbol{\pi}^b}) \leqslant 1/8\}$ be the mixing time of the MC induced by $\boldsymbol{\pi}^b$. Let $\varepsilon > 0$ and $\delta \in (0, 1)$. Let $\widehat{\mathbf{P}}$ denote the estimated transition kernel as in Line 10, with $\upsilon \leqslant \varepsilon/2\sqrt{|\mathcal{S}||\mathcal{A}|}$. Then, with probability at least $1 - \delta$, to guarantee that:*

$$\left\|\widehat{\mathbf{P}} - \mathbf{P}\right\|_2 \leqslant 2\varepsilon, \tag{9}$$

*the sample complexity (i.e., length of the trajectory) is at most:*

$$T \leqslant \widetilde{O}\left(\frac{(1 + t_{\mathrm{mix}}^b)|\mathcal{S}||\mathcal{A}|\ln\left(\frac{1}{\delta}\right)}{\eta_{\min}^b \xi \varepsilon^2}\left(\ln\left(\frac{1}{\delta}\right) + |\mathcal{S}|\right)\right). \tag{10}$$

In the above, the notation $\widetilde{O}$ suppresses logarithmic factors dependent on $|\mathcal{S}|$, $|\mathcal{A}|$, and $\eta_{\min}^b$, as well as absolute constants. Notably, the sample complexity exhibits a tight dependence on the target accuracy $\varepsilon$ and scales polynomially w.r.t. the problem dimensionality $\widetilde{O}(|\mathcal{S}|^2|\mathcal{A}|)$. Dependences on the minimum entry $0 < \eta_{\min}^b \leqslant 1/|\mathcal{S}|$ of the steady-state distribution induced by $\boldsymbol{\pi}^b$ and on $0 < \xi \leqslant 1/|\mathcal{A}|$ the minimum action probability are present. They are unavoidable to cover the whole state-action pair for controlling the error in Euclidean norm. Notice that these two quantities are non-zero by Assumptions 2.1 and 5.1. Additionally, we emphasize a linear dependence on the *mixing time* $t_{\mathrm{mix}}^b$ of $\mathbf{P}^{\boldsymbol{\pi}}$, representing, in this specific case, the minimum number of steps needed to reduce the TV divergence from the steady-state distribution over $\mathcal{S}$ below the threshold $1/8$. This term can be related to the SLSV of $\mathbf{P}^{\boldsymbol{\pi}^b}$ as follows (Levin & Peres, 2017):

$$t_{\mathrm{mix}}^b \leqslant \frac{\ln\left(\frac{4}{\eta_{\min}^b}\right)}{\ln\left(\frac{1}{\sigma_2(\mathbf{P}^{\boldsymbol{\pi}^b})}\right)}. \tag{11}$$

The dependence on $t_{\mathrm{mix}}^b$ is necessary to guarantee that the state-action pairs are visited with a probability sufficiently close to the equilibrium one, i.e., $\boldsymbol{\eta}^b$. We further observe a dual-regime behavior characterized by the term $\ln\left(\frac{1}{\delta}\right) + |\mathcal{S}|$, which distinguishes between small and large $\delta$ regimes. This structure is typical of reward-free exploration and inverse RL frameworks (Jin et al., 2020a; Kaufmann et al., 2021; Ménard et al., 2021; Lazzati et al., 2024; Metelli et al., 2023). Furthermore, this term is multiplied by an additional factor of $\ln(1/\delta)$, a recurrent feature in offline RL (Xie et al., 2021) necessary to control the minimum number of samples across all reachable state-action pairs. Also, notice that, in the best case, in which $\eta_{\min}^b = 1/|\mathcal{S}|$ and $\xi = 1/|\mathcal{A}|$, we would retrieve a sample complexity scaling as $\widetilde{O}((1 + t_{\mathrm{mix}}^b)\ln(1/\delta)|\mathcal{S}|^2|\mathcal{A}|^2(|\mathcal{S}| + \ln(1/\delta))\varepsilon^{-2})$.

## 5.2. Error Propagation Analysis

In this section, we provide the error propagation analysis of `FMSS-SV`. We first study how the error on the transition model estimate propagates in the projection step PROJ and, then, we analyze its effect on the performance of the policy learned from FMSS. The proofs of the results presented here are reported in Appendix D.

**Projection Step.** We start by analyzing how the error introduced by using the estimated transition model $\widehat{\mathbf{P}}$ instead of the true one $\mathbf{P}$ impacts the projection, i.e., PROJ$(\overline{\eta},\widehat{\mathbf{P}})$ compared with PROJ$(\overline{\eta},\mathbf{P})$. We employ the following assumption.

**Assumption 5.2** (Lipschitz Divergence). *Let $\overline{\eta} \in \Delta(\mathcal{S})$. $\mathcal{D}(\overline{\eta}\|\cdot)$ is L-Lipschitz continuous w.r.t. the $\ell^1$-norm.*

This assumption is satisfied when $\mathcal{D}(\overline{\eta}\|\cdot)$ is the TV divergence or the $\ell^2$-norm (with $L = 1$) and for the KL-divergence whenever the underlying MDP is ergodic. Indeed, in such a case, we have that for every policy $\pi \in \Pi^{\mathrm{MR}}$, it holds that $\eta_{\min}^{\pi} = \min_{s \in \mathcal{S}} \eta^{\pi}(s) > 0$ and, thus, $L = (\min_{\pi \in \Pi^{\mathrm{MR}}} \eta_{\min}^{\pi})^{-1}$. The following result relates the projection error $\mathcal{D}(\overline{\eta}\|\widehat{\widetilde{\eta}})$ when using the estimated model $\widehat{\mathbf{P}}$ with that $\mathcal{D}(\overline{\eta}\|\widetilde{\eta})$ when using the true one $\mathbf{P}$.

**Theorem 5.2.** *Let $\overline{\eta} \in \Delta(\mathcal{S})$ be a target distribution and let $\mathbf{P}, \widehat{\mathbf{P}}$ be transition models of ergodic (Assumption 2.1) MDPs. Let $\widetilde{\eta}, \widehat{\widetilde{\eta}}$ be the solutions of the projection problems PROJ$(\overline{\eta},\mathbf{P})$ and PROJ$(\overline{\eta},\widehat{\mathbf{P}})$, respectively. Then, under Assumption 5.2, it holds that:*

$$\left|\mathcal{D}(\overline{\eta}\|\widetilde{\eta}) - \mathcal{D}(\overline{\eta}\|\widehat{\widetilde{\eta}})\right| \leqslant L\sqrt{|\mathcal{S}|} \sup_{\pi \in \Pi^{\mathrm{MR}}} \|\mathbf{Z}^{\pi}\|_2 \left\|\mathbf{P} - \widehat{\mathbf{P}}\right\|_2,$$

*where $\mathbf{Z}^{\pi} = (\mathbf{I} - \mathbf{P}^{\pi} + \mathbf{1}_{|\mathcal{S}|}(\eta^{\pi})^{\top})^{-1}$ is the* fundamental *matrix (Seneta, 1988) of the MC induced by policy $\pi$.*

The result shows that the difference in the projection error scales with the estimation error between the transition models $\|\mathbf{P} - \widehat{\mathbf{P}}\|_2$, as expected, multiplied by a term related to the ergodicity properties encoded in the fundamental matrix $\mathbf{Z}^{\pi}$, which relates to the SLSV of the MC, as shown in Lemma E.7:

$$\|\mathbf{Z}^{\pi}\|_2 \leqslant \|\mathbf{D}_{\eta^{\pi}}^{-1/2}(\mathbf{I} - \widetilde{\mathbf{A}}_{\eta^{\pi}}^{\pi})^{-1}\mathbf{D}_{\eta^{\pi}}^{1/2}\|_2 \leqslant \frac{\eta_{\min}^{-1/2}}{1 - \sigma_2(\mathbf{P}^{\pi})}.$$

**FMSS Problem.** We now analyze the effect of making use of the estimated transition model $\widehat{\mathbf{P}}$ when addressing the FMSS problem, i.e., comparing the solutions of the FMSS$(\widehat{\widetilde{\eta}},\widehat{\mathbf{P}})$ and FMSS$(\widetilde{\eta},\mathbf{P})$ problems. Notice that the effect of the estimation process impacts, as discussed above, the target distribution employed for the definition of the constraint of the FMSS problem too, i.e., $\widehat{\widetilde{\eta}}$ vs. $\widetilde{\eta}$. We evaluate two indexes of the approximation: $(i)$ the distance between

the desired target distribution $\widehat{\widetilde{\eta}} = \widehat{\eta}^{\widehat{\pi}^*}$[7] and the realized one $\eta^{\widehat{\pi}^*}$ obtained by playing the policy $\widehat{\pi}^*$, solution of the FMSS$(\widehat{\widetilde{\eta}},\widehat{\mathbf{P}})$, in the true $\mathbf{P}$ and $(ii)$ the gap between the SLSV $\sigma_2(\mathbf{P}^{\pi^*})$ of the MC $\mathbf{P}^{\pi^*}$ obtained when executing policy $\pi^*$, solution of FMSS$(\widetilde{\eta},\mathbf{P})$, and $\sigma_2(\mathbf{P}^{\widehat{\pi}^*})$ of the MC $\mathbf{P}^{\widehat{\pi}^*}$ obtained when executing policy $\widehat{\pi}^*$, solution of FMSS$(\widehat{\widetilde{\eta}},\widehat{\mathbf{P}})$, with the true transition model $\mathbf{P}$. The following result quantifies the first index of approximation.

**Lemma 5.3.** *Let $\overline{\eta} \in \Delta(\mathcal{S})$ be a target distribution and let $\mathbf{P}, \widehat{\mathbf{P}}$ be transition models of ergodic (Assumption 2.1) MDPs. Let $\widehat{\pi}^*$ a solution of FMSS$(\widehat{\widetilde{\eta}},\widehat{\mathbf{P}})$ problem. Then, it holds that:*

$$\left\|\widehat{\widetilde{\eta}} - \eta^{\widehat{\pi}^*}\right\|_2 \leqslant \|\mathbf{Z}^{\widehat{\pi}^*}\|_2 \left\|\widehat{\mathbf{P}} - \mathbf{P}\right\|_2. \tag{12}$$

This result is an application of Lemma E.5, itself obtained from the seminal result of perturbation theory for MCs (Seneta, 1988). The bound highlights that this gap depends, once again as expected, on the estimation error $\|\widehat{\mathbf{P}} - \mathbf{P}\|_2$ on the transition models. This term is now modulated by an index of ergodicity $\|\mathbf{Z}^{\widehat{\pi}^*}\|_2$ of the MC $\mathbf{P}^{\widehat{\pi}^*}$.

Let us now move to the analysis of the SLSV.

**Theorem 5.4.** *Let $\overline{\eta} \in \Delta(\mathcal{S})$ be a target distribution and let $\mathbf{P}, \widehat{\mathbf{P}}$ be transition models of ergodic (Assumption 2.1) MDPs. Let $\widetilde{\eta}, \widehat{\widetilde{\eta}}$ be the solutions of the projection problems PROJ$(\overline{\eta},\mathbf{P})$ and PROJ$(\overline{\eta},\widehat{\mathbf{P}})$, respectively. Let $\pi^*, \widehat{\pi}^*$ be solutions of FMSS$(\widetilde{\eta},\mathbf{P})$ and FMSS$(\widehat{\widetilde{\eta}},\widehat{\mathbf{P}})$ problems, respectively. Let $\widetilde{\eta}_{\min} = \min_{s \in \mathcal{S}} \widetilde{\eta}(s) > 0$ and $\widehat{\widetilde{\eta}}_{\min} = \min_{s \in \mathcal{S}} \widehat{\widetilde{\eta}}(s) > 0$. Then, it holds that:*

$$\left|\sigma_2(\mathbf{P}^{\pi^*}) - \sigma_2(\mathbf{P}^{\widehat{\pi}^*})\right| \leqslant 5\|\mathbf{Z}^{\widehat{\pi}^*}\|_2\sqrt{|\mathcal{S}|}\left((\eta_{\min}^{\widehat{\pi}^*})^{-1/2}\right.$$
$$\left. + \sqrt{|\mathcal{A}|}\widetilde{\eta}_{\min}^{-3/2}\widehat{\widetilde{\eta}}_{\min}^{-3/2}(1+\widetilde{H})\right)\left(2\sqrt{\left\|\widehat{\widetilde{\eta}} - \widetilde{\eta}\right\|_2} + \left\|\mathbf{P} - \widehat{\mathbf{P}}\right\|_2\right),$$

*where $\widetilde{H} > 0$ is a finite constant depending on $|\mathcal{S}| |\mathcal{A}|$ and on $\widehat{\mathbf{P}}$ and $\mathbf{P}$ only.*

To obtain this result, we introduce a general device for analyzing the perturbation of optimization problems when varying both the objective function and the feasible sets/constraints (Lemma D.1). Then, we leverage *Hoffman's inequalities* (Hoffman, 1952; Pena et al., 2021) to control the distance between the feasible sets of the two problems FMSS$(\widetilde{\eta},\mathbf{P})$ and FMSS$(\widehat{\widetilde{\eta}},\widehat{\mathbf{P}})$ making the *Hoffman constant* $\widetilde{H}$, which is guaranteed to be finite, appear in the bound.[8] The bound, as expected, depends on the distance between the quantities which differ due to estimation, i.e., the target

---

[7]Notice, instead, that $\widetilde{\eta} = \eta^{\pi^*}$.

[8]Several approaches to obtain bounds or approximations of $\widetilde{H}$ are discussed in (Pena et al., 2021).

distributions $\|\widehat{\widetilde{\boldsymbol{\eta}}} - \widetilde{\boldsymbol{\eta}}\|_2$ and the transition models $\|\mathbf{P} - \widehat{\mathbf{P}}\|_2$. Additionally, a dependence on the minimum probabilities $\widetilde{\eta}_{\min}$, $\widehat{\widetilde{\eta}}_{\min}$ and $\eta^{\widehat{\boldsymbol{\pi}}^*}_{\min}$ appears. These terms are guaranteed to be non-zero as both $\mathbf{P}$ and $\widehat{\mathbf{P}}$ are ergodic. Finally, from Theorem 5.4 together with the bounds in Table 1, we can characterize the absolute SLEM difference obtained when executing $\boldsymbol{\pi}^*, \widehat{\boldsymbol{\pi}}^*$ in the MDP with true transition model $\mathbf{P}$ as follows,

$$
\begin{aligned}
|\nu(\mathbf{P}^{\boldsymbol{\pi}^*}) - \nu(\mathbf{P}^{\widehat{\boldsymbol{\pi}}^*})| &\leqslant |\sigma_2(\mathbf{P}^{\boldsymbol{\pi}^*}) - \sigma_2(\mathbf{P}^{\widehat{\boldsymbol{\pi}}^*})| \\
&\quad + \max\{\|\mathbf{N}^{\boldsymbol{\pi}^*}\|_2, \|\mathbf{N}^{\widehat{\boldsymbol{\pi}}^*}\|_2\}.
\end{aligned}
$$

This result confirms that the SLEM performance difference is explicitly governed by the SLSV error and the system's non-normality.

**Experimental Evaluation:** Appendix F provides empirical validation on an $8 \times 8$ grid-world. We characterize the sensitivity of FMSS-SV, PROJ, and FMSS to the length $T$ of a trajectory collected under a uniform $\boldsymbol{\pi}^b$, confirming the predicted dependences on the estimation error $\|\mathbf{P} - \widehat{\mathbf{P}}\|_2$.

## 6. Conclusions

In this work, we introduced the *Fast-Mixing Steady-State* (FMSS) control problem, addressing it via a convex relaxation based on the *second-largest singular value* (SLSV). We provided finite-sample guarantees for the offline learning setting and analyzed the sensitivity of our algorithm to estimation errors. While exact for normal Markov chains, $\sigma_2(\mathbf{P}^{\boldsymbol{\pi}})$ remains a surrogate for the SLEM $\nu(\mathbf{P}^{\boldsymbol{\pi}})$ in general settings. Our analysis in Appendix A suggests that solving FMSS for general MDPs requires explicitly inducing non-normality; however, formulating an objective to control this property through policy selection remains a non-trivial open question. Future research directions include: $(i)$ investigating methods to steer policy selection toward highly non-normal dynamics; $(ii)$ characterizing the trade-off between asymptotic convergence properties characterized by the SLEM and finite-time mixing guarantees, as noted in Huang & Mao (2017), non-reversible chains also achieve faster finite-time convergence; $(iii)$ formalizing the hardness of the FMSS problem via conductance and Cheeger constants; $(iv)$ extending this framework to continuous state-action spaces, where we expect most of the challenges arising not to be conceptual but rather technical, primarily related to the mathematical tools required. In particular, one would need to address the mixing properties and spectral analysis of the densities of a Markov chain defined over a Hilbert space. While the core ideas remain intuitively similar, the mathematical treatment becomes significantly more involved.

## Acknowledgements

This publication was funded with the contribution of Ministero dell'Università e della ricerca pursuant to D.D. n. 7206 of 17 April 2025 – BANDO FIS 2. Project FIS-2023-02598 (Starting Grant), title: "Unified Learning from Diverse Human Feedback" (HUmLrn). CUP: D53C25000710001

## Impact Statement

This paper presents work whose goal is to advance the field of machine learning. There are many potential societal consequences of our work, none of which we feel must be specifically highlighted here.

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

# A. Some Simple MDP Environments

This appendix provides a targeted empirical evaluation of the theoretical framework developed in Sections 2 through 4. Our primary objective is to demonstrate that by relaxing the reversibility assumption and optimizing over singular values via Program 4, we can achieve significantly tighter convergence rates compared to standard reversible policies within the same MDP instance. Furthermore, we confirm a critical structural insight: the inability to explicitly account for the non-normality of the underlying MDP dynamics remains the predominant bottleneck in obtaining an *exact* solution to the FMSS control problem in the tabular setting.

As established in our preliminary analysis, characterizing the spectral properties of the induced kernel $\mathbf{P}^{\boldsymbol{\pi}}$ as a function of the policy $\boldsymbol{\pi} \in \Pi^{\mathrm{MR}}$ is notoriously difficult due to the operator's inherent non-normality and non-linearity. For this reason, we constrain our empirical analysis to some arbitrarily chosen MDP instances whose structures are the simplest yet non-trivial in the context of the FMSS problem. This allows for an extremely accurate characterization of the dependence on $\boldsymbol{\pi}$ of the spectral and normality properties of the induced $\mathbf{P}^{\boldsymbol{\pi}}$, while maintaining enough generality to provide meaningful insights into the broader FMSS problem. Finally, we remark that in order to overcome the substantial analytical challenge of providing closed-form expressions for the quantities of interest such as the SLEM $\nu(\mathbf{P}^{\boldsymbol{\pi}})$, the SLSV $\sigma_2(\mathbf{P}^{\boldsymbol{\pi}})$ and the stationary distribution $\boldsymbol{\eta}^{\boldsymbol{\pi}}$ as explicit functions of the policy parameters even for extremely small MDPs, we relied on SymPy (Meurer et al., 2017) to perform all analytical computations. Importantly, to preserve the highest level of accuracy in the results, all quantities have been first expressed in an *exact* symbolic representation and computed numerically only at the very end to visualize the results.

## A.1. Experimental Setting

We restrict our analysis to the three specific MDP instances reported in Figure 1. The choice is motivated by Proposition 4.1. Specifically, given a specific transition kernel $\mathbf{P}$ of an ergodic MDP and given a feasible target equilibrium distribution $\widetilde{\boldsymbol{\eta}} \in \mathcal{N}_{\mathbf{P}}$, the values $|\mathcal{S}| \in \{2, 3\}$ and $|\mathcal{A}| = 2$ are the smallest possible ensuring that the set $\Pi_{\widetilde{\boldsymbol{\eta}}}$, as in Equation (8), contains an infinite number of elements. In the following the policy entries are assumed to be parametrized as follows,

**Policy Parametrization for $\mathbf{P} \in \mathbb{R}^{6 \times 3}$**

$$\mathbf{\Pi} = \begin{bmatrix} \theta_1 & 1 - \theta_1 & 0 & 0 & 0 & 0 \\ 0 & 0 & \theta_2 & 1 - \theta_2 & 0 & 0 \\ 0 & 0 & 0 & 0 & \theta_3 & 1 - \theta_3 \end{bmatrix}$$

**Policy Parametrization for $\mathbf{P} \in \mathbb{R}^{4 \times 2}$**

$$\mathbf{\Pi} = \begin{bmatrix} \theta_1 & 1 - \theta_1 & 0 & 0 \\ 0 & 0 & \theta_2 & 1 - \theta_2 \end{bmatrix}$$

For convenience, we collect the parameters in a vector notation $\boldsymbol{\theta}$.

**About the Graph Structures and the Choice of P.** In Figure 1 are represented the directed graphs structures of the class of considered MDPs. These graphs are designed so that to ensure ergodicity (Assumption 2.1) by simply selecting for any two communicating states $s_i$ and $s_j$ entries $p(s_j|s_i, a) > 0$, $\forall a \in \mathcal{A}$. We further observe the following: the structure in Figure 1a is chosen so that to ensure $\forall \boldsymbol{\pi} \in \Pi^{\mathrm{MR}}$ that the induced $\mathbf{P}^{\boldsymbol{\pi}}$ is non-normal. This property is guaranteed by the missing path $s_1 \to s_3$. Also, as any $2 \times 2$ ergodic MC is reversible, the structure in Figure 1c is specifically chosen to address the case in which $\mathbf{P}^{\boldsymbol{\pi}}$ is reversible $\forall \boldsymbol{\pi} \in \Pi^{\mathrm{MR}}$. The specific values of $p(s'|s, a) > 0$ are then arbitrarily chosen. **About the Choice of the Feasible Target Distribution $\widetilde{\boldsymbol{\eta}}$.** For each instance displayed in Figure 1, a feasible target distribution is derived by solving,

$$\begin{cases} \boldsymbol{\eta} = (\mathbf{\Pi P})^{\top} \boldsymbol{\eta} \\ \mathbf{1}^{\top} \boldsymbol{\eta} = 1 \end{cases} \tag{13}$$

for $\mathbf{\Pi}$ set as the uniform distribution, i.e., $\theta_i = 1/|\mathcal{A}|$, $\forall i \in [\![|\mathcal{S}|]\!]$ and for $\mathbf{P}$ as specified in the directed graphs.

**Determining the Set $\Pi_{\widetilde{\boldsymbol{\eta}}}$.** To explicitly find the set of policies realizing $\widetilde{\boldsymbol{\eta}}$, we solve the stationarity constraints for $\mathbf{P}$ with respect to the parameters $\boldsymbol{\theta}$,

$$\begin{cases} \widetilde{\boldsymbol{\eta}} = (\mathbf{\Pi P})^{\top} \widetilde{\boldsymbol{\eta}} \\ \mathbf{\Pi 1} = \mathbf{1} \end{cases} \tag{14}$$

Importantly, this allow us to obtain the following sets of parameters $\boldsymbol{\theta}$,

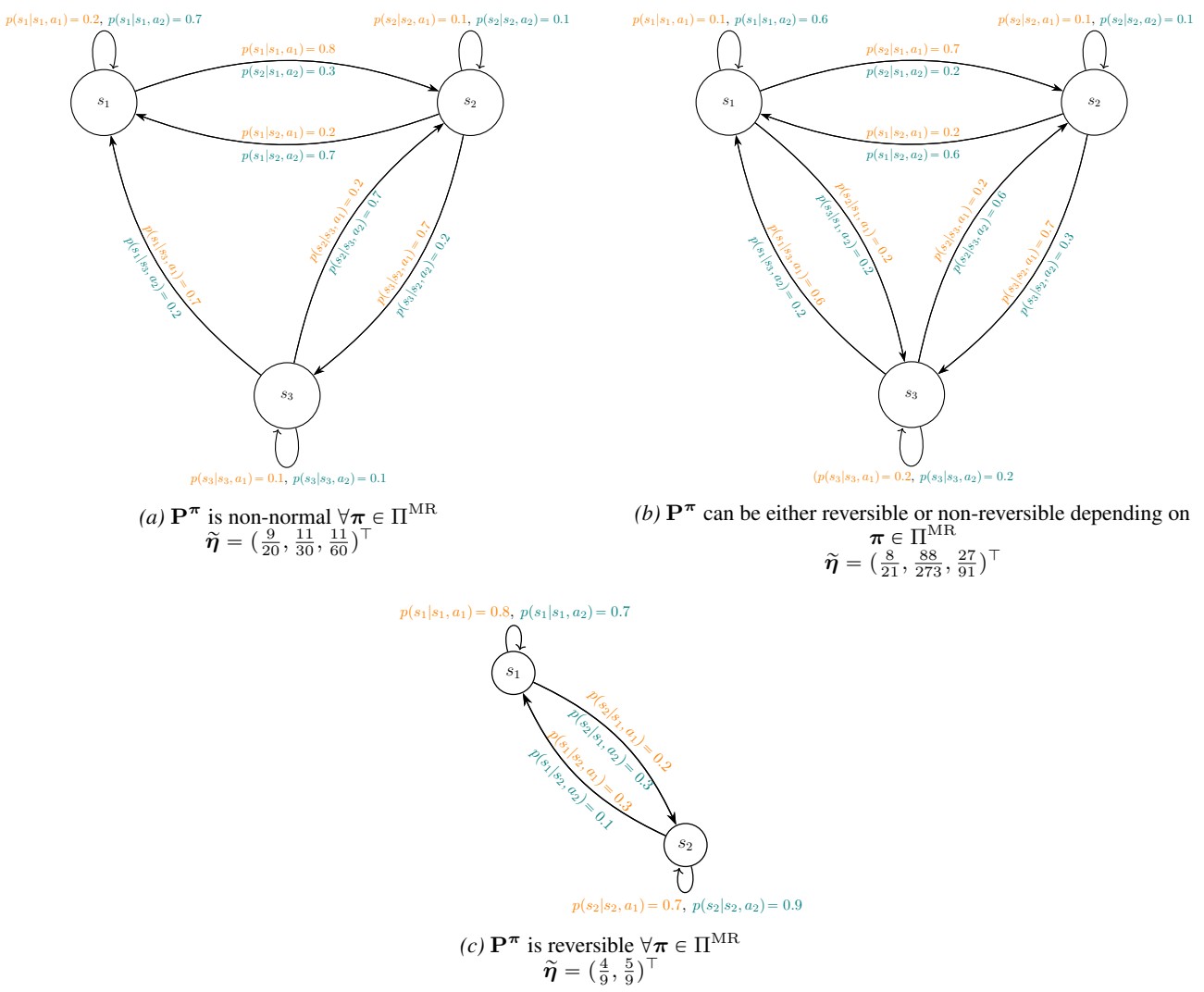

(a) $\mathbf{P}^{\boldsymbol{\pi}}$ is non-normal $\forall \boldsymbol{\pi} \in \Pi^{\mathrm{MR}}$
$\widetilde{\boldsymbol{\eta}} = \left( \frac{9}{20}, \frac{11}{30}, \frac{11}{60} \right)^{\top}$

(b) $\mathbf{P}^{\boldsymbol{\pi}}$ can be either reversible or non-reversible depending on $\boldsymbol{\pi} \in \Pi^{\mathrm{MR}}$
$\widetilde{\boldsymbol{\eta}} = \left( \frac{8}{21}, \frac{88}{273}, \frac{27}{91} \right)^{\top}$

(c) $\mathbf{P}^{\boldsymbol{\pi}}$ is reversible $\forall \boldsymbol{\pi} \in \Pi^{\mathrm{MR}}$
$\widetilde{\boldsymbol{\eta}} = \left( \frac{4}{9}, \frac{5}{9} \right)^{\top}$

*Figure 1.* Topologies of the considered MDP instances. Arranged from left to right, the structures exhibit a diminishing degree of non-normality. Ergodicity is guaranteed across all instances; irreducibility is ensured by the strong connectivity of the underlying transition graphs, while aperiodicity is maintained through the inclusion of self-loops.

**Policy Parameters for P in Figure 1a**  **Policy Parameters for P in Figure 1b**  **Policy Parameters for P in Figure 1c**

$$\begin{cases} \theta_1 = \frac{11}{27}\theta_3 + \frac{8}{27} \\ \theta_2 = \frac{1}{2} \\ \theta_3 = \theta_3, \text{ for } \theta_3 \in [0,1] \end{cases}$$

$$\begin{cases} \theta_1 = \frac{81}{130}\theta_3 + \frac{49}{260} \\ \theta_2 = \frac{1}{2} \\ \theta_3 = \theta_3, \text{ for } \theta_3 \in [0,1] \end{cases}$$

$$\begin{cases} \theta_1 = -\frac{5}{2}\theta_2 + \frac{7}{4} \\ \theta_2 = \theta_2, \text{ for } \theta_2 \in \left[\frac{3}{10}, \frac{7}{10}\right] \end{cases}$$

Importantly, notice how solving the system in Equation (14) leads to a solution $\boldsymbol{\theta}$ with one degree of freedom, as anticipated in Proposition 4.1. Denote such a free variable as $\theta_{\widetilde{\boldsymbol{\eta}}}$.

**Quantifying the Deviation from Normality.** A primary objective of our empirical analysis is to elucidate the mechanisms by which non-normality compromises the fidelity of the SLSV as a proxy for the SLEM. Specifically, we characterize the intricate interplay between the SLEM and the degree of non-normality exhibited by the induced kernel $\mathbf{P}^{\boldsymbol{\pi}}$ across the policy space $\boldsymbol{\pi} \in \Pi_{\widetilde{\boldsymbol{\eta}}}$. To this end, we employ the *relative departure from normality* (Henrici, 1962) of the matrix $\mathbf{P}^{\boldsymbol{\pi}}$, defined as follows:

$$h_{\mathrm{rel}}^2(\mathbf{P}^{\boldsymbol{\pi}}) = \frac{\|\widetilde{\mathbf{A}}_{\widetilde{\boldsymbol{\eta}}}^{\boldsymbol{\pi}}\|_F^2 - \|\widetilde{\boldsymbol{\Lambda}}^{\boldsymbol{\pi}}\|_F^2}{\|\widetilde{\mathbf{A}}_{\widetilde{\boldsymbol{\eta}}}^{\boldsymbol{\pi}}\|_F^2} \tag{15}$$

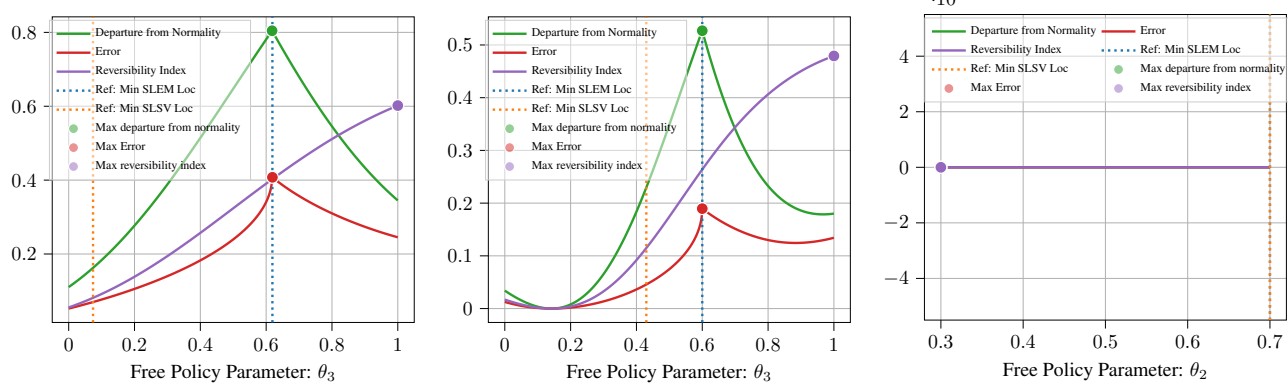

*(a)* Deviation from normality for the MDP in Figure 1a.

*(b)* Deviation from normality for the MDP in Figure 1b.

*(c)* Deviation from normality for the MDP in Figure 1c.

*Figure 2.* Quantitative analysis of the deviation from normality, non-reversibility indices, and SLEM approximation error. From left to right, we observe a simultaneous attenuation in both non-normality and non-reversibility, alongside a reduction in the SLEM approximation error. Notably, the minimum SLEM, indicating optimal convergence rate, coincides with the policy inducing the maximal deviation from normality, suggesting that non-normal dynamics are instrumental in accelerating convergence. Furthermore, our results demonstrate that maximizing the departure from reversibility is not, by itself, a sufficient condition for reducing the SLEM. Finally, Figure 2b demonstrates that restricting the search space to reversible policies severely limits the achievable convergence rate toward the target distribution $\widetilde{\boldsymbol{\eta}}$.

where $\|\cdot\|_F$ denotes the Frobenius norm and $\widetilde{\boldsymbol{\Lambda}}^{\boldsymbol{\pi}}$ is a diagonal matrix of the subdominant eigenvalues of $\mathbf{P}^{\boldsymbol{\pi}}$, counting algebraic multiplicity. This normalized metric $h_{\mathrm{rel}}^2(\mathbf{P}^{\boldsymbol{\pi}}) \in [0, 1]$ provides a bounded measure of the kernel's spectral behavior: $h_{\mathrm{rel}}^2(\mathbf{P}^{\boldsymbol{\pi}}) = 0$ if and only if $\mathbf{P}^{\boldsymbol{\pi}}$ is a normal operator, while $h_{\mathrm{rel}}^2(\mathbf{P}^{\boldsymbol{\pi}}) = 1$ denotes the maximal departure from normality achievable. The explicit quantification of this metric for the topologies depicted in Figure 1 is presented in Figure 2.

**Quantifying the Deviation from Reversibility.** For completeness, Figure 2 includes the *reversibility index* $\mathfrak{X}(\mathbf{P}^{\boldsymbol{\pi}})$, which quantifies the departure of $\mathbf{P}^{\boldsymbol{\pi}}$ from reversibility, or, equivalently, the degree of asymmetry of the matrix $\widetilde{\mathbf{A}}_{\widetilde{\boldsymbol{\eta}}}^{\boldsymbol{\pi}}$ over $\Pi_{\widetilde{\boldsymbol{\eta}}}$. We define this index as:

$$\mathfrak{X}(\mathbf{P}^{\boldsymbol{\pi}}) := \frac{\|\widetilde{\mathbf{Z}}_{\widetilde{\boldsymbol{\eta}}}^{\boldsymbol{\pi}}\|_F^2}{\|\widetilde{\mathbf{A}}_{\widetilde{\boldsymbol{\eta}}}^{\boldsymbol{\pi}}\|_F^2} \tag{16}$$

where $\widetilde{\mathbf{Z}}_{\widetilde{\boldsymbol{\eta}}}^{\boldsymbol{\pi}} = \frac{1}{2}(\widetilde{\mathbf{A}}_{\widetilde{\boldsymbol{\eta}}}^{\boldsymbol{\pi}} - (\widetilde{\mathbf{A}}_{\widetilde{\boldsymbol{\eta}}}^{\boldsymbol{\pi}})^\top)$ represents the *skew-symmetric* component of $\widetilde{\mathbf{A}}_{\widetilde{\boldsymbol{\eta}}}^{\boldsymbol{\pi}}$. Notably, $\mathfrak{X}(\mathbf{P}^{\boldsymbol{\pi}}) = 0$ corresponds to a symmetric $\widetilde{\mathbf{A}}_{\widetilde{\boldsymbol{\eta}}}^{\boldsymbol{\pi}}$ (implying reversibility), while $\mathfrak{X}(\mathbf{P}^{\boldsymbol{\pi}}) = 1/2$ means that it is balanced in its symmetric and skew-symmetric components. The extremal case $\mathfrak{X}(\mathbf{P}^{\boldsymbol{\pi}}) = 1$ occurs when $\widetilde{\mathbf{A}}_{\widetilde{\boldsymbol{\eta}}}^{\boldsymbol{\pi}}$ is purely skew-symmetric. As observed in Figure 2, maximizing the deviation from reversibility does not necessarily minimize the SLEM, underscoring that while non-reversibility can accelerate mixing, its relationship with the convergence rate is non-monotonic in general.

The analytical expressions for the relative departure from normality, $h_{\mathrm{rel}}^2(\mathbf{P}^{\boldsymbol{\pi}})$, and the reversibility index, $\mathfrak{X}(\mathbf{P}^{\boldsymbol{\pi}})$, as functions of the free policy parameter $\theta_{\widetilde{\boldsymbol{\eta}}}$, are omitted for brevity. Given the intricate dependencies of the matrix elements on the policy, these expressions are highly non-linear and offer limited additional intuition beyond the qualitative trends discussed in the following sections.

**Quantifying the Approximation Error.** To assess the fidelity of the SLSV as a proxy for the SLEM, Figure 2 further characterizes the approximation error. This absolute discrepancy is defined as:

$$\mathrm{err}(\boldsymbol{\pi}) := |\sigma_2(\mathbf{P}^{\boldsymbol{\pi}}) - \nu(\mathbf{P}^{\boldsymbol{\pi}})| \tag{17}$$

evaluated across the manifold of policies $\boldsymbol{\pi} \in \Pi_{\widetilde{\boldsymbol{\eta}}}$. This metric quantifies the conservatism of the SLSV as a spectral proxy; as the operator $\mathbf{P}^{\boldsymbol{\pi}}$ departs from normality, this error typically increases, illustrating the decoupling of the operator norm from its spectral radius. Notably, we observe a precise alignment between the maximal approximation error and the maximal departure from normality, both occurring at the same critical value of the free parameter $\theta_{\widetilde{\boldsymbol{\eta}}}$.

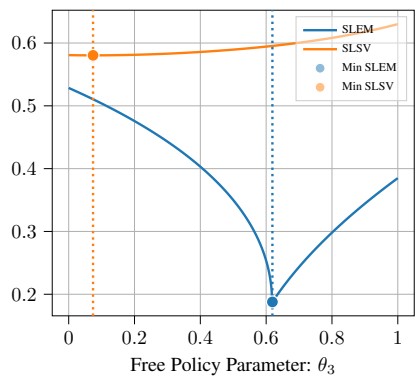

*(a)* SLEM vs SLSV for the MDP in Figure 1a.

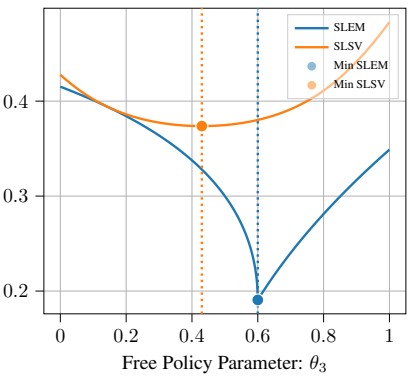

*(b)* SLEM vs SLSV for the MDP in Figure 1b.

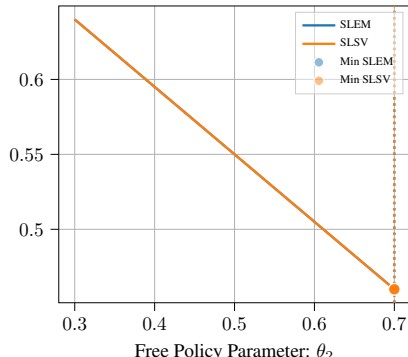

*(c)* SLEM vs SLSV for the MDP in Figure 1c.

*Figure 3.* Comparison of the Second Largest Eigenvalue Magnitude (SLEM) and the Second Largest Singular Value (SLSV) as functions of the free policy parameter. Moving from left to right, we observe that as the degree of non-normality in the underlying MDP transition structure decreases, the SLEM exhibits increased convexity and the SLSV serves as a progressively tighter approximation. The minimum value of the SLSV correspond to the solution of Program 4.

*Table 2.* Analysis of three simple finite MDP with different degree of non-normality. The function $h_{\mathrm{rel}}^2(\cdot)$ is the one defined in Equation (15). Also, the function $\mathfrak{X}(\mathbf{P}^\pi)$ is defined in Equation (16), and quantifies the departure from reversibility. Finally, $t_{\mathrm{mix}}^\pi$ denote the mixing time of $\mathbf{P}^\pi$ as in Equation (39) for $\zeta = 0.01$.

| MDP type | $\min_{\pi \in \Pi^{\mathrm{MR}}} \nu(\mathbf{P}^\pi)$ | $\min_{\pi \in \Pi^{\mathrm{MR}}} \sigma_2(\mathbf{P}^\pi)$ | $\max_{\pi \in \Pi^{\mathrm{MR}}} h_{\mathrm{rel}}^2(\mathbf{P}^\pi)$ | $\max_{\pi \in \Pi^{\mathrm{MR}}} \mathfrak{X}(\mathbf{P}^\pi)$ | $\max_{\pi \in \Pi^{\mathrm{MR}}} \lvert\sigma_2(\mathbf{P}^\pi) - \nu(\mathbf{P}^\pi)\rvert$ | $\min_{\pi \in \Pi^{\mathrm{MR}}} t_{\mathrm{mix}}^\pi$ | **FMSS-SV** $t_{\mathrm{mix}}^{\pi*}$ |
|---|---|---|---|---|---|---|---|
| **Highly Non-Normal, Fig 1a** | 0.1877 | 0.5804 | 0.8044 | 0.6018 | 0.4076 | 4 | 6 |
| **Moderately Non-Normal, Fig 1b** | 0.1906 | 0.3737 | 0.5270 | 0.4792 | 0.1895 | 4 | 4 |
| **Reversible, Fig 1c** | 0.46 | 0.46 | 0 | 0 | 0 | 6 | 6 |

### A.1.1. DISCUSSION OF THE RESULTS

Given the prescribed target distribution $\widetilde{\boldsymbol{\eta}}$, the transition kernel $\mathbf{P}$, and the closed-form characterization of the set of policies $\Pi_{\widetilde{\eta}}$ compatible with $\widetilde{\eta}$, the SLEM, $\nu(\mathbf{P}^\pi)$, can be explicitly parameterized by the free variable in $\theta_{\widetilde{\eta}}$. Furthermore, the objective function of Program 4 can be expressed analytically as well, permitting a closed-form derivation of the SLSV, $\sigma_2(\mathbf{P}^\pi)$, as a function of the same decision variable $\theta_{\widetilde{\eta}}$. The explicit analytical expressions for these quantities are omitted here because of their complexity and because they provide limited additional intuition.

As illustrated in Figure 3c, the SLEM and SLSV coincide perfectly for reversible instances of $\mathbf{P}^\pi$, and they are both convex functions of $\boldsymbol{\pi}$, consistently with the theoretical preliminaries established in Section 2. Consequently, the solution yielded by Program 4 is identical to the optimal mixing policy, thereby providing an *exact* solution to the FMSS control problem in the reversible regime. This equivalence is further stressed by the results in Figure 2b, where the relative departure from normality $h_{\mathrm{rel}}^2(\mathbf{P}^\pi)$, the reversibility index $\mathfrak{X}(\mathbf{P}^\pi)$, and the approximation error are all identically zero. Consequently, as shown in Figure 4c, the total variation distance from equilibrium as defined in Theorem 2.1 is equal. Also the mixing time, as defined in Equation (39), is equal. See Table 2 for precise numerical values extracted from the figures.

In contrast, Figure 3a demonstrates that when the underlying MDP topology is inherently non-normal for all $\boldsymbol{\pi} \in \Pi^{\mathrm{MR}}$ (as depicted in Figure 1a), the SLSV may serve as a significantly biased proxy for the SLEM. In such regimes, the SLEM $\nu(\mathbf{P}^\pi)$ exhibits a highly non-linear and potentially non-convex landscape. Consequently, the optimal policy parameters yielded by the singular-value-based Program 4 may deviate substantially from the configurations that minimize the true spectral radius $\nu(\mathbf{P}^\pi)$. A quantitative characterization of this behavior is provided in Figure 2a, which illustrates that the parameters corresponding to the fastest mixing policy coincide with the points of maximal departure from normality and peak approximation error. The resulting performance gap between the SLSV-optimal policy and the true spectral-optimal policy is visualized in Figure 4a, with detailed numerical comparisons summarized in Table 2. Finally, Figure 3b presents an intermediate regime corresponding to the topology depicted in Figure 1b. In this setting, the SLSV exhibits variable accuracy, transitioning between an exact and a biased approximation of $\nu(\mathbf{P}^\pi)$ as a function of the policy configuration. Most significantly, a joint analysis of Figure 3b and Figure 2b reveals that minimizing the SLSV can potentially provide convergence rates that are substantially superior to those achievable by any reversible policy within the same MDP.

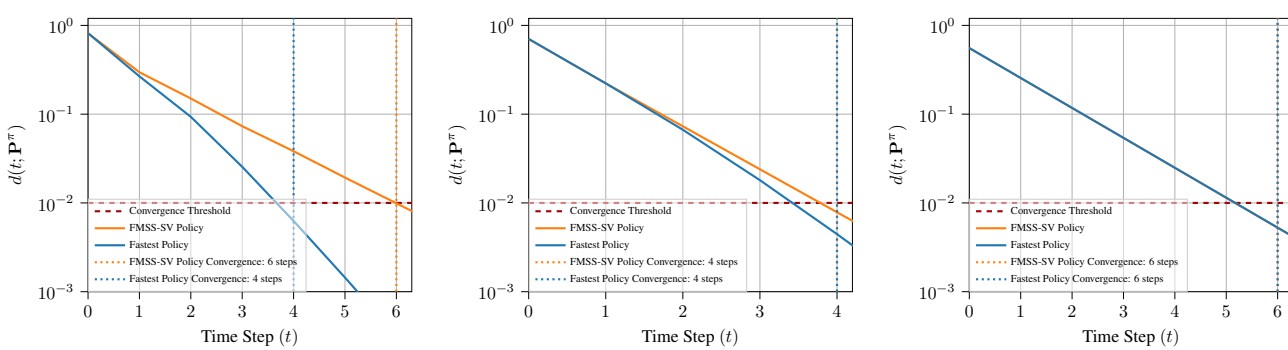

*(a)* Distance from Equilibrium, MDP in Figure 1a

*(b)* Distance from Equilibrium, MDP in Figure 1b

*(c)* Distance from Equilibrium, MDP in Figure 1c

*Figure 4.* Evolution of the distance from equilibrium as defined in Theorem 2.1. From left to right, the panels compare the convergence trajectories of policies optimized via SLSV minimization against those minimizing the SLEM. Notably, the performance profiles of the two objectives converge as the underlying MDP topology approaches reversibility, emphasizing the theoretical equivalence of spectral and singular-value-based measures in normal regimes.

Consequently, as illustrated in Figure 4b, the total variation distance trajectory for the SLSV-optimal policy is remarkably congruent with that of the true spectral-optimal policy. This suggests that despite the inherent bias in the singular-value proxy, it is possible for the resulting control policy to remain highly effective at capturing the accelerated mixing dynamics characteristic of non-reversible Markov chains. Again, see Table 2 for numerical values.

## B. Analysis of the Optimization Problem

**Proposition 4.1.** *Let $\mathcal{M}$ be an ergodic MDP with $|\mathcal{S}| \geqslant 1$ and $|\mathcal{A}| \geqslant 2$, let $\widetilde{\boldsymbol{\eta}} \in \mathcal{N}_{\mathbf{P}}$ be a feasible target distribution and let $\dim(\cdot)$ denote the dimension of a polytope. Define:*

$$\mathcal{X}_{\widetilde{\boldsymbol{\eta}}} := \left\{ \mathbf{x} \in \mathbb{R}^{|\mathcal{S}||\mathcal{A}|} \big| \mathbf{K}\mathbf{x} = \widetilde{\boldsymbol{\eta}}, \ \mathbf{P}^\top \mathbf{x} = \widetilde{\boldsymbol{\eta}}, \ \mathbf{x} \geqslant 0 \right\}.$$

*Then, $\Pi_{\widetilde{\boldsymbol{\eta}}}$ and $\mathcal{X}_{\widetilde{\boldsymbol{\eta}}}$ are affinely isomorphic through $x(s, a) = \widetilde{\eta}(s)\pi(a|s)$, and, if $\Pi_{\widetilde{\boldsymbol{\eta}}}$ admits at least a $\boldsymbol{\pi}$ such that $\pi(a|s) > 0$ for all pairs $(s, a)$, then:*

$$\dim(\Pi_{\widetilde{\boldsymbol{\eta}}}) = \dim(\mathcal{X}_{\widetilde{\boldsymbol{\eta}}}) \geqslant 1.$$

*Otherwise, the feasible set $\Pi_{\widetilde{\boldsymbol{\eta}}}$ lies on the boundary of the policy simplex $\Pi^{\mathrm{MR}}$ and $\dim(\Pi_{\widetilde{\boldsymbol{\eta}}}) \geqslant 0$.*

*Proof.* Fix a feasible target distribution $\widetilde{\boldsymbol{\eta}} \in \mathcal{N}_{\mathbf{P}}$ and define

$$\mathcal{X}_{\widetilde{\boldsymbol{\eta}}} := \left\{ \mathbf{x} \in \mathbb{R}^{|\mathcal{S}||\mathcal{A}|} \big| \mathbf{K}\mathbf{x} = \widetilde{\boldsymbol{\eta}}, \ \mathbf{P}^\top \mathbf{x} = \widetilde{\boldsymbol{\eta}}, \ \mathbf{x} \geqslant 0 \right\}. \tag{18}$$

By ergodicity, every stationary distribution induced by a stationary randomized policy has full support on $\mathcal{S}$; hence $\widetilde{\eta}(s) > 0$ for all $s \in \mathcal{S}$. Consider the maps

$$\Phi : \Pi_{\widetilde{\boldsymbol{\eta}}} \to \mathcal{X}_{\widetilde{\boldsymbol{\eta}}}, \qquad\qquad [\Phi(\boldsymbol{\pi})](s, a) = \widetilde{\eta}(s)\pi(a|s), \tag{19}$$

$$\Psi : \mathcal{X}_{\widetilde{\boldsymbol{\eta}}} \to \Pi_{\widetilde{\boldsymbol{\eta}}}, \qquad\qquad [\Psi(\mathbf{x})](a|s) = \frac{x(s, a)}{\widetilde{\eta}(s)}. \tag{20}$$

The positivity of $\widetilde{\boldsymbol{\eta}}$ makes $\Psi$ well defined. Moreover, $\Psi(\Phi(\boldsymbol{\pi})) = \boldsymbol{\pi}$ and $\Phi(\Psi(\mathbf{x})) = \mathbf{x}$, so $\Phi$ and $\Psi$ are inverse affine maps. Therefore

$$\dim(\Pi_{\widetilde{\boldsymbol{\eta}}}) = \dim(\mathcal{X}_{\widetilde{\boldsymbol{\eta}}}). \tag{21}$$

Let $\mathcal{L}_{\widetilde{\boldsymbol{\eta}}}$ be the affine hull of $\mathcal{X}_{\widetilde{\boldsymbol{\eta}}}$ obtained by dropping the non-negativity constraints:

$$\mathcal{L}_{\widetilde{\boldsymbol{\eta}}} := \left\{ \mathbf{x} \in \mathbb{R}^{|\mathcal{S}||\mathcal{A}|} \big| \mathbf{C}\mathbf{x} = \mathbf{b} \right\}, \quad \mathbf{C} := (\mathbf{P}^\top, \mathbf{K})^\top, \quad \mathbf{b} := (\widetilde{\boldsymbol{\eta}}, \widetilde{\boldsymbol{\eta}})^\top. \tag{22}$$

Since $\mathbf{P}$ is stochastic and $\mathbf{K}$ marginalizes over actions,

$$\mathbf{1}^\top \mathbf{P}^\top = \mathbf{1}^\top \mathbf{K}, \qquad (\mathbf{1}^\top, -\mathbf{1}^\top)\mathbf{b} = 0. \tag{23}$$

Hence one row constraint in $\mathbf{C}\mathbf{x} = \mathbf{b}$ is redundant, and

$$\mathrm{rank}(\mathbf{C}) \leqslant 2|\mathcal{S}| - 1. \tag{24}$$

Since $\mathcal{X}_{\widetilde{\boldsymbol{\eta}}} \neq \varnothing$, also $\mathcal{L}_{\widetilde{\boldsymbol{\eta}}} \neq \varnothing$, and thus

$$\begin{aligned}
\dim(\mathcal{L}_{\widetilde{\boldsymbol{\eta}}}) = \dim(\ker \mathbf{C}) &= |\mathcal{S}||\mathcal{A}| - \mathrm{rank}(\mathbf{C}) \\
&\geqslant |\mathcal{S}||\mathcal{A}| - (2|\mathcal{S}| - 1) = |\mathcal{S}|(|\mathcal{A}| - 2) + 1.
\end{aligned} \tag{25}$$

Suppose first that $\widetilde{\boldsymbol{\eta}}$ is realized by a full-support policy $\boldsymbol{\pi}^0$. Let $\mathbf{x}^0 = \Phi(\boldsymbol{\pi}^0)$, i.e.,

$$x^0(s, a) = \widetilde{\eta}(s)\boldsymbol{\pi}^0(a|s) > 0, \qquad \forall (s, a) \in \mathcal{S} \times \mathcal{A}. \tag{26}$$

For any $\boldsymbol{v} \in \ker \mathbf{C}$, $\mathbf{x}^0 + t\boldsymbol{v} \in \mathcal{L}_{\widetilde{\boldsymbol{\eta}}}$ for all $t \in \mathbb{R}$. Moreover, if $\boldsymbol{v} \neq 0$, choosing

$$0 < \varepsilon < \min_{(s,a):\, v(s,a) \neq 0} \frac{x^0(s, a)}{|v(s, a)|}, \tag{27}$$

gives, for every $(s,a)$ and every $|t| \leqslant \varepsilon$,

$$x^0(s,a) + tv(s,a) \geqslant x^0(s,a) - \varepsilon|v(s,a)| > 0. \tag{28}$$

Hence $\mathbf{x}^0 + t\boldsymbol{v} \in \mathcal{X}_{\widehat{\boldsymbol{\eta}}}$ for all $|t| \leqslant \varepsilon$. Therefore every direction in $\ker \mathbf{C}$ is a feasible local direction of $\mathcal{X}_{\widehat{\boldsymbol{\eta}}}$ at $\mathbf{x}^0$. Consequently,

$$\dim(\mathcal{X}_{\widehat{\boldsymbol{\eta}}}) = \dim(\mathcal{L}_{\widehat{\boldsymbol{\eta}}}) \geqslant |\mathcal{S}|\,(|\mathcal{A}| - 2) + 1 \geqslant 1, \tag{29}$$

where the last inequality follows from $|\mathcal{A}| \geqslant 2$ and $|\mathcal{S}| \geqslant 1$. The same bound for $\dim(\Pi_{\widehat{\boldsymbol{\eta}}})$ follows from (21).

Suppose now that no full-support policy realizes $\widetilde{\boldsymbol{\eta}}$. By the bijection above, this is equivalent to

$$\forall \mathbf{x} \in \mathcal{X}_{\widetilde{\boldsymbol{\eta}}}, \quad \exists (s,a) \in \mathcal{S} \times \mathcal{A} \text{ such that } x(s,a) = 0. \tag{30}$$

We claim that this implies

$$\exists (s_0, a_0) \in \mathcal{S} \times \mathcal{A} \quad \text{such that} \quad x(s_0, a_0) = 0, \quad \forall \mathbf{x} \in \mathcal{X}_{\widetilde{\boldsymbol{\eta}}}. \tag{31}$$

To prove the claim, assume by contradiction that Equation (31) is false. Then, for every $(s,a)$, there exists $\mathbf{x}^{s,a} \in \mathcal{X}_{\widetilde{\boldsymbol{\eta}}}$ such that $x^{s,a}(s,a) > 0$. Since $\mathcal{X}_{\widetilde{\boldsymbol{\eta}}}$ is convex,

$$\bar{\mathbf{x}} := \frac{1}{|\mathcal{S}|\,|\mathcal{A}|} \sum_{(s,a) \in \mathcal{S} \times \mathcal{A}} \mathbf{x}^{s,a} \in \mathcal{X}_{\widetilde{\boldsymbol{\eta}}}. \tag{32}$$

For any fixed $(\bar{s}, \bar{a})$,

$$\bar{x}(\bar{s}, \bar{a}) = \frac{1}{|\mathcal{S}|\,|\mathcal{A}|} \sum_{(s,a)} x^{s,a}(\bar{s}, \bar{a}) \geqslant \frac{1}{|\mathcal{S}|\,|\mathcal{A}|} x^{\bar{s},\bar{a}}(\bar{s}, \bar{a}) > 0, \tag{33}$$

contradicting Equation (30). Hence Equation (31) holds. Since $\widetilde{\boldsymbol{\eta}}(s_0) > 0$, for every $\boldsymbol{\pi} \in \Pi_{\widehat{\boldsymbol{\eta}}}$ and the corresponding $\mathbf{x} = \Phi(\boldsymbol{\pi})$,

$$\boldsymbol{\pi}(a_0|s_0) = \frac{x(s_0, a_0)}{\widetilde{\boldsymbol{\eta}}(s_0)} = 0. \tag{34}$$

Thus $\Pi_{\widehat{\boldsymbol{\eta}}}$ is contained in the proper face of the policy simplex defined by $\boldsymbol{\pi}(a_0|s_0) = 0$. Equivalently, $\mathcal{X}_{\widetilde{\boldsymbol{\eta}}}$ is contained in the coordinate face $\{\mathbf{x} : x(s_0, a_0) = 0\}$. More generally, let

$$I_0 := \big\{(s,a) \in \mathcal{S} \times \mathcal{A} : x(s,a) = 0, \ \forall \mathbf{x} \in \mathcal{X}_{\widetilde{\boldsymbol{\eta}}}\big\} \tag{35}$$

denote the set of coordinates that are active for all feasible occupancies. Then

$$\mathcal{X}_{\widetilde{\boldsymbol{\eta}}} \subseteq \mathcal{L}_{\widetilde{\boldsymbol{\eta}}} \cap \bigcap_{(s,a) \in I_0} \big\{\mathbf{x} \in \mathbb{R}^{|\mathcal{S}||\mathcal{A}|} : x(s,a) = 0\big\}. \tag{36}$$

If $\mathbf{E}_{I_0}$ denotes the coordinate-selection matrix associated with $I_0$, this implies

$$\dim(\mathcal{X}_{\widehat{\boldsymbol{\eta}}}) \leqslant \dim\big(\mathcal{L}_{\widetilde{\boldsymbol{\eta}}} \cap \ker \mathbf{E}_{I_0}\big) = \dim\big(\ker \mathbf{C} \cap \ker \mathbf{E}_{I_0}\big) \leqslant \dim(\ker \mathbf{C}). \tag{37}$$

Hence the active nonnegativity constraints may strictly reduce the dimension relative to the affine lower bound in Equation (25). In particular, the dimension can be zero. Finally, assume $\Pi_{\widehat{\boldsymbol{\eta}}} = \{\boldsymbol{\pi}\}$. Let $\mathbf{x} \in \mathcal{X}_{\widehat{\boldsymbol{\eta}}}$. By Equation (20), $\mathbf{x}$ induces a feasible policy $\Psi(\mathbf{x}) \in \Pi_{\widehat{\boldsymbol{\eta}}}$, and therefore $\Psi(\mathbf{x}) = \boldsymbol{\pi}$. Hence, for every $(s,a)$,

$$x(s,a) = \widetilde{\boldsymbol{\eta}}(s)[\Psi(\mathbf{x})](a|s) = \widetilde{\boldsymbol{\eta}}(s)\boldsymbol{\pi}(a|s). \tag{38}$$

Thus $\mathbf{x}$ is uniquely determined and equals $\Phi(\boldsymbol{\pi})$. $\qquad\square$

# C. Sample Complexity of Transition Model Estimation

In this appendix, we provide the full derivation of the complexity analysis of the offline learning setting introduced in Section 5. We first provide the preliminary concentration lemmas (Appendix C.1) and, then, move to the proof of the main result (Appendix C.2).

## C.1. Concentration Lemmas

Define the good event $\mathcal{E} := \mathcal{E}_1 \cap \mathcal{E}_2 \cap \mathcal{E}_3$ as the intersection of the events $\mathcal{E}_1$, which guarantees full coverage of $\mathcal{S} \times \mathcal{A}$ under $\pi^b$ and $\mathcal{E}_2$ and $\mathcal{E}_3$ which allow the concentration of the rows of $\check{\mathbf{P}}$, (i.e., the transition kernel derived from the counts as in Line 8) to their expected values $\mathbf{P}$. Let us first introduce formally the $\zeta$-*mixing time* for a MC with kernel $\mathbf{X}$ (Levin & Peres, 2017):

$$t_{\mathrm{mix}}(\zeta; \mathbf{X}) := \min\{t \in \mathbb{N} \mid d(t; \mathbf{X}) \leqslant \zeta\}. \tag{39}$$

$t_{\mathrm{mix}}(\zeta; \mathbf{X})$ represents the minimum number of steps needed so that the distance from equilibrium falls below the threshold $\zeta$.

We begin by providing an upper bound on the sample complexity (i.e., the trajectory length) required to have full coverage of the space $\mathcal{S} \times \mathcal{A}$.

**Lemma C.1.** *(Full-coverage) Let $\mathcal{M}$ be an ergodic MDP (Assumption 2.1) and $\pi^b \in \Pi^{\mathrm{MR}}$ be a policy satisfying Assumption 5.1. Let $\tau^b = (S_t, A_t)_{t \in [\![0, T-1]\!]}$ be a trajectory of length $T \in \mathbb{N}$ collected through $\pi^b$ and denote with $N^b(s, a)$ the visitation count for every state-action pair $(s, a) \in \mathcal{S} \times \mathcal{A}$. Finally, let $\eta^b_{\min} = \min_{s \in \mathcal{S}} \eta^{\pi^b}(s) > 0$ and let $t^b_{\mathrm{mix}} = \min\{t \geqslant 0 | d(t; \mathbf{P}^{\pi^b}) \leqslant 1/8\}$ be the mixing time of the MC induced by $\pi^b$. Then, for every $\delta \in (0, 1)$, event $\mathcal{E}_1$ defined as:*

$$\mathcal{E}_1 := \left\{ N^b(s, a) \geqslant 1 \quad \forall (s, a) \in \mathcal{S} \times \mathcal{A} \right\}, \tag{40}$$

*holds with probability at least $1 - \frac{\delta}{2}$ whenever:*

$$T \geqslant \frac{72(1 + t^b_{\mathrm{mix}})}{\eta^b_{\min}\xi} \ln\left(\frac{c\,|\mathcal{S}|\,|\mathcal{A}|}{\delta\sqrt{\eta^b_{\min}}}\right), \tag{41}$$

*where $c > 0$ is a universal constant.*

*Proof.* Consider a state-action pair $(s, a) \in \mathcal{S} \times \mathcal{A}$ and recall that $N^b(s, a) := \sum_{t=0}^{T-1} \mathbf{1}\{(S_t, A_t) = (s, a)\}$. By Assumption 5.1, $(S_t, A_t)_{t \geqslant 0}$ is ergodic and we denote its equilibrium distribution as $\mathbf{x}^b$ defined as $x^b(s, a) = \eta^b(s)\pi^b(a|s)$ for every $(s, a) \in \mathcal{S} \times \mathcal{A}$, being $\eta^b$ the equilibrium distribution of $\mathbf{P}^{\pi^b}$. Moreover, recall its underlying probability measure denoted as $\mathbb{P}^{\mathbf{P}, \pi^b}_{\boldsymbol{\mu}_0}$. Then we have:

$$\mathbb{P}^{\mathbf{P}, \pi^b}_{\boldsymbol{\mu}_0}(\mathcal{E}_1) = \mathbb{P}^{\mathbf{P}, \pi^b}_{\boldsymbol{\mu}_0}\left( \bigcap_{(s,a) \in \mathcal{S} \times \mathcal{A}} \{N^b(s, a) \geqslant 1\} \right) \tag{42}$$

$$= 1 - \mathbb{P}^{\mathbf{P}, \pi^b}_{\boldsymbol{\mu}_0}\left( \bigcup_{(s,a) \in \mathcal{S} \times \mathcal{A}} \{N^b(s, a) = 0\} \right) \tag{43}$$

$$\geqslant 1 - \sum_{(s,a) \in \mathcal{S} \times \mathcal{A}} \mathbb{P}^{\mathbf{P}, \pi^b}_{\boldsymbol{\mu}_0}(N^b(s, a) = 0) \tag{44}$$

$$= 1 - \sum_{(s,a) \in \mathcal{S} \times \mathcal{A}} \mathbb{P}^{\mathbf{P}, \pi^b}_{\boldsymbol{\mu}_0}(N^b(s, a) \leqslant 0) \tag{45}$$

$$\geqslant 1 - \sum_{(s,a) \in \mathcal{S} \times \mathcal{A}} c\|(\mathbf{\Pi}^b)^\top \boldsymbol{\mu}_0\|_{\ell^2(1/\mathbf{x}^b)} \exp\left(-x^b(s, a)T/72t_{\mathrm{mix}}(1/8; \mathbf{P}\mathbf{\Pi}^b)\right) \tag{46}$$

$$\geqslant 1 - c\,|\mathcal{S}|\,|\mathcal{A}|\,\|(\mathbf{\Pi}^b)^\top \boldsymbol{\mu}_0\|_{\ell^2(1/\mathbf{x}^b)} \exp\left(-\eta^b_{\min}\xi T/72t_{\mathrm{mix}}(1/8; \mathbf{P}\mathbf{\Pi}^b)\right), \tag{47}$$

where line (44) comes from a union bound; line (45) derives from the fact that $N^b(s,a) \in \{0, 1, \ldots, T\}$; line (46) comes from Theorem E.1 with $f(Z_t) = \mathbf{1}\{(S_t, A_t) = (s,a)\}$, and the fact that $\mathbb{E}_{\boldsymbol{x}^b}[\mathbf{1}\{(S_t, A_t) = (s,a)\}] = x^b(s,a)$. Finally the last line is obtained having defined $x^b_{\min} = \min_{(s,a) \in \mathcal{S} \times \mathcal{A}} x^b(s,a) \geqslant \eta^b_{\min} \xi$.

Furthermore, we enforce $c |\mathcal{S}| |\mathcal{A}| \|(\boldsymbol{\Pi}^b)^\top \boldsymbol{\mu}_0\|_{\ell^2(1/\mathbf{x}^b)} \exp\left(-\eta^b_{\min} \xi T / 72 t_{\mathrm{mix}}(1/8; \mathbf{P}\boldsymbol{\Pi}^b)\right) \leqslant \delta/2$, solving for $T$ and noticing that,

$$
\begin{aligned}
\|(\boldsymbol{\Pi}^b)^\top \boldsymbol{\mu}_0\|_{\ell^2(1/\mathbf{x}^b)} &:= \sqrt{\langle \boldsymbol{\mu}_0^\top \boldsymbol{\Pi}^b, (\boldsymbol{\Pi}^b)^\top \boldsymbol{\mu}_0 \rangle_{1/\mathbf{x}^b}} \\
&= \sqrt{\sum_{(s,a) \in \mathcal{S} \times \mathcal{A}} \frac{\mu_0^2(s)(\pi^b(a|s))^2}{\eta^b(s)\pi^b(a|s)}} \\
&\leqslant \sqrt{\sum_{(s,a) \in \mathcal{S} \times \mathcal{A}} \frac{\mu_0(s)\pi^b(a|s)}{\eta^b(s)}} \\
&\leqslant \sqrt{\frac{1}{\eta^b_{\min}}}.
\end{aligned}
$$

Finally, we conclude the proof by observing that for every $t \geqslant 1$, we have:

$$
d(t; \mathbf{P}\boldsymbol{\Pi}^b) = \frac{1}{2} \|(\mathbf{P}\boldsymbol{\Pi}^b)^t - \mathbf{1}(\boldsymbol{x}^b)^\top\|_\infty \tag{48}
$$

$$
= \frac{1}{2} \|\mathbf{P}(\boldsymbol{\Pi}^b \mathbf{P})^{t-1} \boldsymbol{\Pi}^b - \mathbf{1}(\boldsymbol{\eta}^b)^\top \boldsymbol{\Pi}^b\|_\infty \tag{49}
$$

$$
\leqslant \frac{1}{2} \|\mathbf{P}(\boldsymbol{\Pi}^b \mathbf{P})^{t-1} - \mathbf{1}(\boldsymbol{\eta}^b)^\top\|_\infty \|\boldsymbol{\Pi}^b\|_\infty \tag{50}
$$

$$
\leqslant \frac{1}{2} \max_{s \in \mathcal{S}} \|\mathbf{p}^{\boldsymbol{\pi}^b, t-1}(\cdot|s) - \boldsymbol{\eta}^b\|_1 = d(t-1; \mathbf{P}^{\boldsymbol{\pi}^b}), \tag{51}
$$

having observed that $\|\boldsymbol{\Pi}^b\|_\infty = 1$. From this, it follows that $t_{\mathrm{mix}}(1/8; \mathbf{P}\boldsymbol{\Pi}^b) \leqslant 1 + t_{\mathrm{mix}}(1/8; \mathbf{P}^{\boldsymbol{\pi}^b})$. $\square$

Before presenting the concentration lemmas for $\check{\mathbf{P}}$, we define the symbol:

$$
\beta(n, \delta) := \ln(4 |\mathcal{S}| |\mathcal{A}| / \delta) + (|\mathcal{S}| - 1) \ln(e(1 + n/(|\mathcal{S}| - 1))). \tag{52}
$$

**Lemma C.2** (Concentration). *Let $\mathcal{M}$ be a tabular and ergodic MDP (Assumption 2.1) and let $\boldsymbol{\pi}^b \in \Pi^{\mathrm{MR}}$ be a policy satisfying Assumption 5.1 inducing the steady-state distribution $\boldsymbol{\eta}^b$. Let $\tau^b = (S_t, A_t)_{t \in [\![0, T-1]\!]}$ be a trajectory of length $T \in \mathbb{N}$ collected through $\boldsymbol{\pi}^b$. Let $\eta^b_{\min} = \min_{s \in \mathcal{S}} \eta^{\boldsymbol{\pi}^b}(s) > 0$ and let $t^b_{\mathrm{mix}} = \min\{t \geqslant 0 | d(t; \mathbf{P}^{\boldsymbol{\pi}^b}) \leqslant 1/8\}$ be the mixing time of the MC induced by $\boldsymbol{\pi}^b$. Then, for every $\delta \in (0, 1)$, event $\mathcal{E}_2 \cap \mathcal{E}_3$, where $\mathcal{E}_2$ and $\mathcal{E}_3$ are defined as:*

$$
\mathcal{E}_2 := \left\{ N^b(s,a) \mathcal{D}_{\mathrm{KL}}(\check{\mathbf{p}}(\cdot|s,a) \| \mathbf{p}(\cdot|s,a)) \leqslant \beta(N^b(s,a), \delta), \ \forall T \in \mathbb{N}, \forall (s,a) \in \mathcal{S} \times \mathcal{A} \right\}, \tag{53}
$$

$$
\mathcal{E}_3 := \left\{ \frac{1}{\max\{N^b(s,a), 1\}} \leqslant \frac{288}{x^b(s,a)} \frac{1 + t^b_{\mathrm{mix}}}{T} \ln\left( \frac{4c |\mathcal{S}| |\mathcal{A}|}{\delta \sqrt{\eta^b_{\min}}} \right), \ \forall (s,a) \in \mathcal{S} \times \mathcal{A} \right\}, \tag{54}
$$

*holds with probability $1 - \frac{\delta}{2}$.*

*Proof.* We follow the main rationale as in (Lazzati et al., 2024, Lemma F.2), thus, we show that each event holds with probability at least $1 - \frac{\delta}{4}$, and complete the argument through a union bound.

*Event $\mathcal{E}_3$, lower bounding $N^b(s,a)$:* Consider a $(s,a) \in \mathcal{S} \times \mathcal{A}$ and recall the definition $N^b(s,a) := \sum_{t=0}^{T-1} \mathbf{1}\{(S_t, A_t) = (s,a)\}$. We then apply Theorem E.1 with $\varepsilon = 1/2$, similarly to (Xie et al., 2021, Lemma A.1), so that to have:

$$
\mathbb{P}^{\mathbf{P}, \boldsymbol{\pi}^b}_{\boldsymbol{\mu}_0}\left( N^b(s,a) \leqslant \frac{x^b(s,a)}{2} T \right) \leqslant c \|(\boldsymbol{\Pi}^b)^\top \boldsymbol{\mu}_0\|_{\ell^2(1/\mathbf{x}^b)} \exp(-x^b(s,a)T/288 t_{\mathrm{mix}}(1/8; \mathbf{P}\boldsymbol{\Pi}^b)) \tag{55}
$$

We then derive a lower bound to $x^b(s,a)$ by solving, for $\delta \in (0,1)$,

$$c\|(\mathbf{\Pi}^b)^\top \boldsymbol{\mu}_0\|_{\ell^2(1/\mathbf{x}^b)} \exp(-x^b(s,a)T/288 t_{\mathrm{mix}}(1/8; \mathbf{P}\mathbf{\Pi}^b)) \leqslant \frac{\delta}{4|\mathcal{S}||\mathcal{A}|} \tag{56}$$

$$\iff \quad \exp\left(-\frac{x^b(s,a)T}{288 t_{\mathrm{mix}}(1/8; \mathbf{P}\mathbf{\Pi}^b)}\right) \leqslant \frac{\delta}{4|\mathcal{S}||\mathcal{A}| c\|(\mathbf{\Pi}^b)^\top \boldsymbol{\mu}_0\|_{\ell^2(1/\mathbf{x}^b)}} \tag{57}$$

$$\iff \quad \frac{x^b(s,a)T}{288 t_{\mathrm{mix}}(1/8; \mathbf{P}\mathbf{\Pi}^b)} \geqslant \ln\left(\frac{4|\mathcal{S}||\mathcal{A}| c\|(\mathbf{\Pi}^b)^\top \boldsymbol{\mu}_0\|_{\ell^2(1/\mathbf{x}^b)}}{\delta}\right) \tag{58}$$

$$\iff \quad x^b(s,a) \geqslant 288 \frac{t_{\mathrm{mix}}(1/8; \mathbf{P}\mathbf{\Pi}^b)}{T} \ln\left(\frac{4|\mathcal{S}||\mathcal{A}| c\|(\mathbf{\Pi}^b)^\top \boldsymbol{\mu}_0\|_{\ell^2(1/\mathbf{x}^b)}}{\delta}\right) \tag{59}$$

Furthermore, let us observe that:

$$\|(\mathbf{\Pi}^b)^\top \boldsymbol{\mu}_0\|_{\ell^2(1/\mathbf{x}^b)} := \sqrt{\langle \boldsymbol{\mu}_0^\top \mathbf{\Pi}^b, (\mathbf{\Pi}^b)^\top \boldsymbol{\mu}_0 \rangle_{1/\mathbf{x}^b}}$$

$$= \sqrt{\sum_{(s,a)\in\mathcal{S}\times\mathcal{A}} \frac{\mu_0^2(s)(\pi^b(a|s))^2}{\eta^b(s)\pi^b(a|s)}}$$

$$\leqslant \sqrt{\sum_{(s,a)\in\mathcal{S}\times\mathcal{A}} \frac{\mu_0(s)\pi^b(a|s)}{\eta^b(s)}}$$

$$\leqslant \sqrt{\frac{1}{\eta^b_{\min}}}.$$

Thus, leveraging the result in Equation (39) we can write a more demanding condition on $x^b(s,a)$ as:

$$x^b(s,a) \geqslant 288 \frac{1 + t_{\mathrm{mix}}(1/8; \mathbf{P}^{\boldsymbol{\pi}^b})}{T} \ln\left(\frac{4c|\mathcal{S}||\mathcal{A}|}{\delta\sqrt{\eta^b_{\min}}}\right). \tag{60}$$

Suppose now that $x^b(s,a) < 288 \frac{1 + t_{\mathrm{mix}}(1/8; \mathbf{P}^{\boldsymbol{\pi}^b})}{T} \ln\left(\frac{4c|\mathcal{S}||\mathcal{A}|}{\delta\sqrt{\eta^b_{\min}}}\right)$. In such a case:

$$\frac{x^b(s,a)}{\max\{N^b(s,a), 1\}} \leqslant x^b(s,a) \leqslant 288 \frac{1 + t_{\mathrm{mix}}(1/8; \mathbf{P}^{\boldsymbol{\pi}^b})}{T} \ln\left(\frac{4c|\mathcal{S}||\mathcal{A}|}{\delta\sqrt{\eta^b_{\min}}}\right). \tag{61}$$

Instead, if it holds that $x^b(s,a) \geqslant 288 \frac{1 + t_{\mathrm{mix}}(1/8; \mathbf{P}^{\boldsymbol{\pi}^b})}{T} \ln\left(\frac{4c|\mathcal{S}||\mathcal{A}|}{\delta\sqrt{\eta^b_{\min}}}\right)$. Then, with probability at least $1 - \delta/(4|\mathcal{S}||\mathcal{A}|)$, we have that:

$$\frac{x^b(s,a)}{\max\{N^b(s,a), 1\}} \leqslant \frac{x^b(s,a)}{N^b(s,a)} \leqslant \frac{2}{T} \leqslant 288 \frac{1 + t_{\mathrm{mix}}(1/8; \mathbf{P}^{\boldsymbol{\pi}^b})}{T} \ln\left(\frac{4c|\mathcal{S}||\mathcal{A}|}{\delta\sqrt{\eta^b_{\min}}}\right). \tag{62}$$

Thus, combining both cases, we have:

$$\mathbb{P}^{\mathbf{P}, \boldsymbol{\pi}^b}_{\boldsymbol{\mu}_0}\left(\frac{1}{\max\{N^b(s,a), 1\}} \leqslant \frac{288}{x^b(s,a)} \frac{1 + t_{\mathrm{mix}}(1/8; \mathbf{P}^{\boldsymbol{\pi}^b})}{T} \ln\left(\frac{4c|\mathcal{S}||\mathcal{A}|}{\delta\sqrt{\eta^b_{\min}}}\right)\right) \geqslant 1 - \frac{\delta}{4|\mathcal{S}||\mathcal{A}|} \tag{63}$$

We conclude the proof by applying a union bound to get a condition holding uniformly over $\mathcal{S} \times \mathcal{A}$:

$$\mathbb{P}^{\mathbf{P},\boldsymbol{\pi}^b}_{\boldsymbol{\mu}_0} \left( \bigcap_{(s,a)\in\mathcal{S}\times\mathcal{A}} \left\{ \frac{1}{\max\{N^b(s,a),1\}} \leqslant \frac{288}{x^b(s,a)} \frac{1+t_{\mathrm{mix}}(1/8;\mathbf{P}^{\boldsymbol{\pi}^b})}{T} \ln\left( \frac{4c\,|\mathcal{S}|\,|\mathcal{A}|}{\delta\sqrt{\eta^b_{\min}}} \right) \right\} \right) \tag{64}$$

$$= 1 - \mathbb{P}^{\mathbf{P},\boldsymbol{\pi}^b}_{\boldsymbol{\mu}_0} \left( \bigcup_{(s,a)\in\mathcal{S}\times\mathcal{A}} \left\{ \frac{1}{\max\{N^b(s,a),1\}} \geqslant \frac{288}{x^b(s,a)} \frac{1+t_{\mathrm{mix}}(1/8;\mathbf{P}^{\boldsymbol{\pi}^b})}{T} \ln\left( \frac{4c\,|\mathcal{S}|\,|\mathcal{A}|}{\delta\sqrt{\eta^b_{\min}}} \right) \right\} \right) \tag{65}$$

$$\geqslant 1 - \sum_{(s,a)\in\mathcal{S}\times\mathcal{A}} \mathbb{P}^{\mathbf{P},\boldsymbol{\pi}^b}_{\boldsymbol{\mu}_0} \left( \left\{ \frac{1}{\max\{N^b(s,a),1\}} \geqslant \frac{288}{x^b(s,a)} \frac{1+t_{\mathrm{mix}}(1/8;\mathbf{P}^{\boldsymbol{\pi}^b})}{T} \ln\left( \frac{4c\,|\mathcal{S}|\,|\mathcal{A}|}{\delta\sqrt{\eta^b_{\min}}} \right) \right\} \right) \geqslant 1 - \frac{\delta}{4}. \tag{66}$$

*Event $\mathcal{E}_2$, concentration of $\mathcal{D}_{KL}$:* To prove that $\mathcal{E}_2$ holds with probability at least $1 - \frac{\delta}{4}$, we proceed as follows:

$$\mathbb{P}^{\mathbf{P},\boldsymbol{\pi}^b}_{\boldsymbol{\mu}_0}(\mathcal{E}_2^{\mathsf{C}}) = \mathbb{P}^{\mathbf{P},\boldsymbol{\pi}^b}_{\boldsymbol{\mu}_0} \left( \bigcup_{(s,a)\in\mathcal{S}\times\mathcal{A}} \left\{ \exists\, T \in \mathbb{N} \mid N^b(s,a)\mathcal{D}_{KL}(\check{\mathbf{p}}(\cdot|s,a)\|\mathbf{p}(\cdot|s,a)) \geqslant \beta(N^b(s,a),\delta) \right\} \right) \tag{67}$$

$$\leqslant \sum_{(s,a)\in\mathcal{S}\times\mathcal{A}} \mathbb{P}^{\mathbf{P},\boldsymbol{\pi}^b}_{\boldsymbol{\mu}_0} \left( \left\{ \exists\, T \in \mathbb{N} \mid N^b(s,a)\mathcal{D}_{KL}(\check{\mathbf{p}}(\cdot|s,a)\|\mathbf{p}(\cdot|s,a)) \geqslant \beta(N^b(s,a),\delta) \right\} \right) \tag{68}$$

$$\leqslant |\mathcal{S}|\,|\mathcal{A}| \frac{\delta}{4\,|\mathcal{S}|\,|\mathcal{A}|} = \frac{\delta}{4}, \tag{69}$$

where line (69) derives from applying Lemma E.2. $\qquad\square$

## C.2. Proof of Theorem 5.1

**Theorem 5.1** (Sample Complexity Guarantees). *Let $\mathcal{M}$ be a tabular and ergodic MDP (Assumption 2.1) and let $\boldsymbol{\pi}^b \in \Pi^{\mathrm{MR}}$ be a policy satisfying Assumption 5.1 inducing the steady-state distribution $\boldsymbol{\eta}^b$. Let $\tau^b = (S_t, A_t, S_{t+1})_{t\in[\![0,T-1]\!]}$ be a trajectory of length $T \in \mathbb{N}$ collected through $\boldsymbol{\pi}^b$. Let $\eta^b_{\min} = \min_{s\in\mathcal{S}} \eta^{\boldsymbol{\pi}^b}(s) > 0$ and let $t^b_{\mathrm{mix}} = \min\{t \geqslant 0 | d(t;\mathbf{P}^{\boldsymbol{\pi}^b}) \leqslant 1/8\}$ be the mixing time of the MC induced by $\boldsymbol{\pi}^b$. Let $\varepsilon > 0$ and $\delta \in (0,1)$. Let $\widehat{\mathbf{P}}$ denote the estimated transition kernel as in Line 10, with $\upsilon \leqslant \varepsilon/2\sqrt{|\mathcal{S}|\,|\mathcal{A}|}$. Then, with probability at least $1 - \delta$, to guarantee that:*

$$\left\| \widehat{\mathbf{P}} - \mathbf{P} \right\|_2 \leqslant 2\varepsilon, \tag{9}$$

*the sample complexity (i.e., length of the trajectory) is at most:*

$$T \leqslant \widetilde{O}\left( \frac{(1+t^b_{\mathrm{mix}})|\mathcal{S}||\mathcal{A}|\ln\left(\frac{1}{\delta}\right)}{\eta^b_{\min}\xi\varepsilon^2} \left( \ln\left(\frac{1}{\delta}\right) + |\mathcal{S}| \right) \right). \tag{10}$$

*Proof.* We divide the proof into two parts. First, we control the approximation introduced by the mixture with a uniform transition kernel (i.e, Line 10) to enforce ergodicity, and then we control the estimation error derived from the trajectory data (i.e., Line 8). In the following, we denote with the matrix $\mathbf{J} \in \mathbb{R}^{|\mathcal{S}||\mathcal{A}|\times|\mathcal{S}|}$ the uniform transition kernel, such that $p(s'|s,a) = 1/|\mathcal{S}|$ for every $(s,a,s') \in \mathcal{S} \times \mathcal{A} \times \mathcal{S}$.

Let us proceed by observing that:

$$\|\widehat{\mathbf{P}} - \mathbf{P}\|_2 \leqslant \underbrace{\|\widehat{\mathbf{P}} - \check{\mathbf{P}}\|_2}_{\text{Term I}} + \underbrace{\|\check{\mathbf{P}} - \mathbf{P}\|_2}_{\text{Term II}}. \tag{70}$$

$$\tag{71}$$

In what follows, we will prove that both Term I and Term II can be made smaller than $\varepsilon$. The result will then follow by putting together the analysis of these two separate terms.

For Term I, observe that:

$$\|\widehat{\mathbf{P}} - \check{\mathbf{P}}\|_2 = \|(1-\upsilon)\check{\mathbf{P}} + \upsilon\mathbf{J} - \check{\mathbf{P}}\|_2 \tag{72}$$

$$= \upsilon\|\mathbf{J} - \check{\mathbf{P}}\|_2 \tag{73}$$

$$\leqslant \upsilon(\|\mathbf{J}\|_2 + \|\check{\mathbf{P}}\|_2) \tag{74}$$

$$\leqslant 2\upsilon\sqrt{|\mathcal{S}|\,|\mathcal{A}|}, \tag{75}$$

where line (73) derives from the triangle inequality, while line (75) derives from the properties of the spectral norm of any row stochastic matrix, namely $\|\mathbf{P}\|_2 \leqslant \sqrt{\|\mathbf{P}\|_1\|\mathbf{P}\|_\infty} \leqslant \sqrt{|\mathcal{S}|\,|\mathcal{A}|}$. We then enforce $2\upsilon\sqrt{|\mathcal{S}|\,|\mathcal{A}|} \leqslant \varepsilon$, retrieving the optimal value of $\upsilon$ to ensure the desired accuracy $\varepsilon$.

For Term II, let us consider the following derivation:

$$\left\|\check{\mathbf{P}} - \mathbf{P}\right\|_2 \leqslant 2\sqrt{|\mathcal{S}|\,|\mathcal{A}|} \max_{(s,a)\in\mathcal{S}\times\mathcal{A}} \sqrt{2\mathcal{D}_{\mathrm{KL}}(\check{\mathbf{p}}(\cdot|s,a)\|\mathbf{p}(\cdot|s,a))} \tag{76}$$

$$\leqslant 2\sqrt{|\mathcal{S}|\,|\mathcal{A}|} \max_{(s,a)\in\mathcal{S}\times\mathcal{A}} \sqrt{2\frac{\beta(N^b(s,a),\delta)}{N^b(s,a)}} \tag{77}$$

$$\leqslant 2\sqrt{|\mathcal{S}|\,|\mathcal{A}|} \max_{(s,a)\in\mathcal{S}\times\mathcal{A}} \sqrt{2\frac{\beta(T,\delta)}{N^b(s,a)}} \tag{78}$$

$$\leqslant 2\sqrt{|\mathcal{S}|\,|\mathcal{A}|}\sqrt{2\beta(T,\delta)} \max_{(s,a)\in\mathcal{S}\times\mathcal{A}} \sqrt{288\frac{(1+t^b_{\mathrm{mix}})}{T}\frac{1}{x^b(s,a)}\ln\left(\frac{4c\,|\mathcal{S}|\,|\mathcal{A}|}{\delta\sqrt{\eta^b_{\min}}}\right)} \tag{79}$$

$$\leqslant 2\sqrt{\frac{288(1+t^b_{\mathrm{mix}})\,|\mathcal{S}|\,|\mathcal{A}|}{\eta^b_{\min}\xi}}\sqrt{2\frac{\beta(T,\delta)}{T}\ln\left(\frac{4c\,|\mathcal{S}|\,|\mathcal{A}|}{\delta\sqrt{\eta^b_{\min}}}\right)}, \tag{80}$$

where in line (76) we applied Lemma E.4; line (77) holds under the good event $\mathcal{E} = \mathcal{E}_1 \cap \mathcal{E}_2 \cap \mathcal{E}_3$ that holds with probability at least $1 - \delta$, as proved in Lemma C.1 and in Lemma C.2; line (78) comes from observing that $T \geqslant N^b(s,a)$ and noticing that, by definition, $\beta(N^b(s,a),\delta) := \ln(4\,|\mathcal{S}|\,|\mathcal{A}|\,/\delta) + (|\mathcal{S}| - 1)\ln(e(1 + N^b(s,a)/(|\mathcal{S}| - 1)))$ is non-decreasing in the first argument; line (79) is obtained using event $\mathcal{E}_3$ to rewrite the ratio $1/N^b(s,a)$; finally, the last inequality derives from the fact that $x^b(s,a) \geqslant \eta^b_{\min}\xi$ for every $(s,a) \in \mathcal{S} \times \mathcal{A}$.

We move on by introducing for ease of notation the following constants $c_1 = 2\sqrt{\frac{288(1+t^b_{\mathrm{mix}})|\mathcal{S}||\mathcal{A}|}{\eta^b_{\min}\xi}}$, and $c_2 = 2c_1^2$.

$$2\sqrt{\frac{288(1+t^b_{\mathrm{mix}})\,|\mathcal{S}|\,|\mathcal{A}|}{\eta^b_{\min}\xi}}\sqrt{2\frac{\beta(T,\delta)}{T}\ln\left(\frac{4c\,|\mathcal{S}|\,|\mathcal{A}|}{\delta\sqrt{\eta^b_{\min}}}\right)} \leqslant \varepsilon \quad\Longleftrightarrow\quad c_2\frac{\beta(T,\delta)}{T}\ln\left(\frac{4c\,|\mathcal{S}|\,|\mathcal{A}|}{\delta\sqrt{\eta^b_{\min}}}\right) \leqslant \varepsilon^2. \tag{81}$$

Solving for $T$, we get:

$$T \geqslant \frac{c_2}{\varepsilon^2}\ln\left(\frac{4c\,|\mathcal{S}|\,|\mathcal{A}|}{\delta\sqrt{\eta^b_{\min}}}\right)\left(\ln\left(\frac{4\,|\mathcal{S}|\,|\mathcal{A}|}{\delta}\right) + (|\mathcal{S}| - 1)\ln\left(\frac{e}{|\mathcal{S}| - 1}T + e\right)\right) \tag{82}$$

To find the minimum value of $T$ satisfying Equation (82), we apply Lemma E.3 with:

$$a := \frac{c_2 \ln\left(\frac{4c|\mathcal{S}||\mathcal{A}|}{\delta\sqrt{\eta_{\min}^b}}\right) \ln\left(\frac{4|\mathcal{S}||\mathcal{A}|}{\delta}\right)}{\varepsilon^2}, \tag{83}$$

$$b := \frac{c_2 \ln\left(\frac{4c|\mathcal{S}||\mathcal{A}|}{\delta\sqrt{\eta_{\min}^b}}\right)}{\varepsilon^2}(|\mathcal{S}|-1), \tag{84}$$

$$c := \frac{e}{|\mathcal{S}|-1}, \tag{85}$$

$$d := e. \tag{86}$$

$$\tag{87}$$

We obtain:

$$T \geqslant 2c_2 \frac{\ln\left(\frac{4c|\mathcal{S}||\mathcal{A}|}{\delta\sqrt{\eta_{\min}^b}}\right) \ln\left(\frac{4|\mathcal{S}||\mathcal{A}|}{\delta}\right)}{\varepsilon^2} + (|\mathcal{S}|-1) + 3c_2 \frac{\ln\left(\frac{4c|\mathcal{S}||\mathcal{A}|}{\delta\sqrt{\eta_{\min}^b}}\right)}{\varepsilon^2}(|\mathcal{S}|-1)\ln\left(\frac{2c_2}{\varepsilon^2}\ln\left(\frac{4c|\mathcal{S}||\mathcal{A}|}{\delta\sqrt{\eta_{\min}^b}}\right)e\right). \tag{88}$$

Finally, replacing $c_2$ with its definition we get:

$$T \leqslant \tilde{O}\left(\frac{(1+t_{\mathrm{mix}}^b)|\mathcal{S}||\mathcal{A}|\ln\left(\frac{1}{\delta}\right)}{\epsilon^2\eta_{\min}^b\xi}\left(\ln\left(\frac{1}{\delta}\right)+|\mathcal{S}|\right)\right). \tag{89}$$

We conclude the analysis by noticing that the above dependence dominates the upper-bound on $T$ derived from event $\mathcal{E}_1$. $\qquad\square$

# D. Error Propagation

In this appendix, we provide the analysis of the error propagation. We first review the involved optimization problems (Appendix D.1), we provide a perturbation result for general optimization problems (Appendix D.2), we analyze the error propagation for the projection step (Appendix D.3), and, finally, we analyze the error propagation for the FMSS problem (Appendix D.4).

## D.1. Preliminaries

Suppose we want to regulate the MDP towards the target distribution $\overline{\eta} \in \Delta(\mathcal{S})$. Let $\mathbf{P}, \widehat{\mathbf{P}} \in \mathbb{R}^{|\mathcal{S}||\mathcal{A}| \times |\mathcal{S}|}$ be two transition models. We first perform the *projection* step to obtain $\widetilde{\eta}, \widehat{\widetilde{\eta}} \in \Delta(\mathcal{S})$ by solving the two programs, respectively:



**Exact program: PROJ($\overline{\eta}$,P)**            **Perturbed program: PROJ($\overline{\eta}$,$\widehat{\mathbf{P}}$)**



$$
\begin{array}{ll}
\underset{\mathbf{x} \in \mathbb{R}^{|\mathcal{S}||\mathcal{A}|}}{\text{minimize}} & \mathcal{D}(\overline{\eta}\|\mathbf{Kx}) \\
\text{subject to} & \mathbf{Kx} = \mathbf{P}^\top \mathbf{x}, \\
& \mathbf{1}^\top \mathbf{x} = 1, \\
& \mathbf{x} \geqslant \mathbf{0}
\end{array} \tag{90}
$$

$$
\begin{array}{ll}
\underset{\mathbf{x} \in \mathbb{R}^{|\mathcal{S}||\mathcal{A}|}}{\text{minimize}} & \mathcal{D}(\overline{\eta}\|\mathbf{Kx}) \\
\text{subject to} & \mathbf{Kx} = \widehat{\mathbf{P}}^\top \mathbf{x}, \\
& \mathbf{1}^\top \mathbf{x} = 1, \\
& \mathbf{x} \geqslant \mathbf{0}
\end{array} \tag{91}
$$

We denote with $\widetilde{\mathbf{x}}$ and $\widehat{\widetilde{\mathbf{x}}}$ the solutions of the two programs which allow us to define $\widetilde{\eta} = \mathbf{K}\widetilde{\mathbf{x}}$ and $\widehat{\widetilde{\eta}} = \mathbf{K}\widehat{\widetilde{\mathbf{x}}}$, respectively.

Then, we solve the FMSS control problem by using the obtained projected distributions:



**Exact program: FMSS($\widetilde{\eta}, \mathbf{P}$)**         **Perturbed program: FMSS($\widehat{\widetilde{\eta}}, \widehat{\mathbf{P}}$)**



$$
\begin{array}{ll}
\underset{\boldsymbol{\pi} \in \Pi^{\mathrm{MR}}}{\text{minimize}} & \left\|\widetilde{\mathbf{A}}_{\widetilde{\eta}}^{\boldsymbol{\pi}}\right\|_2 \\
\text{subject to} & \widetilde{\eta} = (\mathbf{P}^{\boldsymbol{\pi}})^\top \widetilde{\eta}
\end{array} \tag{92}
$$

$$
\begin{array}{ll}
\underset{\boldsymbol{\pi} \in \Pi^{\mathrm{MR}}}{\text{minimize}} & \left\|\widehat{\widetilde{\mathbf{A}}}_{\widehat{\widetilde{\eta}}}^{\boldsymbol{\pi}}\right\|_2 \\
\text{subject to} & \widehat{\widetilde{\eta}} = (\widehat{\mathbf{P}}^{\boldsymbol{\pi}})^\top \widehat{\widetilde{\eta}}
\end{array} \tag{93}
$$

where $\widetilde{\mathbf{A}}_{\widetilde{\eta}}^{\boldsymbol{\pi}} = \mathbf{D}_{\widetilde{\eta}}^{1/2}\mathbf{P}^{\boldsymbol{\pi}}\mathbf{D}_{\widetilde{\eta}}^{-1/2} - \sqrt{\widetilde{\eta}}\sqrt{\widetilde{\eta}}^\top$, $\widehat{\widetilde{\mathbf{A}}}_{\widehat{\widetilde{\eta}}}^{\boldsymbol{\pi}} = \mathbf{D}_{\widehat{\widetilde{\eta}}}^{1/2}\widehat{\mathbf{P}}^{\boldsymbol{\pi}}\mathbf{D}_{\widehat{\widetilde{\eta}}}^{-1/2} - \sqrt{\widehat{\widetilde{\eta}}}\sqrt{\widehat{\widetilde{\eta}}}^\top$, $\mathbf{P}^{\boldsymbol{\pi}} = \Pi\mathbf{P}$, $\widehat{\mathbf{P}}^{\boldsymbol{\pi}} = \Pi\widehat{\mathbf{P}}$. They lead to the solutions $\boldsymbol{\pi}^*, \widehat{\boldsymbol{\pi}}^* \in \Pi^{\mathrm{MR}}$, respectively.

For a policy $\boldsymbol{\pi} \in \Pi^{\mathrm{MR}}$, we denote with $\eta^{\boldsymbol{\pi}}$ the steady-state distribution induced in the MC with kernel $\mathbf{P}$ and with $\widehat{\eta}^{\boldsymbol{\pi}}$ the steady-state distribution induced in the MC with kernel $\widehat{\mathbf{P}}$. Thus, we have that $\eta^{\boldsymbol{\pi}^*} = \widetilde{\eta}$ and $\widehat{\eta}^{\widehat{\boldsymbol{\pi}}^*} = \widehat{\widetilde{\eta}}$.

## D.2. General Perturbation Bound

In this appendix, we provide a general result for the perturbation of the objective function and feasible set of an optimization problem. We make use of the following standard Hausdorff distance between sets.

**Definition D.1** (Hausdorff Distance). *Let $\mathcal{X}, \widehat{\mathcal{X}} \subset \mathbb{R}^d$ be two sets. Let $d : \mathbb{R}^d \times \mathbb{R}^d \to \mathbb{R}_{\geqslant 0}$. The Hausdorff distance induced by the metric $d$ is defined as:*

$$
d_H(\mathcal{X}, \widehat{\mathcal{X}}) := \max \left\{ \sup_{\mathbf{x} \in \mathcal{X}} \inf_{\widehat{\mathbf{x}} \in \widehat{\mathcal{X}}} d(\mathbf{x}, \widehat{\mathbf{x}}), \sup_{\widehat{\mathbf{x}} \in \widehat{\mathcal{X}}} \inf_{\mathbf{x} \in \mathcal{X}} d(\mathbf{x}, \widehat{\mathbf{x}}) \right\}. \tag{94}
$$

The perturbation result is provided as follows.

**Lemma D.1.** *Let $f, \widehat{f} : \mathbb{R}^d \to \mathbb{R}$ be L-Lipschitz continuous functions w.r.t. the metric $d : \mathbb{R}^d \times \mathbb{R}^d \to \mathbb{R}_{\geqslant 0}$. Let $\mathcal{X}, \widehat{\mathcal{X}} \subseteq \mathbb{R}^d$ be two sets. Then, it holds that:*

$$
\left| \min_{\mathbf{x} \in \mathcal{X}} f(\mathbf{x}) - \min_{\mathbf{x} \in \widehat{\mathcal{X}}} \widehat{f}(\mathbf{x}) \right| \leqslant L d_H(\mathcal{X}, \widehat{\mathcal{X}}) + \min \left\{ \max_{\mathbf{x} \in \mathcal{X}} |f(\mathbf{x}) - \widehat{f}(\mathbf{x})|, \max_{\mathbf{x} \in \widehat{\mathcal{X}}} |f(\mathbf{x}) - \widehat{f}(\mathbf{x})| \right\}. \tag{95}
$$

*Proof.* Let us consider the following derivation:

$$
\left| \min_{\mathbf{x} \in \mathcal{X}} f(\mathbf{x}) - \min_{\mathbf{x} \in \widehat{\mathcal{X}}} \widehat{f}(\mathbf{x}) \right| \leqslant \left| \min_{\mathbf{x} \in \mathcal{X}} f(\mathbf{x}) - \min_{\mathbf{x} \in \mathcal{X}} \widehat{f}(\mathbf{x}) \right| + \left| \min_{\mathbf{x} \in \mathcal{X}} \widehat{f}(\mathbf{x}) - \min_{\mathbf{x} \in \widehat{\mathcal{X}}} \widehat{f}(\mathbf{x}) \right|. \tag{96}
$$

For the first term, we observe that:

$$\left| \min_{\mathbf{x} \in \mathcal{X}} f(\mathbf{x}) - \min_{\mathbf{x} \in \mathcal{X}} \widehat{f}(\mathbf{x}) \right| \leqslant \max_{\mathbf{x} \in \mathcal{X}} |f(\mathbf{x}) - \widehat{f}(\mathbf{x})|. \tag{97}$$

Let $\mathbf{x}^* \in \operatorname{argmax}_{\mathbf{x} \in \mathcal{X}} f(\mathbf{x})$ and $\widehat{\mathbf{x}}^* \in \operatorname{argmin}_{\mathbf{x} \in \widehat{\mathcal{X}}} \widehat{f}(\mathbf{x})$. For the second term, instead, suppose that the content of the absolute value is non-negative and take $\mathbf{x}' \in \mathcal{X}$:

$$\left| \min_{\mathbf{x} \in \mathcal{X}} \widehat{f}(\mathbf{x}) - \min_{\mathbf{x} \in \widehat{\mathcal{X}}} \widehat{f}(\mathbf{x}) \right| = \min_{\mathbf{x} \in \mathcal{X}} \widehat{f}(\mathbf{x}) - \min_{\mathbf{x} \in \widehat{\mathcal{X}}} \widehat{f}(\mathbf{x}) \tag{98}$$

$$\leqslant \widehat{f}(\mathbf{x}') - \widehat{f}(\widehat{\mathbf{x}}^*) \tag{99}$$

$$\leqslant L d(\mathbf{x}', \widehat{\mathbf{x}}^*) \tag{100}$$

$$\leqslant L \sup_{\widehat{\mathbf{x}}' \in \widehat{\mathcal{X}}} \inf_{\mathbf{x}' \in \mathcal{X}} d(\mathbf{x}', \widehat{\mathbf{x}}'). \tag{101}$$

By performing analogous derivation in the case in which the content of the absolute value is negative, we get:

$$\left| \min_{\mathbf{x} \in \mathcal{X}} \widehat{f}(\mathbf{x}) - \min_{\mathbf{x} \in \widehat{\mathcal{X}}} \widehat{f}(\mathbf{x}) \right| = \min_{\mathbf{x} \in \widehat{\mathcal{X}}} \widehat{f}(\mathbf{x}) - \min_{\mathbf{x} \in \mathcal{X}} \widehat{f}(\mathbf{x}) \leqslant L \sup_{\mathbf{x}' \in \mathcal{X}} \inf_{\widehat{\mathbf{x}}' \in \widehat{\mathcal{X}}} d(\mathbf{x}', \widehat{\mathbf{x}}'). \tag{102}$$

Combining the two cases with the max, the Hausdorff distance as per Definition D.1 appears. If we repeat the derivation from the beginning, summing and subtracting the other mixed term, we obtain an analogous result where the first term is replaced with $\max_{\mathbf{x} \in \widehat{\mathcal{X}}} |f(\mathbf{x}) - \widehat{f}(\mathbf{x})|$. $\qquad \square$

### D.3. Error Propagation in the Projection Step

In this appendix, we provide the error propagation analysis of the projection step.

**Theorem 5.2.** *Let $\overline{\eta} \in \Delta(\mathcal{S})$ be a target distribution and let $\mathbf{P}, \widehat{\mathbf{P}}$ be transition models of ergodic (Assumption 2.1) MDPs. Let $\widetilde{\eta}, \widehat{\widetilde{\eta}}$ be the solutions of the projection problems PROJ($\overline{\eta}$,$\mathbf{P}$) and PROJ($\overline{\eta}$,$\widehat{\mathbf{P}}$), respectively. Then, under Assumption 5.2, it holds that:*

$$\left| \mathcal{D}(\overline{\eta} \| \widetilde{\eta}) - \mathcal{D}(\overline{\eta} \| \widehat{\widetilde{\eta}}) \right| \leqslant L \sqrt{|\mathcal{S}|} \sup_{\pi \in \Pi^{\mathrm{MR}}} \| \mathbf{Z}^{\pi} \|_2 \left\| \mathbf{P} - \widehat{\mathbf{P}} \right\|_2 ,$$

*where $\mathbf{Z}^{\pi} = (\mathbf{I} - \mathbf{P}^{\pi} + \mathbf{1}_{|\mathcal{S}|}(\eta^{\pi})^{\top})^{-1}$ is the* fundamental matrix *(Seneta, 1988) of the MC induced by policy $\pi$.*

*Proof.* We apply Lemma D.1, observing that the objective functions do not change and, thus, the Hausdorff distance term only appears. In particular, we are interested in bounding the Hausdorff distance between the sets $\{\mathbf{K}\mathbf{x} : \mathbf{x} \in \mathcal{X}_{\mathbf{P}}\}$ and $\{\mathbf{K}\mathbf{x} : \mathbf{x} \in \mathcal{X}_{\widehat{\mathbf{P}}}\}$ that correspond to the sets $\mathcal{N}_{\mathbf{P}}$ and $\mathcal{N}_{\widehat{\mathbf{P}}}$, respectively. Let us consider the first side of the Hausdorff distance:

$$\sup_{\eta \in \mathcal{N}_{\mathbf{P}}} \inf_{\widehat{\eta} \in \mathcal{N}_{\widehat{\mathbf{P}}}} \| \eta - \widehat{\eta} \|_1 = \sup_{\pi \in \Pi^{\mathrm{MR}}} \inf_{\widehat{\pi} \in \Pi^{\mathrm{MR}}} \| \eta^{\pi} - \widehat{\eta}^{\widehat{\pi}} \|_1 \leqslant \sup_{\pi \in \Pi^{\mathrm{MR}}} \| \eta^{\pi} - \widehat{\eta}^{\widehat{\pi}} \|_1 \tag{103}$$

$$\leqslant \sqrt{|\mathcal{S}|} \sup_{\pi \in \Pi^{\mathrm{MR}}} \| \eta^{\pi} - \widehat{\eta}^{\widehat{\pi}} \|_2 \leqslant \sqrt{|\mathcal{S}|} \sup_{\pi \in \Pi^{\mathrm{MR}}} \| \mathbf{Z}^{\pi} \|_2 \left\| \mathbf{P} - \widehat{\mathbf{P}} \right\|_2 , \tag{104}$$

having applied Lemma E.5. Note that the other side of the Hausdorff distance behaves the same. Recalling that the divergence is $L$-Lipschitz continuous in its second argument, we obtain the result. $\qquad \square$

### D.4. Error Propagation in the FMSS Control Problem

In this appendix, we provide the error propagation analysis of the FMSS control problem.

**Theorem 5.4.** *Let $\overline{\eta} \in \Delta(\mathcal{S})$ be a target distribution and let $\mathbf{P}, \widehat{\mathbf{P}}$ be transition models of ergodic (Assumption 2.1) MDPs. Let $\widetilde{\eta}, \widehat{\widetilde{\eta}}$ be the solutions of the projection problems PROJ($\overline{\eta}$,$\mathbf{P}$) and PROJ($\overline{\eta}$,$\widehat{\mathbf{P}}$), respectively. Let $\pi^*, \widehat{\pi}^*$ be solutions of*

*FMSS($\widetilde{\eta}$,$\mathbf{P}$) and FMSS($\widehat{\widetilde{\eta}}$,$\widehat{\mathbf{P}}$) problems, respectively. Let $\widetilde{\eta}_{\min} = \min_{s \in \mathcal{S}} \widetilde{\eta}(s) > 0$ and $\widehat{\widetilde{\eta}}_{\min} = \min_{s \in \mathcal{S}} \widehat{\widetilde{\eta}}(s) > 0$. Then, it holds that:*

$$
\left| \sigma_2(\mathbf{P}^{\boldsymbol{\pi}^*}) - \sigma_2(\mathbf{P}^{\widehat{\boldsymbol{\pi}}^*}) \right| \leqslant 5 \| \mathbf{Z}^{\widehat{\boldsymbol{\pi}}^*} \|_2 \sqrt{|\mathcal{S}|} \Big( (\eta_{\min}^{\widehat{\boldsymbol{\pi}}^*})^{-1/2}
$$
$$
+ \sqrt{|\mathcal{A}|} \widetilde{\eta}_{\min}^{-3/2} \widehat{\widetilde{\eta}}_{\min}^{-3/2} (1+\widetilde{H}) \Big) \left( 2 \sqrt{\left\| \widehat{\widetilde{\eta}} - \widetilde{\eta} \right\|_2} + \left\| \mathbf{P} - \widehat{\mathbf{P}} \right\|_2 \right),
$$

*where $\widetilde{H} > 0$ is a finite constant depending on $|\mathcal{S}|\,|\mathcal{A}|$ and on $\widehat{\mathbf{P}}$ and $\mathbf{P}$ only.*

*Proof.* Let us consider the following derivation:

$$
\left\| \widetilde{\mathbf{A}}_{\widetilde{\eta}}^{\boldsymbol{\pi}^*} \right\|_2 - \left\| \widetilde{\mathbf{A}}_{\boldsymbol{\eta}^{\widehat{\pi}^*}}^{\widehat{\boldsymbol{\pi}}^*} \right\|_2 \leqslant \underbrace{\left| \left\| \widetilde{\mathbf{A}}_{\widetilde{\eta}}^{\boldsymbol{\pi}^*} \right\|_2 - \left\| \widehat{\widetilde{\mathbf{A}}}_{\widehat{\widetilde{\eta}}}^{\widehat{\boldsymbol{\pi}}^*} \right\|_2 \right|}_{(A)} + \underbrace{\left| \left\| \widehat{\widetilde{\mathbf{A}}}_{\widehat{\widetilde{\eta}}}^{\widehat{\boldsymbol{\pi}}^*} \right\|_2 - \left\| \widetilde{\mathbf{A}}_{\widehat{\widetilde{\eta}}}^{\widehat{\boldsymbol{\pi}}^*} \right\|_2 \right|}_{(B)} + \underbrace{\left| \left\| \widetilde{\mathbf{A}}_{\widehat{\widetilde{\eta}}}^{\widehat{\boldsymbol{\pi}}^*} \right\|_2 - \left\| \widetilde{\mathbf{A}}_{\boldsymbol{\eta}^{\widehat{\pi}^*}}^{\widehat{\boldsymbol{\pi}}^*} \right\|_2 \right|}_{(C)}. \tag{105}
$$

Let us start with term (B):

$$
\left| \left\| \widehat{\widetilde{\mathbf{A}}}_{\widehat{\widetilde{\eta}}}^{\widehat{\boldsymbol{\pi}}^*} \right\|_2 - \left\| \widetilde{\mathbf{A}}_{\widehat{\widetilde{\eta}}}^{\widehat{\boldsymbol{\pi}}^*} \right\|_2 \right| \leqslant \left\| \widehat{\widetilde{\mathbf{A}}}_{\widehat{\widetilde{\eta}}}^{\widehat{\boldsymbol{\pi}}^*} - \widetilde{\mathbf{A}}_{\widehat{\widetilde{\eta}}}^{\widehat{\boldsymbol{\pi}}^*} \right\|_2 \tag{106}
$$

$$
= \left\| \mathbf{D}_{\widehat{\widetilde{\eta}}}^{1/2} \widehat{\boldsymbol{\Pi}}^* \widehat{\mathbf{P}} \mathbf{D}_{\widehat{\widetilde{\eta}}}^{-1/2} - \sqrt{\widehat{\widetilde{\eta}}} \sqrt{\widehat{\widetilde{\eta}}}^{\top} - \mathbf{D}_{\widehat{\widetilde{\eta}}}^{1/2} \widehat{\boldsymbol{\Pi}}^* \mathbf{P} \mathbf{D}_{\widehat{\widetilde{\eta}}}^{-1/2} + \sqrt{\widehat{\widetilde{\eta}}} \sqrt{\widehat{\widetilde{\eta}}}^{\top} \right\|_2 \tag{107}
$$

$$
\leqslant \left\| \mathbf{D}_{\widehat{\widetilde{\eta}}}^{1/2} \right\|_2 \left\| \widehat{\boldsymbol{\Pi}}^* \right\|_2 \left\| \mathbf{D}_{\widehat{\widetilde{\eta}}}^{-1/2} \right\|_2 \left\| \widehat{\mathbf{P}} - \mathbf{P} \right\|_2 \tag{108}
$$

$$
\leqslant \widehat{\widetilde{\eta}}_{\min}^{-1/2} \left\| \widehat{\mathbf{P}} - \mathbf{P} \right\|_2, \tag{109}
$$

where we bounded $\left\| \mathbf{D}_{\widehat{\widetilde{\eta}}}^{1/2} \right\|_2 \leqslant 1$, $\left\| \widehat{\boldsymbol{\Pi}}^* \right\|_2 \leqslant \max_{s \in \mathcal{S}} \| \widehat{\boldsymbol{\pi}}^*(\cdot|s) \|_2 \leqslant 1$, $\left\| \mathbf{D}_{\widehat{\widetilde{\eta}}}^{-1/2} \right\|_2 \leqslant \widehat{\widetilde{\eta}}_{\min}^{-1/2}$. We now move to term (C):

$$
\left| \left\| \widetilde{\mathbf{A}}_{\widehat{\widetilde{\eta}}}^{\widehat{\boldsymbol{\pi}}^*} \right\|_2 - \left\| \widetilde{\mathbf{A}}_{\boldsymbol{\eta}^{\widehat{\pi}^*}}^{\widehat{\boldsymbol{\pi}}^*} \right\|_2 \right| \leqslant \left\| \widetilde{\mathbf{A}}_{\widehat{\widetilde{\eta}}}^{\widehat{\boldsymbol{\pi}}^*} - \widetilde{\mathbf{A}}_{\boldsymbol{\eta}^{\widehat{\pi}^*}}^{\widehat{\boldsymbol{\pi}}^*} \right\|_2 \tag{110}
$$

$$
= \left\| \mathbf{D}_{\widehat{\widetilde{\eta}}}^{1/2} \widehat{\boldsymbol{\Pi}}^* \mathbf{P} \mathbf{D}_{\widehat{\widetilde{\eta}}}^{-1/2} - \sqrt{\widehat{\widetilde{\eta}}} \sqrt{\widehat{\widetilde{\eta}}}^{\top} - \mathbf{D}_{\boldsymbol{\eta}^{\widehat{\pi}^*}}^{1/2} \widehat{\boldsymbol{\Pi}}^* \mathbf{P} \mathbf{D}_{\boldsymbol{\eta}^{\widehat{\pi}^*}}^{-1/2} + \sqrt{\boldsymbol{\eta}^{\widehat{\pi}^*}} \sqrt{\boldsymbol{\eta}^{\widehat{\pi}^*}}^{\top} \right\|_2 \tag{111}
$$

$$
\leqslant \left\| \mathbf{D}_{\widehat{\widetilde{\eta}}}^{1/2} \widehat{\boldsymbol{\Pi}}^* \mathbf{P} \mathbf{D}_{\widehat{\widetilde{\eta}}}^{-1/2} - \mathbf{D}_{\boldsymbol{\eta}^{\widehat{\pi}^*}}^{1/2} \widehat{\boldsymbol{\Pi}}^* \mathbf{P} \mathbf{D}_{\boldsymbol{\eta}^{\widehat{\pi}^*}}^{-1/2} \right\|_2 + \left\| \sqrt{\widehat{\widetilde{\eta}}} \sqrt{\widehat{\widetilde{\eta}}}^{\top} - \sqrt{\boldsymbol{\eta}^{\widehat{\pi}^*}} \sqrt{\boldsymbol{\eta}^{\widehat{\pi}^*}}^{\top} \right\|_2 \tag{112}
$$

$$
\leqslant \left\| \mathbf{D}_{\widehat{\widetilde{\eta}}}^{1/2} - \mathbf{D}_{\boldsymbol{\eta}^{\widehat{\pi}^*}}^{1/2} \right\|_2 \left\| \widehat{\boldsymbol{\Pi}}^* \mathbf{P} \right\|_2 \left\| \mathbf{D}_{\widehat{\widetilde{\eta}}}^{-1/2} \right\|_2 + \left\| \mathbf{D}_{\boldsymbol{\eta}^{\widehat{\pi}^*}}^{1/2} \right\|_2 \left\| \widehat{\boldsymbol{\Pi}}^* \mathbf{P} \right\|_2 \left\| \mathbf{D}_{\boldsymbol{\eta}^{\widehat{\pi}^*}}^{-1/2} - \mathbf{D}_{\widehat{\widetilde{\eta}}}^{-1/2} \right\|_2 \tag{113}
$$

$$
+ \left\| \sqrt{\widehat{\widetilde{\eta}}} \sqrt{\widehat{\widetilde{\eta}}}^{\top} - \sqrt{\boldsymbol{\eta}^{\widehat{\pi}^*}} \sqrt{\boldsymbol{\eta}^{\widehat{\pi}^*}}^{\top} \right\|_2 \tag{114}
$$

$$
\leqslant 2\sqrt{|\mathcal{S}|} \sqrt{\left\| \widehat{\widetilde{\eta}} - \boldsymbol{\eta}^{\widehat{\pi}^*} \right\|_2} \left( \widehat{\widetilde{\eta}}_{\min}^{-1/2} + (\eta_{\min}^{\widehat{\boldsymbol{\pi}}^*})^{-1/2} + 1 \right), \tag{115}
$$

where we used Lemma E.6 and bounded $\left\| \widehat{\boldsymbol{\Pi}}^* \mathbf{P} \right\|_2 \leqslant \left\| \widehat{\boldsymbol{\Pi}}^* \mathbf{P} \right\|_F \leqslant \sqrt{|\mathcal{S}|}$, $\left\| \mathbf{D}_{\widehat{\widetilde{\eta}}}^{-1/2} \right\|_2 \leqslant \widehat{\widetilde{\eta}}_{\min}^{-1/2}$, $\left\| \mathbf{D}_{\boldsymbol{\eta}^{\widehat{\pi}^*}}^{-1/2} \right\|_2 \leqslant (\eta_{\min}^{\widehat{\boldsymbol{\pi}}^*})^{-1/2}$. Then, by Lemma E.5, recalling that $\widehat{\widetilde{\eta}} = \widehat{\eta}^{\widehat{\pi}^*}$, we have:

$$
\left\| \widehat{\widetilde{\eta}} - \boldsymbol{\eta}^{\widehat{\pi}^*} \right\|_2 \leqslant \| \mathbf{Z}^{\widehat{\boldsymbol{\pi}}^*} \|_2 \left\| \widehat{\mathbf{P}} - \mathbf{P} \right\|_2. \tag{116}
$$

We defer the analysis of term (A) to Lemma D.2. Putting all together and doing some bounds, we get the result. $\qquad \square$

**Lemma D.2.** *Let $\overline{\eta} \in \Delta(\mathcal{S})$ be a target distribution and let $\mathbf{P}, \widehat{\mathbf{P}}$ be transition models of ergodic (Assumption 2.1) MDPs. Let $\widetilde{\eta}, \widehat{\widetilde{\eta}}$ be the solutions of the projection problems PROJ($\overline{\eta}$,$\mathbf{P}$) and PROJ($\overline{\eta}$,$\widehat{\mathbf{P}}$), respectively. Let $\boldsymbol{\pi}^*, \widehat{\boldsymbol{\pi}}^*$ be solutions of*

*FMSS($\widetilde{\boldsymbol{\eta}}$,$\mathbf{P}$) and FMSS($\widehat{\widetilde{\boldsymbol{\eta}}}$,$\widehat{\mathbf{P}}$) problems, respectively. Let $\widetilde{\eta}_{\min} = \min_{s \in \mathcal{S}} \widetilde{\eta}(s) > 0$ and $\widehat{\widetilde{\eta}}_{\min} = \min_{s \in \mathcal{S}} \widehat{\widetilde{\eta}}(s) > 0$. Then, it holds that:*

$$\left| \left\| \widetilde{\mathbf{A}}_{\widetilde{\boldsymbol{\eta}}}^{\boldsymbol{\pi}^*} \right\|_2 - \left\| \widehat{\widetilde{\mathbf{A}}}_{\widehat{\widetilde{\boldsymbol{\eta}}}}^{\widehat{\boldsymbol{\pi}}^*} \right\|_2 \right| \leq 4\sqrt{|\mathcal{S}| \, |\mathcal{A}|} \widetilde{\eta}_{\min}^{-3/2} \widehat{\widetilde{\eta}}_{\min}^{-3/2} (1 + \widetilde{H}(\mathbf{P}, \widehat{\mathbf{P}}, |\mathcal{S}||\mathcal{A}|)) \left( 2\sqrt{\|\widehat{\widetilde{\boldsymbol{\eta}}} - \widetilde{\boldsymbol{\eta}}\|_2} + \left\| \mathbf{P} - \widehat{\mathbf{P}} \right\|_2 \right). \tag{117}$$

*Proof.* We apply Lemma D.1. First, we derive the Lipschitz constant for the objective functions. Thus, for $\boldsymbol{\pi}, \boldsymbol{\pi}' \in \Pi^{\mathrm{MR}}$:

$$\left| \left\| \widetilde{\mathbf{A}}_{\widetilde{\boldsymbol{\eta}}}^{\boldsymbol{\pi}} \right\|_2 - \left\| \widetilde{\mathbf{A}}_{\widetilde{\boldsymbol{\eta}}}^{\boldsymbol{\pi}'} \right\|_2 \right| \leq \left\| \widetilde{\mathbf{A}}_{\widetilde{\boldsymbol{\eta}}}^{\boldsymbol{\pi}} - \widetilde{\mathbf{A}}_{\widetilde{\boldsymbol{\eta}}}^{\boldsymbol{\pi}'} \right\|_2 \leq \sqrt{|\mathcal{S}| \, |\mathcal{A}|} \widetilde{\eta}_{\min}^{-1/2} \left\| \boldsymbol{\Pi} - \boldsymbol{\Pi}' \right\|_2. \tag{118}$$

A similar bound where $\widetilde{\eta}_{\min}$ is replaced with $\widehat{\widetilde{\eta}}_{\min}$ holds for the perturbed problem. Thus, we set:

$$L = \sqrt{|\mathcal{S}| \, |\mathcal{A}|} \min\{\widetilde{\eta}_{\min}, \widehat{\widetilde{\eta}}_{\min}\}^{-1/2}. \tag{119}$$

Then, we compute the difference of the objective functions, for a generic $\boldsymbol{\pi} \in \Pi^{\mathrm{MR}}$:

$$\left| \left\| \widehat{\widetilde{\mathbf{A}}}_{\widehat{\widetilde{\boldsymbol{\eta}}}}^{\boldsymbol{\pi}} \right\|_2 - \left\| \widetilde{\mathbf{A}}_{\widetilde{\boldsymbol{\eta}}}^{\boldsymbol{\pi}} \right\|_2 \right| \leq \left\| \widehat{\widetilde{\mathbf{A}}}_{\widehat{\widetilde{\boldsymbol{\eta}}}}^{\boldsymbol{\pi}} - \widetilde{\mathbf{A}}_{\widetilde{\boldsymbol{\eta}}}^{\boldsymbol{\pi}} \right\|_2 \tag{120}$$

$$= \left\| \mathbf{D}_{\widehat{\widetilde{\boldsymbol{\eta}}}}^{1/2} \boldsymbol{\Pi} \widehat{\mathbf{P}} \mathbf{D}_{\widehat{\widetilde{\boldsymbol{\eta}}}}^{-1/2} - \sqrt{\widehat{\widetilde{\boldsymbol{\eta}}}} \sqrt{\widehat{\widetilde{\boldsymbol{\eta}}}}^{\top} - \mathbf{D}_{\widetilde{\boldsymbol{\eta}}}^{1/2} \boldsymbol{\Pi} \mathbf{P} \mathbf{D}_{\widetilde{\boldsymbol{\eta}}}^{-1/2} + \sqrt{\widetilde{\boldsymbol{\eta}}} \sqrt{\widetilde{\boldsymbol{\eta}}}^{\top} \right\|_2 \tag{121}$$

$$\leq \left\| \mathbf{D}_{\widehat{\widetilde{\boldsymbol{\eta}}}}^{1/2} - \mathbf{D}_{\widetilde{\boldsymbol{\eta}}}^{1/2} \right\|_2 \left\| \boldsymbol{\Pi} \widehat{\mathbf{P}} \right\|_2 \left\| \mathbf{D}_{\widehat{\widetilde{\boldsymbol{\eta}}}}^{-1/2} \right\|_2 + \tag{122}$$

$$+ \left\| \mathbf{D}_{\widetilde{\boldsymbol{\eta}}}^{1/2} \right\|_2 \| \boldsymbol{\Pi} \|_2 \left\| \mathbf{D}_{\widehat{\widetilde{\boldsymbol{\eta}}}}^{-1/2} \right\|_2 \left\| \widehat{\mathbf{P}} - \mathbf{P} \right\|_2 \tag{123}$$

$$+ \left\| \mathbf{D}_{\widetilde{\boldsymbol{\eta}}}^{1/2} \right\|_2 \| \boldsymbol{\Pi} \mathbf{P} \|_2 \left\| \mathbf{D}_{\widetilde{\boldsymbol{\eta}}}^{-1/2} - \mathbf{D}_{\widehat{\widetilde{\boldsymbol{\eta}}}}^{-1/2} \right\|_2 \tag{124}$$

$$+ \left\| \sqrt{\widehat{\widetilde{\boldsymbol{\eta}}}} \sqrt{\widehat{\widetilde{\boldsymbol{\eta}}}}^{\top} - \sqrt{\widetilde{\boldsymbol{\eta}}} \sqrt{\widetilde{\boldsymbol{\eta}}}^{\top} \right\|_2 \tag{125}$$

$$\leq \widehat{\widetilde{\eta}}_{\min}^{-1/2} \sqrt{|\mathcal{S}|} \sqrt{\|\widehat{\widetilde{\boldsymbol{\eta}}} - \widetilde{\boldsymbol{\eta}}\|_2} + \widehat{\widetilde{\eta}}_{\min}^{-1/2} \left\| \widehat{\mathbf{P}} - \mathbf{P} \right\|_2 + \widehat{\widetilde{\eta}}_{\min}^{-1/2} \widetilde{\eta}_{\min}^{-1/2} \sqrt{|\mathcal{S}|} \sqrt{\|\widehat{\widetilde{\boldsymbol{\eta}}} - \widetilde{\boldsymbol{\eta}}\|_2} \tag{126}$$

$$+ 2\sqrt{|\mathcal{S}|} \sqrt{\|\widehat{\widetilde{\boldsymbol{\eta}}} - \widetilde{\boldsymbol{\eta}}\|_2} \tag{127}$$

$$\leq \widehat{\widetilde{\eta}}_{\min}^{-1/2} \left\| \widehat{\mathbf{P}} - \mathbf{P} \right\|_2 + 3\widehat{\widetilde{\eta}}_{\min}^{-1/2} \widetilde{\eta}_{\min}^{-1/2} \sqrt{|\mathcal{S}|} \sqrt{\|\widehat{\widetilde{\boldsymbol{\eta}}} - \widetilde{\boldsymbol{\eta}}\|_2}. \tag{128}$$

Now, we have to compute the Hausdorff distance in the space of policies. To this end, we recall that we can always relate the policy to the state-action equilibrium distribution. Let $\mathbf{x} \in \Delta(\mathcal{S} \times \mathcal{A})$ such that:

$$\begin{pmatrix} \mathbf{K} \\ \mathbf{P}^{\top} \end{pmatrix} \mathbf{x} = \begin{pmatrix} \widetilde{\boldsymbol{\eta}} \\ \widetilde{\boldsymbol{\eta}} \end{pmatrix}, \tag{129}$$

which defines the set $\mathcal{X}_{\widetilde{\boldsymbol{\eta}}}$ of state-action equilibrium distributions inducing $\widetilde{\boldsymbol{\eta}}$ as marginal over the states. Thus, the set of policies fulfilling constraint $\widetilde{\boldsymbol{\eta}} = (\mathbf{P}^{\boldsymbol{\pi}})^{\top} \widetilde{\boldsymbol{\eta}}$ can be indeed expressed as:

$$\Pi_{\widetilde{\boldsymbol{\eta}}} = \left\{ \boldsymbol{\pi} \in \Pi^{\mathrm{MR}} \mid \pi(a|s) = \frac{x(s,a)}{\widetilde{\eta}(s)} \text{ for all } (s,a) \in \mathcal{S} \times \mathcal{A} \text{ for some } \mathbf{x} \in \mathcal{X}_{\widetilde{\boldsymbol{\eta}}} \right\}. \tag{130}$$

In a similar way, we can define the set $\widehat{\Pi}_{\widehat{\widetilde{\boldsymbol{\eta}}}}$ of policies fulfilling constraint $\widehat{\widetilde{\boldsymbol{\eta}}} = (\widehat{\mathbf{P}}^{\boldsymbol{\pi}})^{\top} \widehat{\widetilde{\boldsymbol{\eta}}}$, defined in terms of the set of state-action equilibrium distributions $\widehat{\mathcal{X}}_{\widehat{\widetilde{\boldsymbol{\eta}}}}$ in which we replace $\mathbf{P}$ with $\widehat{\mathbf{P}}$ and $\widetilde{\boldsymbol{\eta}}$ with $\widehat{\widetilde{\boldsymbol{\eta}}}$. Thus, we can bound one side of the

Hausdorff as follows:

$$\sup_{\boldsymbol{\pi} \in \Pi_{\widetilde{\eta}}} \inf_{\widehat{\boldsymbol{\pi}} \in \widehat{\Pi}_{\widehat{\eta}}} \|\boldsymbol{\Pi} - \widehat{\boldsymbol{\Pi}}\|_2 = \sup_{\boldsymbol{\pi} \in \Pi_{\widetilde{\eta}}} \inf_{\widehat{\boldsymbol{\pi}} \in \widehat{\Pi}_{\widehat{\eta}}} \max_{s \in \mathcal{S}} \|\boldsymbol{\pi}(\cdot|s) - \widehat{\boldsymbol{\pi}}(\cdot|s)\|_2 \tag{131}$$

$$= \sup_{\mathbf{x} \in \mathcal{X}_{\widetilde{\eta}}} \inf_{\widehat{\mathbf{x}} \in \widehat{\mathcal{X}}_{\widehat{\eta}}} \max_{s \in \mathcal{S}} \left\| \frac{\mathbf{x}(s, \cdot)}{\widetilde{\eta}(s)} - \frac{\widehat{\mathbf{x}}(s, \cdot)}{\widehat{\widetilde{\eta}}(s)} \right\|_2 \tag{132}$$

$$\leqslant \widetilde{\eta}_{\min}^{-1} \widehat{\widetilde{\eta}}_{\min}^{-1} \left( \sup_{\mathbf{x} \in \mathcal{X}_{\widetilde{\eta}}} \inf_{\widehat{\mathbf{x}} \in \widehat{\mathcal{X}}_{\widehat{\eta}}} \max_{s \in \mathcal{S}} \|\mathbf{x}(s, \cdot) - \widehat{\mathbf{x}}(s, \cdot)\|_2 + \|\widetilde{\boldsymbol{\eta}} - \widehat{\widetilde{\boldsymbol{\eta}}}\|_2 \right) \tag{133}$$

$$\leqslant \widetilde{\eta}_{\min}^{-1} \widehat{\widetilde{\eta}}_{\min}^{-1} \left( \sup_{\mathbf{x} \in \mathcal{X}_{\widetilde{\eta}}} \inf_{\widehat{\mathbf{x}} \in \widehat{\mathcal{X}}_{\widehat{\eta}}} \|\mathbf{x} - \widehat{\mathbf{x}}\|_2 + \|\widetilde{\boldsymbol{\eta}} - \widehat{\widetilde{\boldsymbol{\eta}}}\|_2 \right). \tag{134}$$

Thus, we have reduced the problem to the computation of the Hausdorff distance between the sets $\mathcal{X}_{\widetilde{\eta}}$ and $\widehat{\mathcal{X}}_{\widehat{\eta}}$. To this end, we employ an approach based on Hoffman's perturbation bounds (Hoffman, 1952). In particular, we consider the setting of Equation (17) of (Pena et al., 2021). Let $\mathcal{R}$ be a reference polyhedral and consider the set $\mathcal{C}$ of points $\mathbf{x} \in \mathcal{R}$ such that $\mathbf{C}\mathbf{x} = \mathbf{b}$. Proposition 5 of (Pena et al., 2021) states that there exists a finite constant $H(\mathbf{C}, \mathcal{R})$ such that for every $\mathbf{y} \in \mathcal{R}$ it holds that:

$$\inf_{\mathbf{x} \in \mathcal{C}} \|\mathbf{y} - \mathbf{x}\|_2 \leqslant H(\mathbf{C}, \mathcal{R}) \|\mathbf{C}\mathbf{y} - \mathbf{b}\|_2. \tag{135}$$

Notice that this result allows bounding the Hausdorff distance by simply taking the supremum over $\mathbf{y}$ in a suitable subset of $\mathcal{R}$. We particularize the result for our case. We take $\mathcal{R} = \Delta(\mathcal{S} \times \mathcal{A})$, $\mathbf{C} = \begin{pmatrix} \mathbf{K} \\ \mathbf{P}^\top \end{pmatrix}$ and $\mathbf{b} = \begin{pmatrix} \widetilde{\boldsymbol{\eta}} \\ \widetilde{\boldsymbol{\eta}} \end{pmatrix}$. Now take $\mathbf{y} = \widehat{\mathbf{x}} \in \widehat{\mathcal{X}}_{\widehat{\eta}}$. We have:

$$\|\mathbf{A}\mathbf{y} - \mathbf{b}\|_2 = \left\| \begin{pmatrix} \mathbf{K} \\ \mathbf{P}^\top \end{pmatrix} \widehat{\mathbf{x}} - \begin{pmatrix} \widetilde{\boldsymbol{\eta}} \\ \widetilde{\boldsymbol{\eta}} \end{pmatrix} \right\|_2 = \left\| \begin{pmatrix} \widehat{\widetilde{\boldsymbol{\eta}}} - \widetilde{\boldsymbol{\eta}} \\ \widehat{\widetilde{\boldsymbol{\eta}}} - \widetilde{\boldsymbol{\eta}} + (\mathbf{P}^\top - \widehat{\mathbf{P}}^\top)\widehat{\mathbf{x}} \end{pmatrix} \right\|_2 \leqslant \sqrt{2}\|\widehat{\widetilde{\boldsymbol{\eta}}} - \widetilde{\boldsymbol{\eta}}\|_2 + \left\| \mathbf{P} - \widehat{\mathbf{P}} \right\|_2, \tag{136}$$

having exploited the definition of Euclidean norm and the sub-additivity of the square root. This allows us to compute the first side of the Hausdorff distance:

$$\sup_{\boldsymbol{\pi} \in \Pi_{\widetilde{\eta}}} \inf_{\widehat{\boldsymbol{\pi}} \in \widehat{\Pi}_{\widehat{\eta}}} \|\boldsymbol{\Pi} - \widehat{\boldsymbol{\Pi}}\|_2 \leqslant \widetilde{\eta}_{\min}^{-1} \widehat{\widetilde{\eta}}_{\min}^{-1} \left( (1 + \sqrt{2}\widetilde{H}(\mathbf{P}, |\mathcal{S}||\mathcal{A}|))\|\widehat{\widetilde{\boldsymbol{\eta}}} - \widetilde{\boldsymbol{\eta}}\|_2 + \widetilde{H}(\mathbf{P}, |\mathcal{S}||\mathcal{A}|) \left\| \mathbf{P} - \widehat{\mathbf{P}} \right\|_2 \right), \tag{137}$$

having noted that the constant $\widetilde{H}$ depends on $\mathbf{P}$ and $|\mathcal{S}||\mathcal{A}|$ only. We can repeat the derivation for the other side of the Hausdorff distance, generating a dependence on $\widehat{\mathbf{P}}$ in the definition of the constant $\widetilde{H}$. Let us define $\widetilde{H}(\mathbf{P}, \widehat{\mathbf{P}}, |\mathcal{S}||\mathcal{A}|) = \max\{\widetilde{H}(\mathbf{P}, |\mathcal{S}||\mathcal{A}|), \widetilde{H}(\widehat{\mathbf{P}}, |\mathcal{S}||\mathcal{A}|)\}$ that we abbreviate with $\widetilde{H}$ in the statement. Combining everything, we get:

$$\left| \left\| \widetilde{\mathbf{A}}_{\widetilde{\eta}}^{\boldsymbol{\pi}*} \right\|_2 - \left\| \widehat{\widetilde{\mathbf{A}}}_{\widehat{\eta}}^{\widehat{\boldsymbol{\pi}}*} \right\|_2 \right| \leqslant \widehat{\widetilde{\eta}}_{\min}^{-1/2} \left\| \widehat{\mathbf{P}} - \mathbf{P} \right\|_2 + 3\widehat{\widetilde{\eta}}_{\min}^{-1/2}\widetilde{\eta}_{\min}^{-1/2}\sqrt{|\mathcal{S}|}\sqrt{\|\widehat{\widetilde{\boldsymbol{\eta}}} - \widetilde{\boldsymbol{\eta}}\|_2} \tag{138}$$

$$+ \sqrt{|\mathcal{S}||\mathcal{A}|} \min\{\widetilde{\eta}_{\min}, \widehat{\widetilde{\eta}}_{\min}\}^{-1/2}\widetilde{\eta}_{\min}^{-1}\widehat{\widetilde{\eta}}_{\min}^{-1} \left( (1 + \sqrt{2}\widetilde{H}(\mathbf{P}, \widehat{\mathbf{P}}, |\mathcal{S}||\mathcal{A}|))\|\widehat{\widetilde{\boldsymbol{\eta}}} - \widetilde{\boldsymbol{\eta}}\|_2 + \widetilde{H}(\mathbf{P}, \widehat{\mathbf{P}}, |\mathcal{S}||\mathcal{A}|) \left\| \mathbf{P} - \widehat{\mathbf{P}} \right\|_2 \right) \tag{139}$$

$$\leqslant 4\sqrt{|\mathcal{S}||\mathcal{A}|}\widetilde{\eta}_{\min}^{-3/2}\widehat{\widetilde{\eta}}_{\min}^{-3/2}(1 + \widetilde{H}(\mathbf{P}, \widehat{\mathbf{P}}, |\mathcal{S}||\mathcal{A}|)) \left( 2\sqrt{\|\widehat{\widetilde{\boldsymbol{\eta}}} - \widetilde{\boldsymbol{\eta}}\|_2} + \left\| \mathbf{P} - \widehat{\mathbf{P}} \right\|_2 \right). \tag{140}$$

$\square$

# E. Auxiliary Lemmas

**Theorem E.1** (Chernoff-Hoeffding Bounds for Ergodic Markov Chains – Chung et al. 2012, Theorem 3.1)**.** *Let $\mathbf{T}$ be an ergodic Markov chain with finite state space $\mathcal{Z}$ and equilibrium distribution $\boldsymbol{\eta}$. Let $T_\zeta$ be its $\zeta$-mixing time for $\zeta \leqslant 1/8$. Let $(Z_1, \ldots, Z_t)$ denote a $t$-step random walk on $\mathbf{T}$ starting from an initial distribution $\boldsymbol{\mu}_0$ on $\mathcal{Z}$, i.e., $Z_1 \sim \boldsymbol{\mu}_0$. For every $i \in [\![t]\!]$, let $f_i : \mathcal{Z} \to [0,1]$ be a weight function at step $i$ such that the expected weight $\mathbb{E}_{\boldsymbol{\eta}}[f_i(Z)] = \mu$ for all $i$. Define the total weight of the walk $(Z_1, \ldots, Z_t)$ by $X := \sum_{i=1}^t f_i(Z_i)$. There exists a universal constant $c > 0$ such that:*

*1.* $\mathbb{P}_{\boldsymbol{\mu}_0}^{\mathbf{T}} (X \geqslant (1+\varepsilon)\mu t) \quad \leqslant \quad \begin{cases} c\|\boldsymbol{\mu}_0\|_{\ell^2(1/\boldsymbol{\eta})} \exp\left(-\varepsilon^2 \mu t/(72 T_\zeta)\right) & \text{for } 0 \leqslant \varepsilon \leqslant 1 \\ c\|\boldsymbol{\mu}_0\|_{\ell^2(1/\boldsymbol{\eta})} \exp\left(-\varepsilon \mu t/(72 T_\zeta)\right) & \text{for } \varepsilon > 1 \end{cases};$

*2.* $\mathbb{P}_{\boldsymbol{\mu}_0}^{\mathbf{T}} (X \leqslant (1-\varepsilon)\mu t) \quad \leqslant \quad c\|\boldsymbol{\mu}_0\|_{\ell^2(1/\boldsymbol{\eta})} \exp\left(-\varepsilon^2 \mu t/(72 T_\zeta)\right) \quad \text{for } 0 \leqslant \varepsilon \leqslant 1.$

**Lemma E.2** (Lazzati et al. 2024, Lemma J.2)**.** *Let $X_1, \ldots, X_n, \ldots$ be i.i.d. samples from a distribution $p \in \Delta([\![m]\!])$. We denote by $\widehat{p}_n$ the empirical vector of probabilities, i.e., for all $k \in [\![m]\!]$:*

$$\widehat{p}_{n,k} = \frac{1}{n} \sum_{l=1}^n \mathbf{1}\{X_l = k\}. \tag{141}$$

*For all $p \in \Delta([\![m]\!])$, for all $\delta \in [0,1]$, it holds that:*

$$\mathbb{P}\left(\exists n \in \mathbb{N}_{\geqslant 0}, n\mathcal{D}_{\mathrm{KL}}(\hat{p}_n \| p) > \ln(1/\delta) + (m-1)\ln(e(1+n/(m-1)))\right) \leqslant \delta. \tag{142}$$

**Lemma E.3** (Lazzati et al. 2024, Lemma J.3)**.** *Let $a,b,c,d > 0$ such that $2bc > e$. Then, the inequality $x \geqslant a + b\ln(cx+d)$ is satisfied by all $x \geqslant 2a + 3b\ln(2bc) + d/c$.*

**Lemma E.4.** *Let $\widehat{\mathbf{P}}, \mathbf{P} \in \mathbb{R}^{|\mathcal{S}||\mathcal{A}| \times |\mathcal{S}|}$ be two transition kernels. Then, it holds that:*

$$\left\|\widehat{\mathbf{P}} - \mathbf{P}\right\|_2 \leqslant 2\sqrt{|\mathcal{S}|\,|\mathcal{A}|} \max_{(s,a) \in \mathcal{S} \times \mathcal{A}} \sqrt{2\mathcal{D}_{\mathrm{KL}}(\hat{\mathbf{p}}(\cdot|s,a)\|\mathbf{p}(\cdot|s,a))} \tag{143}$$

*Proof.*

$$\left\|\widehat{\mathbf{P}} - \mathbf{P}\right\|_2 \leqslant \sqrt{|\mathcal{S}|\,|\mathcal{A}|} \left\|\widehat{\mathbf{P}} - \mathbf{P}\right\|_\infty \tag{144}$$

$$= 2\sqrt{|\mathcal{S}|\,|\mathcal{A}|} \max_{(s,a) \in \mathcal{S} \times \mathcal{A}} \|\hat{\mathbf{p}}(\cdot|s,a) - \mathbf{p}(\cdot|s,a)\|_{\mathrm{TV}} \tag{145}$$

$$\leqslant 2\sqrt{|\mathcal{S}|\,|\mathcal{A}|} \max_{(s,a) \in \mathcal{S} \times \mathcal{A}} \sqrt{2\mathcal{D}_{\mathrm{KL}}(\hat{\mathbf{p}}(\cdot|s,a)\|\mathbf{p}(\cdot|s,a))}, \tag{146}$$

where line (144) derives from standard properties of the matrix norms, while line (146) follows from Pinsker's inequality. $\square$

**Lemma E.5** (Perturbation of the steady-state distribution)**.** *Let $\boldsymbol{\pi} \in \Pi^{\mathrm{MR}}$ be a policy and let $\mathbf{P}, \widehat{\mathbf{P}}$ be ergodic transition models. Let $\boldsymbol{\eta}^{\boldsymbol{\pi}}, \widehat{\boldsymbol{\eta}}^{\boldsymbol{\pi}}$ be the respective stationary distributions. Then, it holds that:*

$$\|\widehat{\boldsymbol{\eta}}^{\boldsymbol{\pi}} - \boldsymbol{\eta}^{\boldsymbol{\pi}}\|_2 \leqslant \|\mathbf{Z}^{\boldsymbol{\pi}}\|_2 \left\|\widehat{\mathbf{P}} - \mathbf{P}\right\|_2, \tag{147}$$

*where $\mathbf{Z}^{\boldsymbol{\pi}} = (\mathbf{I} - \boldsymbol{\Pi}\mathbf{P} + \mathbf{1}(\boldsymbol{\eta}^{\boldsymbol{\pi}})^\top)^{-1}$ is the fundamental matrix of the MC.*

*Proof.* We start from Equation (2) of (Seneta, 1988):

$$(\widehat{\boldsymbol{\eta}}^{\boldsymbol{\pi}})^\top - (\boldsymbol{\eta}^{\boldsymbol{\pi}})^\top = (\widehat{\boldsymbol{\eta}}^{\boldsymbol{\pi}})^\top \boldsymbol{\Pi}(\widehat{\mathbf{P}} - \mathbf{P})\mathbf{Z}^{\boldsymbol{\pi}}, \tag{148}$$

where $\mathbf{Z}^{\boldsymbol{\pi}} = (\mathbf{I} - \mathbf{P}^{\boldsymbol{\pi}} + \mathbf{1}(\boldsymbol{\eta}^{\boldsymbol{\pi}})^\top)^{-1}$ is the fundamental matrix of the MC. By applying the Euclidean norm and observing that $\|(\widehat{\boldsymbol{\eta}}^{\boldsymbol{\pi}})^\top\|_2 \leqslant 1$ and $\|\boldsymbol{\Pi}\|_2 \leqslant 1$, we get the result. $\square$

**Lemma E.6.** *Let $\eta, \widehat{\eta} \in \Delta(\mathcal{S})$ such that $\eta_{\min} = \min_{s \in \mathcal{S}} \eta(s) > 0$ and $\widehat{\eta}_{\min} = \min_{s \in \mathcal{S}} \widehat{\eta}(s) > 0$. The following statements hold:*

- $\|\mathbf{D}_{\eta}^{1/2} - \mathbf{D}_{\widehat{\eta}}^{1/2}\|_2 \leqslant \sqrt{\|\eta - \widehat{\eta}\|_2}$;

- $\|\mathbf{D}_{\eta}^{-1/2} - \mathbf{D}_{\widehat{\eta}}^{-1/2}\|_2 \leqslant \frac{1}{\sqrt{\eta_{\min}\widehat{\eta}_{\min}}}\sqrt{\|\eta - \widehat{\eta}\|_2}$;

- $\|\sqrt{\eta}\sqrt{\eta}^{\top} - \sqrt{\widehat{\eta}}\sqrt{\widehat{\eta}}^{\top}\|_2 \leqslant 2\sqrt{|\mathcal{S}|}\sqrt{\|\eta - \widehat{\eta}\|_2}$.

*Proof.* For the first statement, we have the following chain of inequalities $\|\mathbf{D}_{\eta}^{1/2} - \mathbf{D}_{\widehat{\eta}}^{1/2}\|_2 = \|\sqrt{\eta} - \sqrt{\widehat{\eta}}\|_{\infty} = \max_{s \in \mathcal{S}} |\sqrt{\eta(s)} - \sqrt{\widehat{\eta}(s)}| \leqslant \max_{s \in \mathcal{S}} \sqrt{|\eta(s) - \widehat{\eta}(s)|} = \sqrt{\|\eta - \widehat{\eta}\|_{\infty}} \leqslant \sqrt{\|\eta - \widehat{\eta}\|_2}$, having exploited the subadditivity of the square root. For the second statement, we have:

$$\|\mathbf{D}_{\eta}^{-1/2} - \mathbf{D}_{\widehat{\eta}}^{-1/2}\|_2 = \max_{s \in \mathcal{S}} \left| \frac{1}{\sqrt{\eta(s)}} - \frac{1}{\sqrt{\widehat{\eta}(s)}} \right| \tag{149}$$

$$= \max_{s \in \mathcal{S}} \left| \frac{|\sqrt{\widehat{\eta}(s)} - \sqrt{\eta(s)}|}{\sqrt{\eta(s)}\sqrt{\widehat{\eta}(s)}} \right| \leqslant \frac{1}{\sqrt{\eta_{\min}\widehat{\eta}_{\min}}} \max_{s \in \mathcal{S}} |\sqrt{\eta(s)} - \sqrt{\widehat{\eta}(s)}|. \tag{150}$$

For the last statement, we proceed as follows:

$$\|\sqrt{\eta}\sqrt{\eta}^{\top} - \sqrt{\widehat{\eta}}\sqrt{\widehat{\eta}}^{\top}\|_2 \leqslant \|\sqrt{\eta}(\sqrt{\eta}^{\top} - \sqrt{\widehat{\eta}}^{\top}) + (\sqrt{\eta} - \sqrt{\widehat{\eta}})\sqrt{\widehat{\eta}}^{\top}\|_2 \tag{151}$$

$$\leqslant \|\sqrt{\eta}\|_2 \|\sqrt{\eta} - \sqrt{\widehat{\eta}}\|_2 + \|\sqrt{\eta} - \sqrt{\widehat{\eta}}\|_2 \|\sqrt{\widehat{\eta}}\|_2 \tag{152}$$

$$\leqslant 2\|\sqrt{\eta} - \sqrt{\widehat{\eta}}\|_2 \tag{153}$$

having exploited $\|\sqrt{\eta}\|_2 = \sqrt{\|\sqrt{\eta}\|_1} = 1$ and that:

$$\|\sqrt{\eta} - \sqrt{\widehat{\eta}}\|_2^2 = \sum_{s \in \mathcal{S}} (\sqrt{\eta}(s) - \sqrt{\widehat{\eta}}(s))^2 \leqslant \|\eta - \widehat{\eta}\|_1^2 \leqslant |\mathcal{S}| \, \|\eta - \widehat{\eta}\|_2. \tag{154}$$

from the subadditivity of the square root. $\qquad\square$

**Lemma E.7.** *Let $\pi \in \Pi^{\mathrm{MR}}$ be a policy and $\mathbf{P}$ be an ergodic MDP with $\eta^{\pi}$ the respective stationary distribution. Then, it holds that:*

$$\|\mathbf{Z}^{\pi}\|_2 \leqslant \frac{1}{\sqrt{\eta_{\min}}(1 - \sigma_2(\mathbf{P}^{\pi}))}. \tag{155}$$

*Proof.* We proceed as follows:

$$\mathbf{Z}^{\pi} = (\mathbf{I} - \mathbf{P}^{\pi} + \mathbf{1}(\eta^{\pi})^{\top})^{-1} \tag{156}$$

$$= \mathbf{D}_{\eta^{\pi}}^{-1/2}\mathbf{D}_{\eta^{\pi}}^{1/2}(\mathbf{I} - \mathbf{P}^{\pi} + \mathbf{1}(\eta^{\pi})^{\top})^{-1}\mathbf{D}_{\eta^{\pi}}^{-1/2}\mathbf{D}_{\eta^{\pi}}^{1/2} \tag{157}$$

$$= \mathbf{D}_{\eta^{\pi}}^{-1/2}(\mathbf{I} - \mathbf{D}_{\eta^{\pi}}^{1/2}\mathbf{P}^{\pi}\mathbf{D}_{\eta^{\pi}}^{-1/2} + \sqrt{\eta^{\pi}}\sqrt{\eta^{\pi}}^{\top})^{-1}\mathbf{D}_{\eta^{\pi}}^{1/2}. \tag{158}$$

By taking the norm and applying the Neumann series, we have:

$$\|\mathbf{Z}^{\boldsymbol{\pi}}\|_2 \leqslant \eta_{\min}^{-1/2} \left\| (\mathbf{I} - \mathbf{D}_{\boldsymbol{\eta}^{\boldsymbol{\pi}}}^{1/2} \mathbf{P}^{\boldsymbol{\pi}} \mathbf{D}_{\boldsymbol{\eta}^{\boldsymbol{\pi}}}^{-1/2} + \sqrt{\boldsymbol{\eta}^{\boldsymbol{\pi}}} \sqrt{\boldsymbol{\eta}^{\boldsymbol{\pi}}}^{\top})^{-1} \right\|_2 \tag{159}$$

$$= \eta_{\min}^{-1/2} \left\| \sum_{t=0}^{+\infty} \left( \mathbf{D}_{\boldsymbol{\eta}^{\boldsymbol{\pi}}}^{1/2} \mathbf{P}^{\boldsymbol{\pi}} \mathbf{D}_{\boldsymbol{\eta}^{\boldsymbol{\pi}}}^{-1/2} - \sqrt{\boldsymbol{\eta}^{\boldsymbol{\pi}}} \sqrt{\boldsymbol{\eta}^{\boldsymbol{\pi}}}^{\top} \right)^t \right\|_2 \tag{160}$$

$$\leqslant \eta_{\min}^{-1/2} \sum_{t=0}^{+\infty} \left\| \left( \mathbf{D}_{\boldsymbol{\eta}^{\boldsymbol{\pi}}}^{1/2} \mathbf{P}^{\boldsymbol{\pi}} \mathbf{D}_{\boldsymbol{\eta}^{\boldsymbol{\pi}}}^{-1/2} - \sqrt{\boldsymbol{\eta}^{\boldsymbol{\pi}}} \sqrt{\boldsymbol{\eta}^{\boldsymbol{\pi}}}^{\top} \right)^t \right\|_2 \tag{161}$$

$$\leqslant \eta_{\min}^{-1/2} \sum_{t=0}^{+\infty} \left\| \mathbf{D}_{\boldsymbol{\eta}^{\boldsymbol{\pi}}}^{1/2} \mathbf{P}^{\boldsymbol{\pi}} \mathbf{D}_{\boldsymbol{\eta}^{\boldsymbol{\pi}}}^{-1/2} - \sqrt{\boldsymbol{\eta}^{\boldsymbol{\pi}}} \sqrt{\boldsymbol{\eta}^{\boldsymbol{\pi}}}^{\top} \right\|_2^t \tag{162}$$

$$\leqslant \eta_{\min}^{-1/2} \sum_{t=0}^{+\infty} \sigma_2(\mathbf{P}^{\boldsymbol{\pi}})^t \tag{163}$$

$$= \frac{1}{\sqrt{\eta_{\min}}(1 - \sigma_2(\mathbf{P}^{\boldsymbol{\pi}}))}, \tag{164}$$

having also used the compatibility of the spectral norm. $\qquad\square$

# F. `FMSS-SV` Experimental Evaluation in a Grid-World Environment

In this appendix, we provide a detailed empirical evaluation of the `FMSS-SV` approach (Algorithm 1). We first describe the experimental setup and subsequently analyze the performance of our learning algorithm as a function of the trajectory length $T$ collected under the behavioral policy $\pi^b$. Our analysis focuses on two distinct scenarios: (i) a uniform target distribution across the state space, and (ii) a Dirac-like target distribution where the probability mass is concentrated on a single, randomly selected state. Finally, for the uniform case, we compare the performance of our algorithm against a heuristic baseline, showing that our principled approach can provide an improvement over such heuristics in achieving faster convergence rates.

## F.1. The Grid-World Environment

We consider an $8 \times 8$ grid-world environment with a state space of $|\mathcal{S}| = 64$ and an action space of $|\mathcal{A}| = 4$, consisting of the cardinal moves: `up`, `right`, `down`, and `left`. The task of the agent is to move in this grid-world so as to reach as fast as possible the prescribed target distribution $\overline{\eta} \in \Delta(\mathcal{S})$ (or its feasible approximation $\widetilde{\eta}$, as returned by Program 7). Ensuring ergodicity in transition kernels for environments of this scale can be challenging. To guarantee the satisfaction of Assumption 2.1 we thus define the transition dynamics as follows: for each state $s$ and action $a$, we assign a strictly positive probability to each of the four adjacent states, ensuring that each pair $s_i, s_j$ communicates. To further ensure aperiodicity, we introduce a small probability $\delta > 0$ that the agent remains in the current state (a "self-loop"). For a given action $a \in \mathcal{A}$ corresponding to a target neighbor $s'_{\text{target}} \in \mathcal{S}$ (e.g., the target state $s'_{\text{target}}$ for $a = $ `up` will be the cell above the one currently occupied by the agent), the transition probabilities are then instantiated as follows:

$$p(s'|s,a) = \begin{cases} k(1-\delta) & \text{if } s' = s'_{\text{target}} \\ \frac{(1-k)(1-\delta)}{|\mathcal{A}|-1} & \text{if } s' \in \mathcal{N}(s)\setminus\{s'_{\text{target}}\} \\ \delta & \text{if } s' = s \\ 0 & \text{otherwise} \end{cases} \tag{165}$$

where $\mathcal{N}(s)$ denotes the set of four adjacent neighbors and $k \in [0,1]$ controls how the probability $(1-\delta)$ is distributed across the four available actions. We further assume that for states located at the edges or corners of the grid, any action that would result in a transition outside the environment's boundaries instead causes the agent to remain in its current position. Formally, if a move from state $s$ via action $a$ targets a coordinate $s'_{out} \notin \mathcal{S}$, the transition probability is reassigned to the self-loop: $p(s|s,a) = p(s|s,a) + p(s'_{out}|s,a)$. This ensures that the transition kernel remains row-stochastic. In all the examples that follow, we set $\delta = 10^{-3}$, $k = 0.7$.

In this experimental setting, we solve the projection problem (7) by choosing the divergence $\mathcal{D}(\cdot\|\cdot)$ to be the Kullback-Leibler (KL) divergence, defined as

$$\mathcal{D}_{KL}(\overline{\eta}\|\mathbf{Kx}) = \sum_{s \in \mathcal{S}} \overline{\eta}(s) \log\left(\frac{\overline{\eta}(s)}{(\mathbf{Kx})(s)}\right). \tag{166}$$

Here, $[\mathbf{Kx}](s)$ denotes the state-marginal distribution induced by the state-action occupancy $\mathbf{x}$. When $\overline{\eta}$ has full support, the KL divergence is strictly convex in its second argument over the positive orthant, which yields a unique projected state distribution. For non-full-support targets, such as the Dirac-like targets considered below, this strict-convexity argument does not apply globally; in that case, we select one minimizer of Program (7) and denote its state marginal by $\widetilde{\eta}$.

We conclude the description of our experimental setup by specifying the behavioral policy used for data collection. In all experiments, we employ a uniform behavioral policy, $\pi^b(a|s) = \frac{1}{|\mathcal{A}|}$ for all $s \in \mathcal{S}$ and $a \in \mathcal{A}$. This choice is motivated by our theoretical analysis in Theorem 5.1, as the uniform policy ensures broad coverage of the state-action space, thereby maximizing the coefficient $\xi$ in the sample complexity bound.

**Remark F.1.** *It is worth noting that the inherent symmetry of an environment, such as a stochastic grid-world, is very likely to generate a reversible $\mathbf{P}^\pi$. Furthermore, selecting very small values of $\delta$ can introduce bottlenecks that significantly limit the best possible rate of convergence, as established by conductance analysis (see Montenegro & Tetali, 2006, Chapter 4). The most direct consequence of these observations is that, in the results that follow, the values of the SLEM and the SLSV will be extremely close. Moreover, the margin of improvement between a fast-converging and a slow-converging policy will be limited, yet appreciable provided the threshold is set sufficiently small.*

**Remark F.2.** *All experimental results presented in Appendices F.2 and F.3 are averaged over 6 independent trials, each utilizing a unique random seed to ensure statistical robustness. Shaded regions in the figures represent the corresponding 95% confidence intervals.*

## F.2. Point Mass Target Distribution

In this subsection, we evaluate the performance of FMSS-SV (Algorithm 1) in terms of convergence rate and estimation accuracy across various trajectory lengths $T$. We analyze the results obtained from the dataset $\tau^b$ focusing on a scenario where the prescribed target distribution $\overline{\eta}$ is a point mass on a single state within the grid-world.

Figure 5 illustrates that convergence to the desired equilibrium $\widetilde{\eta}$, as defined in Theorem 2.1, is only achieved when the perturbed FMSS Problem (93) is solved using a sufficiently accurate estimate of the transition kernel $\mathbf{P}$. We emphasize that these curves do not represent the convergence of the induced chain $\mathbf{P}^{\widehat{\pi}^*}$ w.r.t its own empirical stationary distribution $\eta^{\widehat{\pi}^*}$; instead, they track the distance to the *true* target distribution $\widetilde{\eta}$. This distinction allows for a clear assessment of how the approximation error $\|\mathbf{P} - \widehat{\mathbf{P}}\|_2$ impacts our ability to realize the intended equilibrium in the underlying unknown environment.

Figure 6a validates the sample complexity analysis provided in Section 5.1. As expected, the estimation error $\|\mathbf{P} - \widehat{\mathbf{P}}\|_2$ decreases with the increase of trajectory length $T$.

Finally, the error propagation analysis conducted in Section 5.2 finds empirical validation in Figures 6b, 7a, and 7b. Specifically:

- Figure 6b shows that the difference between the optimal values of the perturbed FMSS Problem (93) and the exact FMSS Problem (92), denoted by $\left|\|\widetilde{\mathbf{A}}_{\widetilde{\eta}}^{\pi^*} - \widetilde{\mathbf{A}}_{\eta^{\pi^*}}^{\widehat{\pi}^*}\|_2\right|$, vanishes as $T$ increases.

- Figure 7a demonstrates that $\eta^{\widehat{\pi}^*}$ converges to the desired $\widetilde{\eta}$.

- Figure 7b confirms that the solutions of PROJ($\overline{\eta}$,$\mathbf{P}$) and PROJ($\overline{\eta}$,$\widehat{\mathbf{P}}$) also converge.

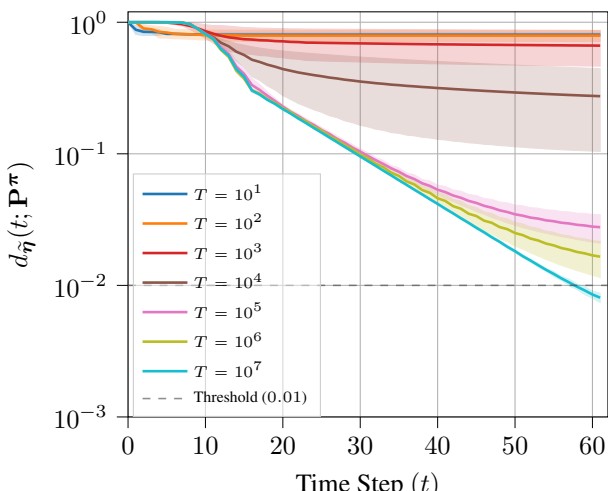

*Figure 5.* Distance $d_{\widetilde{\eta}}(t; \mathbf{P}^{\pi}) := \max_{s \in \mathcal{S}} \|p^{\pi,t}(\cdot|s) - \widetilde{\eta}\|_{\mathrm{TV}}$ from the target distribution $\widetilde{\eta}$ across varying trajectory lengths $T$. We note that convergence is measured relative to the optimal equilibrium distribution $\widetilde{\eta}$ derived from PROJ($\overline{\eta}$,$\mathbf{P}$) Problem (90), rather than to the equilibrium distribution $\eta^{\widehat{\pi}^*}$ induced by $\widehat{\pi}^*$ obtained from the perturbed FMSS Problem (93). This comparison allows to observe the occurrence of convergence to the true desired $\widetilde{\eta}$ when the policy $\widehat{\pi}^*$ (learned from empirical data) is deployed in the true environment $\mathbf{P}$.

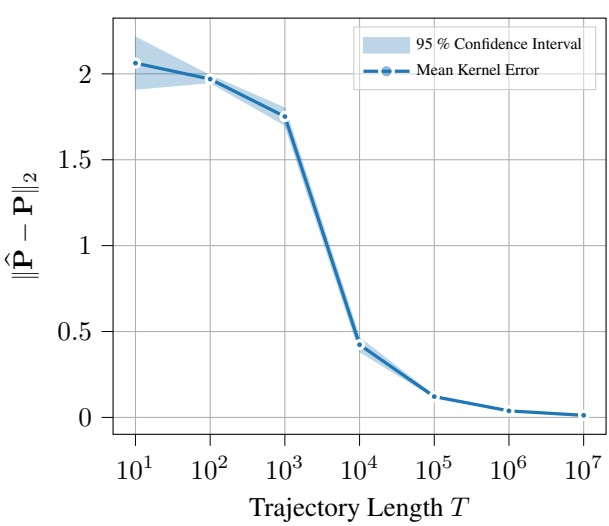

*(a)* Spectral norm of the difference between the true $\mathbf{P}$ and the estimated $\widehat{\mathbf{P}}$.

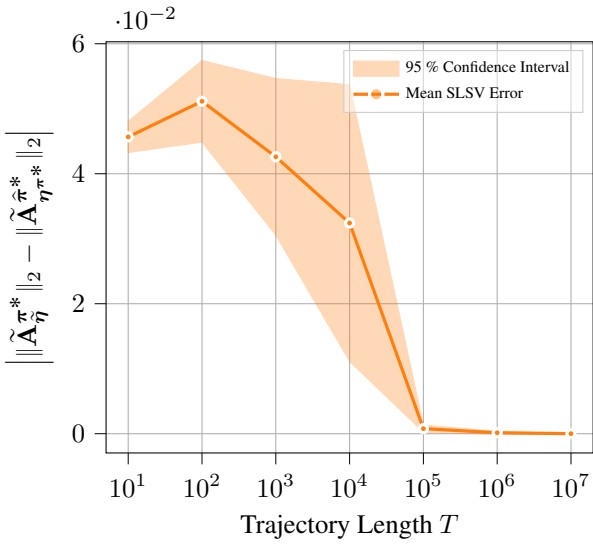

*(b)* SLSV estimation error for different trajectory lengths.

*Figure 6.* Estimation error on $\mathbf{P}$ and suboptimality gap. As anticipated by Theorem 5.1, a higher estimation accuracy is achievable with an increasing value of the trajectory length $T$. Moreover, as the theoretical analysis in Section 5.2 suggested, the difference $\left|\|\widetilde{\mathbf{A}}_{\widetilde{\boldsymbol{\eta}}}^{\boldsymbol{\pi}^*}\|_2 - \|\widetilde{\mathbf{A}}_{\boldsymbol{\eta}^{\widehat{\boldsymbol{\pi}}*}}^{\widehat{\boldsymbol{\pi}}^*}\|_2\right|$ goes to zero as the accuracy in the estimation of $\mathbf{P}$ increases.

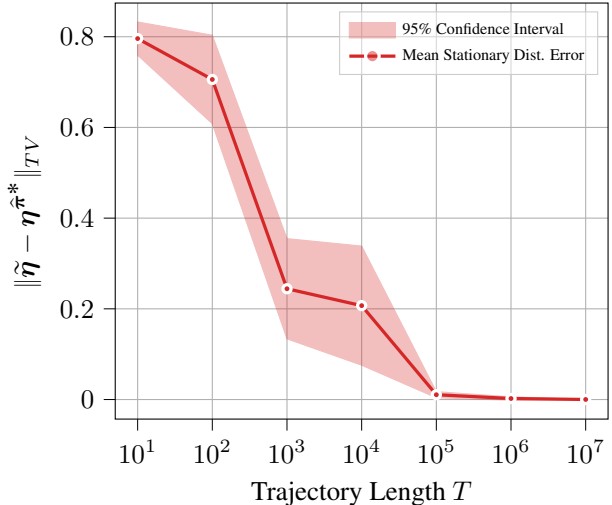

*(a)* Total Variation Distance between the solution $\widetilde{\boldsymbol{\eta}}$ induced by the exact FMSS (92) and $\boldsymbol{\eta}^{\widehat{\boldsymbol{\pi}}^*}$ induced by $\widehat{\boldsymbol{\pi}}^*$ solution of the perturbed FMSS (93).

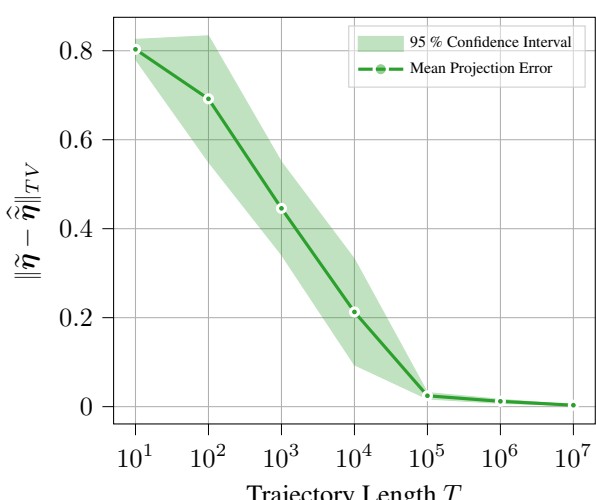

*(b)* Total Variation Distance between the solution $\widetilde{\boldsymbol{\eta}}$ of PROJ$(\overline{\boldsymbol{\eta}},\mathbf{P})$ (90) and $\widehat{\widetilde{\boldsymbol{\eta}}}$ solution of the perturbed PROJ$(\overline{\boldsymbol{\eta}},\widehat{\mathbf{P}})$ (91).

*Figure 7.* Estimation error for the stationary distributions and projection outputs across varying trajectory lengths $T$. These results compare the discrepancies between the exact and perturbed versions of both the FMSS problem and the projection problem. As predicted by the error propagation analysis in Section 5.2, the solutions to the perturbed problems converge toward their exact counterparts as $T$ increases.

## F.3. Uniform Target Distribution

In this part, we report the results obtained in the same grid-world environment, but solving `FMSS-SV` with respect to a uniform target distribution $\overline{\eta}(s) = 1/|\mathcal{S}|$ for all $s \in \mathcal{S}$.

While the overall observations from Appendix F.2 hold in this case, a few specific remarks are in order. The non-monotonic trends observed in the curves can be attributed to the mechanism for constructing the estimate $\widehat{\mathbf{P}}$ in Algorithm 1, which ensures ergodicity according to Assumption 2.1. When the trajectory length $T$ is extremely small, the estimated transition kernel tends to resemble a uniform transition kernel. Consequently, the induced chain is likely to converge rapidly to a uniform stationary distribution over the state space regardless of the policy. This creates a deceptively low error and an extremely fast convergence rate initially, before the true kernel structure is learned. This behavior is further accentuated by the inherent symmetry of the grid-world environment, mentioned in Appendix F.1.

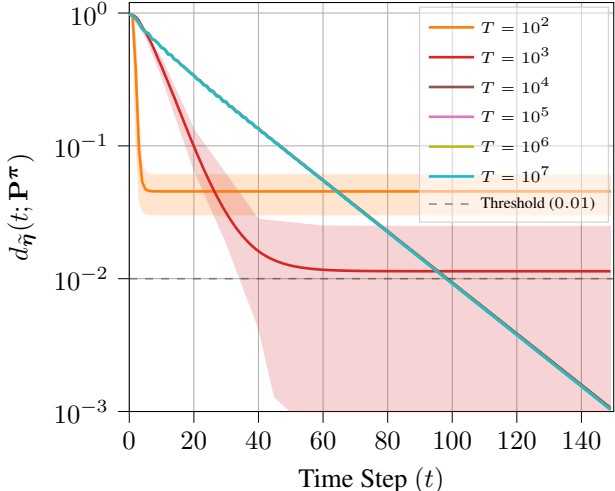

*Figure 8.* Distance $d_{\widetilde{\eta}}(t; \mathbf{P}^{\pi}) := \max_{s \in \mathcal{S}} \|p^{\pi,t}(\cdot|s) - \widetilde{\eta}\|_{\mathrm{TV}}$ from the target distribution $\widetilde{\eta}$ across varying trajectory lengths $T$. Convergence is measured relative to the optimal equilibrium distribution $\widetilde{\eta}$ derived from $\mathrm{PROJ}(\overline{\eta}, \mathbf{P})$ Problem (90), rather than the equilibrium distribution $\eta^{\widehat{\pi}^*}$ induced by $\widehat{\pi}^*$ (obtained from the perturbed FMSS Problem (93)). This comparison allows us to assess whether the policy $\widehat{\pi}^*$ (learned from empirical data) actually leads to the desired $\widetilde{\eta}$ when deployed in the true environment $\mathbf{P}$. Note the extremely steep curves for $T = 10^2$ and $T = 10^3$. This behavior arises because, at such small sample sizes, the estimated transition kernel is close to uniform (ensuring fast mixing toward a uniform distribution) but far from the true structural dynamics, resulting in a failure to converge to the specific target $\widetilde{\eta}$. The large confidence interval characterizing the curve associated with $T = 10^3$ must be attributed to both the logarithmic scale and the large variance of the results for such a value of $T$.

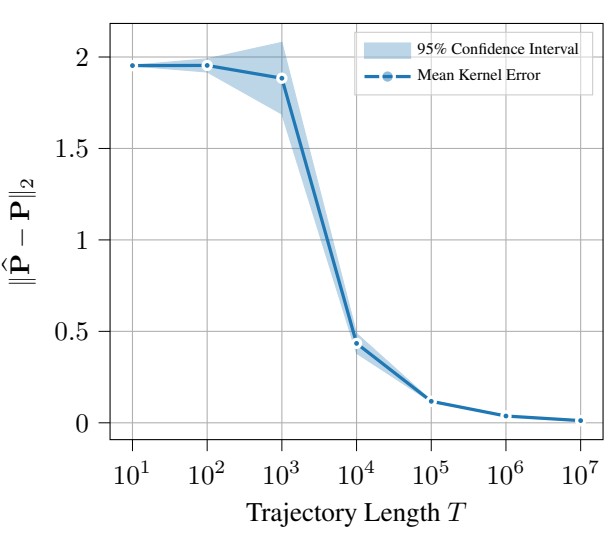

*(a)* Spectral norm of the difference between the true $\mathbf{P}$ and the estimated $\widehat{\mathbf{P}}$.

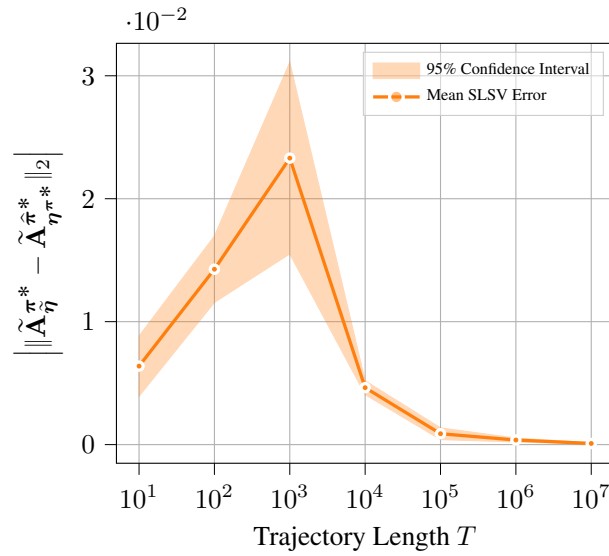

*(b)* SLSV estimation error for different trajectory lengths.

*Figure 9.* Estimation error on $\mathbf{P}$ and suboptimality gap. These figures illustrate that for small values of $T$, the estimated kernel $\widehat{\mathbf{P}}$ remains far from the true kernel $\mathbf{P}$ (as shown in Figure 9a). This structural discrepancy explains the failure to converge to $\widetilde{\boldsymbol{\eta}}$ in Figure 8, despite the low SLEM estimation error observed in Figure 9b, which might misleadingly suggest accurate convergence properties.

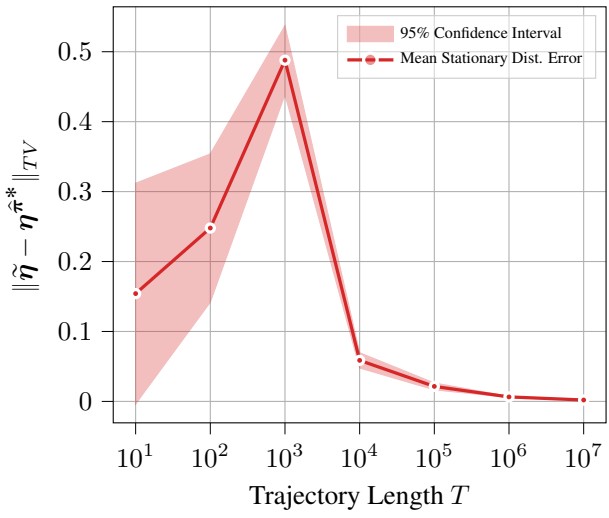

*(a)* Total Variation Distance between the solution $\widetilde{\boldsymbol{\eta}}$ induced by the exact FMSS Problem (92) and $\boldsymbol{\eta}^{\widehat{\boldsymbol{\pi}}^*}$ induced by $\widehat{\boldsymbol{\pi}}^*$ solution of the perturbed FMSS Problem (93).

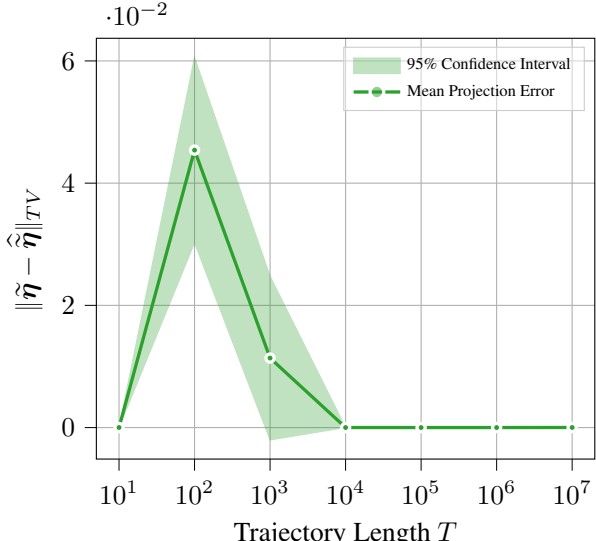

*(b)* Total Variation Distance between the solution $\widetilde{\boldsymbol{\eta}}$ of PROJ($\overline{\boldsymbol{\eta}}$,$\mathbf{P}$) Problem (90) and $\widehat{\widetilde{\boldsymbol{\eta}}}$ solution of the perturbed PROJ($\overline{\boldsymbol{\eta}}$,$\widehat{\mathbf{P}}$) Problem (91).

*Figure 10.* Estimation error for the stationary distributions and projection outputs across varying trajectory lengths $T$. These results confirm that the trends and conclusions observed for the previously discussed quantities also hold for these metrics.

### F.4. Comparison of the `FMSS-SV` Algorithm with a Minimum Exploration Probability Policy for the Grid-World Environment

In this part, we consider the grid-world environment described in Appendix F.1 and compare the convergence rate achieved by solving the exact FMSS Problem (92) against a simple heuristic designed to induce uniform coverage of the state space.

Given a uniform goal distribution $\overline{\eta}$, after solving the projection problem PROJ($\overline{\eta}$,$\mathbf{P}$), we compare $\pi^*$ obtained by solving FMSS($\widetilde{\eta}$,$\mathbf{P}$) with the solution of the following feasibility program,

$$
\begin{aligned}
\underset{\pi \in \Pi^{\mathrm{MR}}}{\text{minimize}} \quad & 0 \\
\text{subject to} \quad & \widetilde{\eta} = (\mathbf{P}^{\pi})^{\top}\widetilde{\eta}, \\
& \pi(a|s) \geqslant \xi, \quad \forall(s,a) \in \mathcal{S} \times \mathcal{A}
\end{aligned}
\tag{167}
$$

Importantly, this program enforces the selection of policies $\pi \in \Pi_{\widetilde{\eta}}$ with a minimum exploration constraint $\xi$, denote this policy as $\pi_{\xi}$. Figure 11 shows how in this specific grid-world environment and for $\xi = 0.2$, our approach can achieve better convergence rates than this simple heuristic.

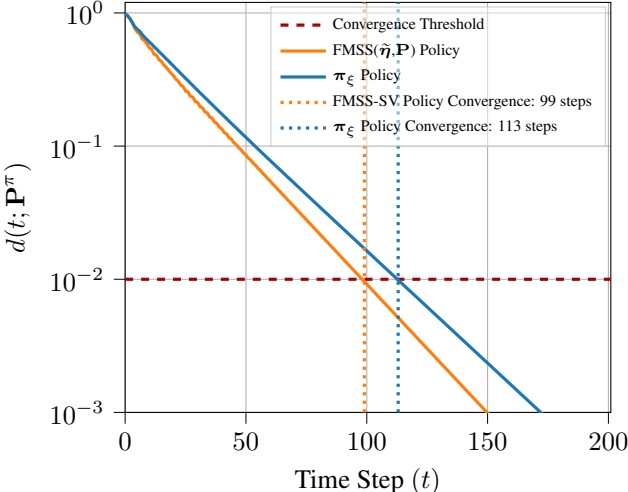

*Figure 11.* Distance from equilibrium for the FMSS($\widetilde{\eta}$,$\mathbf{P}$) and for the minimum entropy policy $\pi_{\xi}$. For completeness we report the values of the SLEM induced by the respective policies, $\nu(\mathbf{P}^{\pi^*}) = 0.9564$, $\nu(\mathbf{P}^{\pi_{\xi}}) = 0.9620$.

