# OpenReview forum: "Fast Mixing Steady-State Control in Markov Decision Processes"
_ICML.cc/2026/Conference — ICML 2026 regular_

### Official Review · Reviewer_QrdV · 2026-02-20

**Soundness:** 4
**Presentation:** 3
**Significance:** 4
**Originality:** 3
**Overall Recommendation:** 5
**Confidence:** 3

**Summary:**

This paper formulates a fast-mixing steady state (FMSS) optimization problem (4) through replacing the non-convex objective function with a surrogate second-largest singular value (SLSV) function, so that the resulting policy will guarantee to converge to the steady state. The main contributions of this paper include: 1)  A convex optimization problem formulation minimizing the SLSV of the original non-convex problem minimizing the SLEM; 2) theoretical analysis of the fomulated optimization problem (4), motivating the need of the fastest convergent policy. The problem is clearly stated and the idea is interesting. The paper is well-structured and easy to follow. The derivation seems reasonable and the examples in the appendix are helpful. I think it is a solid conference paper.

**Compliance With Llm Reviewing Policy:**

Affirmed.

**Final Justification:**

I think this is a good work, from a control design perspective. So I suggest accept.

**Key Questions For Authors:**

Please see ``Strengths And Weaknesses''.

**Limitations:**

Yes

**Strengths And Weaknesses:**

Soundness and presentation: This paper is well-written, self-contained and the idea is new to me. The derivations and examples in the appendices are reasonable and seems correct.

Significance: The studied problem can be quite useful in my opinion since determining the fastest convergent policy in MDP can be a general and useful technique which can increase the computational efficiency.

Originality: I am so sure about the originality since I am not quite familiar with the directly related articles, but it seems reasonable in a controller design perspective.

Other comments: The only concern left for me is the analysis in section 4.2. I think the surrogate approximation gap is usually quite influential for the convergence property. When I read this section, I expect to see more rigorious analysis. Other minor comments:
-Reference Levin et al. 2017 is duplicated.
-There are some formating and abbreviation issues in the references.

---

> ### Author Rebuttal · Authors · 2026-03-30
>
> We thank the Reviewer for their positive assessment of our work and for pointing out the minor formatting issues (e.g., references and abbreviations). We will ensure these are corrected in the final version of the manuscript.
>
> > The only concern left for me is the analysis in section 4.2. I think the surrogate approximation gap is usually quite influential for the convergence property. When I read this section, I expect to see more rigorious analysis.
>
> **A sharper analysis of the approximation gap $\left(\sigma_2(\mathbf{P}^{\pi})-\nu(\mathbf{P}^{\pi})\right)$**: As customary in matrix analysis literature, we quantify the departure from normality using the Frobenius norm of the nilpotent matrix $\mathbf{N}$ from the Schur decomposition of matrix $\tilde{\mathbf{A}}\_{\overline{\eta}}^{\pi}$, defined as **Henrici’s Departure from Normality** $h(\mathbf{P}^{\pi}) = \\|\mathbf{N}\\|\_{F} =\sqrt{\sum_{i \in [|S|]} (\sigma_{i}^{2}(\mathbf{P}^{\pi}) - |\lambda_{i}^{\pi}|^{2})}$ (Table 1). A sharper analysis of the gap between $\sigma_2(\mathbf{P}^\pi)$ and $\nu(\mathbf{P}^\pi)$ can be framed as the task of upper-bounding the difference between the spectral norm and spectral radius ($\rho(\cdot)$) of the matrix $\widetilde{\mathbf{A}}^{\pi}_{\overline{\eta}}$:
>
> $$0 \leq \\|\widetilde{\mathbf{A}}^{\pi}\_{\overline{\eta}}\\|\_2 - \rho(\widetilde{\mathbf{A}}^{\pi}_{\overline{\eta}}) \leq \\|\mathbf{N}\\|\_2$$
>
> While $\\|\mathbf{N}\\|\_2 \leq \\|\mathbf{N}\\|\_F$ provides a tighter bound, this approach faces two limitations. First, since both $\\|\widetilde{\mathbf{A}}^{\pi}\_{\overline{\eta}}\\|\_2$ and $\rho(\widetilde{\mathbf{A}}^{\pi}\_{\overline{\eta}})$ are bounded by 1, the characterization is only informative when $\\|\mathbf{N}\\|\_2 < 1$.
> Second, because these norms derive from the Schur decomposition, it is not clear whether it is possible to analytically determine how $\\|\mathbf{N}\\|\_F$ or $\\|\mathbf{N}\\|\_2$ depend on the MDP dynamics $\mathbf{P}$ or policy $\pi$.
>
> We will include the details discussed above in the final version of the work.

---

> > ### Author Rebuttal · Reviewer_QrdV · 2026-04-01
> >
> > Thank you for the reply. I appreciate the analysis regarding my concern.
> > I have no further comments, and the paper is acceptable at this stage. I will keep my score unchanged.

---

### Official Review · Reviewer_XJcL · 2026-03-12

**Soundness:** 3
**Presentation:** 3
**Significance:** 3
**Originality:** 3
**Overall Recommendation:** 5
**Confidence:** 3

**Summary:**

This manuscript investigates the translation of the steady-state control problem from Control Theory (CT) into the Markov Decision Process (MDP) framework. Given an ergodic MDP and a target equilibrium distribution, the objective is to synthesize a Markovian policy that induces this distribution while maximizing the convergence rate.

The convergence rate is fundamentally dictated by the spectral properties of the induced transition matrix, specifically the second-largest eigenvalue modulus (SLEM). Because direct optimization of the SLEM is generally non-convex for non-reversible Markov Chains, the authors propose a tractable convex surrogate: minimizing the second-largest singular value (SLSV).

The authors further introduce a projection approach (PROJ) to handle non-realizable target distributions and prove that the feasible policy space for a realized distribution typically possesses infinite cardinality, necessitating a selection criterion like fast mixing. Finally, they provide an offline learning algorithm (FMSS-SV) with finite-sample complexity guarantees and an error propagation analysis characterizing the impact of model estimation errors on the resulting policy performance.

**Compliance With Llm Reviewing Policy:**

Affirmed.

**Key Questions For Authors:**

## Minor Comments
1. **Sensitivity to Coverage:** The analysis assumes a fully exploratory behavioral policy (\(\pi^b(a|s) \ge \xi > 0\)). It would be beneficial to discuss the algorithm's sensitivity to weaker coverage or how the framework might be extended to function approximation settings to move beyond tabular MDPs.
2. **Regularization for Normality:** Given that non-normality is the primary bottleneck for the SLSV surrogate, could the objective in Program (4) be augmented with a regularization term to incentivize more normal transition matrices (e.g., minimizing the skew-symmetric component \(\|\tilde{A}_\pi - \tilde{A}_\pi^\top\|_F\))?
3. **Mixing Time Dependence:** In Theorem 5.1, the sample complexity exhibits a linear dependence on \(t_{mix}^b\). Does this dependence persist if the trajectory is assumed to start from the stationary distribution \(\eta^b\)?

**Limitations:**

yes

**Strengths And Weaknesses:**

## Major Comments
1. **SLSV as a Surrogate for SLEM:** The central technical contribution relies on using the SLSV (\(\sigma_2\)) as a proxy for the SLEM (\(\nu\)). The authors acknowledge that while these coincide for normal or reversible chains, the approximation degrades as the system departs from normality. While Table 1 and Section 4.2 discuss this qualitatively, the review would be strengthened by a sharper characterization of the rate of this degradation. Furthermore, the empirical results in Appendix A demonstrate that in highly non-normal settings, the SLSV-optimal policy can deviate significantly from the SLEM-optimal one. Could the error propagation analysis explicitly incorporate the departure from normality without making the problem nonconvex?

2. **End-to-End Sample Complexity:** Theorem 5.1 bounds the trajectory length needed to guarantee \(\|\hat{P} - P\|_2 \leq \varepsilon\), i.e., transition model estimation error. The end-to-end guarantee one really wants is a bound on \(|\sigma_2(P^{\hat{\pi}}) - \sigma_2(P^{\pi^\star})|\), the gap in the SLSV between the learned policy and the optimal policy.

   Theorems 5.2 and 5.3 provide partial steps in this direction, but it would be nice to have an end to end result for a final sample complexity bound.

---

> ### Author Rebuttal · Authors · 2026-03-30
>
> We thank the Reviewer the comments and suggestions.
> ### Major Comments
> **Q1**:
>
> **Characterization of Departure from normality:** Due to space limits, we refer the Reviewer to the bounds in Table 1 and to the answer to **Reviewer QrdV**.
>
> **Departure from normality in the analysis**: Error propagation (Th. 5.3) focuses on how transition matrix estimation errors affect the SLSV ($\sigma_2(\mathbf{P}^\pi)$) objective. Departure from normality can be used to characterize the gap between the SLEM ($\nu(\mathbf{P}^\pi)$) of the nominal FMSS problem and that of the approximate FMSS as follows:
> $$
> \sigma_2(\mathbf{P}^\pi) - \\|\mathbf{N}\\|\_2 \le \nu(\mathbf{P}^\pi) \le \sigma_2(\mathbf{P}^\pi),
> $$
> we can extend Th. 5.3 to bound:
> $$
>     |\nu(\mathbf{P}^{\pi^\*}) - \nu(\mathbf{P}^{\widehat{\pi^\*}})| \le |\sigma\_2(\mathbf{P}^{\pi^\*}) - \sigma\_2(\mathbf{P}^{\widehat{\pi}^\*})| + \max\\{\\|\mathbf{N}^{\pi^*}\\|\_2,\\|\mathbf{N}^{\widehat{\pi^\*}}\\|\_2\\}.
> $$
> This shows how the difference of perfromance in terms of the SLEM is related to the departure from normality, as expected.
>
> **Departure from normality in the algorithms**: Incorporating $h(\mathbf{P}^\pi)$ (see Table 1) directly into the objective would break convexity, as it depends on the absolute values of the eigenvalues. Similarly,  the commutator $\\|(\tilde{A}\_{\eta}^\pi)^{\top} \tilde{A}\_{\eta}^\pi - \tilde{A}\_{\eta}^\pi (\tilde{A}\_{\eta}^\pi)^{\top}\\|^2\_F$ , an alternative measure of non-normality,  is non-convex in the policy entries.
>
> **Q2**: An end-to-end bound requires choosing a divergence $\mathcal{D}$ and will lead to a quite convoluted expression. This is why we decided to omit it in the paper, but we report it here for completeness. For instance, let $\mathcal{D}(\eta\\|\eta') = \\|\eta- \eta'\\|\_2$. First, we deal with the approximation error caused by the non-realizability of the target distribution $\overline{\eta}$:
>
> $$\\|\hat{\tilde{\eta}} - \tilde{\eta}\\|\_2 \le 2 \underbrace{\\|\overline{\eta} - \tilde{\eta}\\|\_2}\_{\\text{approximation error}} + \sqrt{|\mathcal{S}|} \sup_{\pi \in \Pi^{MR}} \\|Z^\pi\\|\_2 \\|\mathbf{P}-\hat{\mathbf{P}}\\|\_2.$$
>
> Last inequality follows from Th. 5.2 ($L=1$). Plugging this into Th. 5.3, for $\\|\hat{\mathbf{P}}-\mathbf{P}\\|\_2 \leq 1$, yields:
>
> $$
> |\sigma\_2(P^{\pi^\*}) - \sigma\_2(P^{\widehat{\pi}^\*})| \le O\\left(\Gamma \\left(\\underbrace{\sqrt{\\|\overline{\eta} - \tilde{\eta}\\|\_2}}\_{\\text{irreducible}} + \\underbrace{\sqrt{\\|\mathbf{P}-\hat{\mathbf{P}}\\|\_2}}\_{\\text{reducible}}\\right)\\right),
> $$
>
> where $\Gamma$ aggregates problem-dependent constants.
>
> Imposing the latter bound to be less than $\epsilon\_{apx} + \epsilon$, where $\epsilon\_{apx}$ is the irreducible approximation error due to $\left\\|\overline{\eta} - \widetilde{\eta}\right\\|\_2$, we can characterize the sample complexity of the reducible error following similar steps as in Th. 5.1:
> $$ T \leq \widetilde{O}\left(\frac{\\|\mathbf{Z}^{\hat{\pi}^\*}\\|^4\_2|\mathcal S|^4|\mathcal A|^3(1+\widetilde{H})^4\left(1+(\sup\_{\pi}\\|\mathbf{Z}^{\pi}\\|\_2)^2\right)(1+t^b\_{\\mathrm{mix}})\ln\left(\frac{1}{\delta}\right)}{(\eta^{\dagger}\_{\min})^{12}\epsilon^4\eta^b\_{\min}\xi}\left(\ln\left(\frac{1}{\delta}\right)+|\mathcal S|\right)\right),$$
> where $\eta\_{\min}^\dagger = \min\\{\eta\_{\min}^{\hat{\pi}^\*}, \tilde{\eta}\_{\min}, \hat{\tilde{\eta}}\_{\min}\\}$.
>
> ### Minor Comments
> **Q1**: If the behavioral policy $\pi^b$ fails to satisfy Assumption 5.1, the induced chain over state-action space, $\mathbf{T}^b = \mathbf{P}\mathbf{\Pi}^b$, may not be ergodic. Specifically, $\mathbf{T}^b$ may be **unichain**, containing transient states where $x^b(s,a) = 0$.  This limited coverage prevents estimation of the transition matrix over the unvisited pairs $\mathcal Z \subset \mathcal{S \times A}$, where the estimation error remains irreducible:
> $$\\|\widehat{\mathbf{P}}-\mathbf{P}\\|\_2 \geq \min\_{(s,a) \in \mathcal Z}\left|\frac{1}{|\mathcal S|}-p(s^\prime|s,a)\right|.$$
> While sample complexity on the ergodic subset remains unchanged, global convergence of the model estimate is no longer guaranteed.
>
> For what concerns the extension to continuous spaces, we refer to the answer to **Reviewer u3xK**.
>
> **Q2**: While the proposed regularizer $\\|\tilde{\mathbf{A}}\_{\eta}^\pi - (\tilde{\mathbf{A}}\_{\eta}^\pi)^{\top}\\|\_F$ is convex, it penalizes the skew-symmetric component of the matrix. Consequently, it favors symmetric matrices and thus reversibility. On top of that, many normal matrices are non-symmetric, so the regularizer may be overly restrictive.
>
> **Q3**: Initial distribution affects the sample complexity only logarithmically through the $\\|\mu\_0\\|\_{\ell^2(1/\eta)}$ term in Lem E.1. When starting from the stationary distribution ($\mu\_0 = \eta$), this term equals $1$. The mixing time dependence ($T\_\zeta$ in Lem E.1) remains unchanged.
>
> We will add all these considerations in the final version of the paper.

---

> > ### Author Rebuttal · Reviewer_XJcL · 2026-04-01
> >
> > I thank the authors for their detailed response. My concerns have been resolved and I am increasing the score to accept.

---

### Official Review · Reviewer_CUXU · 2026-03-12

**Soundness:** 3
**Presentation:** 2
**Significance:** 3
**Originality:** 2
**Overall Recommendation:** 3
**Confidence:** 3

**Summary:**

The submission aims to extend steady-state regulation ideas from deterministic control to finite ergodic MDPs by introducing the fast-mixing steady-state (FMSS) control problem. The goal is to find a stationary randomized policy whose induced Markov chain converges to a desired stationary distribution as quickly as possible. Because directly optimizing the usual spectral measure of mixing speed is generally non-convex, the paper replaces it with a convex surrogate based on singular values, which leads to a tractable convex optimization formulation with a constraint ensuring the target stationary distribution. If the desired distribution cannot be achieved exactly, the method first projects it onto the closest feasible stationary distribution and then solves the FMSS problem for this projected target. For the offline learning setting, the paper proposes the FMSS-SV algorithm: it estimates the transition model from a single exploratory trajectory, regularizes the estimate to maintain ergodicity, performs the projection step on the estimated model, and then solves the convex FMSS problem using the estimated quantities. The paper also provides finite-sample guarantees and analyzes how estimation errors affect the final solution. Overall, the work attempts to study an important concept: explicitly designing RL policies to control steady-state behavior and mixing speed, rather than treating them as secondary properties.

**Compliance With Llm Reviewing Policy:**

Affirmed.

**Key Questions For Authors:**

1. It would be helpful if the authors could further clarify the practical implications of the proposed framework. In particular, what are the intended real-world applications where fast-mixing steady-state control would provide a clear advantage? While the theoretical formulation is interesting, it would strengthen the paper to discuss concrete settings (e.g., robotics, resource allocation, network control) where targeting a specific steady-state distribution and optimizing the mixing rate is practically useful.

2. Additionally, the contribution relative to existing work could be clarified further. There is a substantial literature on fastest-mixing Markov chains, Markov chain design with prescribed stationary distributions, and spectral optimization for mixing time. It would be useful if the authors could more explicitly articulate how the proposed formulation and algorithm differ from or improve upon these existing approaches. In particular, what is the key conceptual or technical novelty compared to prior methods that also optimize spectral properties of Markov chains under stationary distribution constraints?

3. The proposed approach replaces the minimization of the second-largest eigenvalue modulus (SLEM) with the second-largest singular value (SLSV) as a convex surrogate. While this makes the optimization tractable, it would be helpful if the authors could clarify how well minimizing SLSV approximates minimizing the true mixing rate. In particular, for non-normal Markov chains the singular values and eigenvalues can differ substantially. Are there theoretical guarantees or bounds that relate the SLSV objective to the actual mixing time or spectral gap?

4. The paper argues that the SLSV-based objective leads to a convex optimization problem. However, convexity is typically defined in the space of transition matrices, whereas the actual decision variable in an MDP is the policy $\pi$. Could the authors clarify under what conditions convexity is preserved when optimizing over policies? In particular, does the convexity claim still hold when policies are restricted to deterministic or parameterized forms?

5. The framework assumes a desired stationary distribution $\eta$, which may not be realizable by the given MDP. The paper proposes projecting $\eta$ onto the feasible set. It would be helpful to clarify how this feasible set is characterized in practice and whether the projection problem is always convex. Additionally, are there guarantees that the projected distribution corresponds to a valid policy-induced stationary distribution?

6. The paper measures convergence speed via spectral quantities such as SLEM or SLSV. However, practical mixing behavior is typically characterized in terms of total variation mixing time. Could the authors elaborate on how the proposed spectral objective relates to standard mixing-time bounds? In particular, are there guarantees that improving the SLSV objective necessarily improves the actual mixing time?

**Limitations:**

The method is limited to finite tabular MDPs and relies on strong ergodicity and sufficient exploratory data coverage, assumptions that may not hold in many practical control or RL settings with safety constraints or large or continuous state spaces. The optimization objective is based on a singular-value surrogate, which is exact only for certain types of Markov chains and may not accurately reflect the true mixing speed in more general cases. In addition, solving the proposed convex optimization problem may become computationally expensive as the state and action spaces grow, and the offline algorithm depends on accurate transition model estimation. Finally, the experimental evaluation is relatively limited, making it unclear how robust the approach is across a wider range of MDP structures.

**Strengths And Weaknesses:**

The paper introduces the fast-mixing steady-state (FMSS) control problem, which seeks a policy that drives an MDP toward a desired stationary distribution as quickly as possible. Since directly optimizing the usual spectral measure of mixing speed is non-convex, the authors propose a convex surrogate based on singular values, making the problem tractable. If the target distribution is not achievable, the method first projects it onto the closest feasible stationary distribution and then solves the FMSS problem for this projected target. For offline learning, the paper proposes the FMSS-SV algorithm, which estimates the transition model from data, enforces ergodicity, performs the projection step, and solves the resulting convex optimization problem. The paper also provides finite-sample guarantees and analyzes how estimation errors affect the final policy. Overall, the work studies how to explicitly design RL policies to control steady-state behavior and mixing speed.

---

> ### Author Rebuttal · Authors · 2026-03-30
>
> We appreciate the Reviewer’s comments and questions. Below the responses to each of his key questions.
>
> **Q1**: Our work is primarily motivated by stabilization and regulation tasks in stochastic MDPs. Specifically, we address controller design for systems where the transition kernel $p(s'|s,a)$ is irreducible, e.g., because of noise/disturbances. In such scenarios, the only meaningful notion of stability is fast convergence of the state distribution to a target invariant measure, for instance, one concentrated around a specific "equilibrium".
>
> Regarding real-world applications, the method can be useful in swarm robotics [1,2] to control the distribution of the ensemble of agents. Also, our work resemble *optimal transport*, where the objective is to construct a map such that a given probability measure is pushed forward to a target probability measure. Here, applications concern resource allocation problems [3]. Our framework also aligns with ergodic control in robotics [4], where the goal is to ensure the system's time-averaged behavior matches a desired spatial distribution.
> We will mention these applications in the final version of the paper.
>
> [1] Açikmeşe, B. and Bayard, D.S., A Markov chain approach to probabilistic swarm guidance. American Control Conference, 2012.
>
> [2] Elamvazhuthi, K. and Berman, S., Nonlinear generalizations of diffusion-based coverage by robotic swarms. IEEE Conference on Decision and Control, 2018.
>
> [3] Santambrogio, F., Optimal Transport for Applied Mathematicians. Calculus of Variations, PDEs and Modeling, 2015.
>
> [4] G. Mathew and I. Mezic, Metrics for ergodicity and design of ergodic dynamics for multi-agent system. Physica D-nonlinear Phenomena, 2011.
>
> **Q2**: While we acknowledge the vast body of literature concerning Markov Chains (MC) what distinguishes our work is:
>
> **Optimization Constrained by MDP Dynamics:** Existing literature directly optimizes Markov chain transition entries, assigning arbitrary probabilities to the underlying graph edges [Boyd et al., 2004]. Conversely, we are the first to address fast-mixing within MDPs, where the transition kernel imposes additional structure constraining both convergence rates and reachable target distributions. Indeed, the policy is the sole optimization variable here. Importantly, our approach addresses the distributional constraints through the **projection step ($PROJ$)**.
>
> **Generalization to Non-Reversible Chains:** Most literature focuses on reversible Markov Chains (MCs) [Boyd et al. 2004; Mutti & Restelli, 2020; Tarbouriech & Lazarich, 2019] to ensure a convex SLEM optimization (see Sections 1, 2, and 4). However, as shown in Appendix A (Figure 1a), an arbitrary MDP’s dynamics $\mathbf{P}$ may not allow any policy $\pi \in \Pi^{MR}$ to realize a reversible MC. By overcoming this limitation, our **Fast-Mixing Steady-State (FMSS)** formulation avoids infeasibility and enables the synthesis of non-reversible MCs.
>
> **Tractable Convex Formulation via SLSV:** Since non-reversible chains result in a non-convex SLEM objective, we propose a tractable convex surrogate. Our key technical novelty is the derivation of an objective based on the **Second-Largest Singular Value (SLSV)** of the transition matrix $\mathbf{P}^{\pi}$.
>
> **Q3&Q6**: Table 1 and our response to **Reviewer QrdV** characterize the approximation gap between SLEM ($\nu(\mathbf{P}\_\pi)$) and SLSV ($\sigma_2(\mathbf{P}\_\pi)$). As shown in Section 2, spectral properties provide an upper bound on the total variation distance from equilibrium, $d(t; \mathbf{P}^\pi)$. For non-reversible chains, this rate is evaluated via the multiplicative reversibilization $M(\mathbf{P}^\pi) := \mathbf{P}^\pi \tilde{\mathbf{P}}^\pi$. Since $\sigma_2(\mathbf{P}^\pi) = \sqrt{\nu(M(\mathbf{P}^\pi))}$, Fill’s Theorem (1991) establishes the relation between SLSV and mixing time:
> $$d(t;\mathbf{P}^{\pi}) \le\frac{1}{2\sqrt{\eta_{\min}^{\pi}}}\sigma_2(\mathbf{P}^{\pi})^t \implies t_{\text{mix}}(\varepsilon) \le \frac{\log\left( \frac{1}{2\varepsilon \sqrt{\eta^\pi_{\min}}}\right)}{\log\left(1 / \sigma_2(\mathbf{P}^\pi)\right)}$$
> By optimizing the SLSV surrogate, we directly improve an upper bound on the actual mixing time.
>
> **Q4**: In Section 4, we established that, for a fixed feasible target distribution $\tilde{\eta}$, the set of feasible policies is a convex polytope. While the objective is convex w.r.t. randomized policies ($\pi \in \Pi^{MR}$), this property is lost when restricting to deterministic policies, which form a non-convex set. For parametric policies, convexity depends on the functional relationship between the parameters and the policy entries.
>
> **Q5**: Section 3.2 (Eq. 5 and 6) characterizes the feasible set as a linear polytope, ensuring that the projected distribution corresponds to a policy-induced stationary measure. Thus, the projection remains a convex optimization problem provided that a convex divergence is employed, such as $\ell^2$ or Kullback-Leibler (KL).

---

### Official Review · Reviewer_u3xK · 2026-03-14

**Soundness:** 3
**Presentation:** 3
**Significance:** 2
**Originality:** 3
**Overall Recommendation:** 4
**Confidence:** 2

**Summary:**

This paper studies the steady-state control problem within the Markov Decision Process (MDP) framework. Given an ergodic MDP and a target steady-state distribution $\eta$, the target is to find a Markovian policy that induces this distribution with the fastest possible convergence rate. The major contribution of this work is the introduction of Fast Mixing Steady-State control (FMSS) to address the above control problem.  The key novelty of FMSS lies in formalizing the problem as a convex program that minimizes the second-largest singular value rather than the second-largest eigenvalue. Furthermore, the authors establish a finite-time sample complexity guarantee using the FMSS framework.

**Compliance With Llm Reviewing Policy:**

Affirmed.

**Final Justification:**

I would like to keep my initial assessment. I think it is an interesting direction to replace the assumptions about $t_{mix}$ and $1/\eta_{min}$ with some more general assumptions since these two parameters could be very bad due to some dummy states.

**Key Questions For Authors:**

1. How to choose the behavior policy in Algorithm 1? Is it possible to improve the dependence on $t_{min}^b$ and $\eta_{min}^b$ if we have access to the online learning environment?

2. What is the difficulty to extend FMSS to the setting of linear MDP or MDP with general function approximation?

**Limitations:**

Yes.

**Strengths And Weaknesses:**

Strengths:

1. The FMSS problem bridges control theory's stability concepts with the MDP framework. Formulating policy synthesis as a spectral optimization problem with steady-state constraints is both theoretically interesting and practically relevant for applications requiring stability guarantees.

2.  Replacing the non-convex SLEM objective with the convex SLSV surrogate is a clever and well-justified approach. The paper explains when this surrogate is exact (reversible/normal chains) and provides meaningful bounds on the approximation gap.

3. The paper is well-organized and easy to follow.

Weaknesses:

1. The finite-time sample complexity bound depends on the mixing time $t_{mix}^b$ and the minimal stationary probability $1/\eta_{min}^b$ of the behavior policy. These two parameters can be prohibitively large when the behavior policy does not explore the state-action space sufficiently.



2. The algorithm and analysis are restricted to tabular MDPs. Extending FMSS-SV to function approximation or continuous state-action spaces would be more interesting.

---

> ### Author Rebuttal · Authors · 2026-03-30
>
> We thank the Reviewer for the insightful comments and constructive feedback. In the following, we elucidate the raised concerns.
>
> **Key Questions For Authors:**
> > The finite-time sample complexity bound depends on the mixing time $t_{\text{mix}}$ and the minimal stationary probability $1/\eta_{\text{min}}^b$ [...]
> > How to choose the behavior policy in Algorithm 1? [...] access to the online learning environment?
>
> **Tightness of the dependences on $t_{\text{mix}}^b$ and $1/\eta_{\text{min}}^b$**: First of all, we remark that in a concentration bound that controls estimates performed with samples coming from a Markov chain induced by a policy $\pi^b$, the dependence on $t_{\text{mix}}^b$ and $1/\eta_{\text{min}}^b$ **are unavoidable**. In particular:
> 1. The dependence on $t_{\text{mix}}^b$ is intrinsic when considering (dependent) samples coming from a Markov chain at convergence. It is due to the fact that a chain with mixing time $t^b_{\text{mix}}$, at convergence, only provides $O(n / t_{\text{mix}}^b)$ “effectively independent” samples over $n$ steps (see Chung et al., 2012, Theorem 3.1).
>
> 2. The dependence on $1/\eta_{\text{min}}^b$ derives from the fact that we are using offline data and we seek to guarantee a bound that holds uniformly over all states, i.e., evaluating the error in the 2-norm.
>  Thus, on average, the number of samples collected over $n$ steps for a state $s \in \mathcal{S}$ is $O(\eta^b(s) n)$, and the dependence on $1/\eta_{\text{min}}^b$ comes from considering the least visited state.
>
> **Controlling the values of  $t_{\text{mix}}^b$ and $1/\eta_{\text{min}}^b$**: A naive criterion, which does not require prior knowledge of the environment nor an online approach, is to choose $\pi^b$ to maximize $\xi$ in the sample complexity bound. Recalling the dependence $\widetilde{O}(\frac{1}{\eta^b_{\min} \xi})$, one might choose $\pi^b(a|s) = \xi = \frac{1}{|\mathcal{A}|}$, i.e., the uniform policy. However, it is difficult to predict how this choice might impact $\eta^b_{\min}$.
>
> A more grounded approach would be to integrate the insights provided in [1], where the authors tackle the problem of designing highly **exploratory** and fast-mixing policies to facilitate the subsequent policy design process for goal-oriented tasks. Importantly, they recognize the trade-off between ensuring a uniform exploration of the state space (i.e., a large $\eta_{\text{min}}^b$) versus a uniform exploration over the joint state-action space (i.e., a large $\eta_{\text{min}}^b\xi$). They explicitly provide examples in which the choice of a uniform policy might be suboptimal for maximizing the product $\eta^b_{\min}\xi$.
>
> Leveraging the approaches of [1], *when having access to an online environment*, to learn the behavioral policy $\pi^b$, would address the concerns emphasized by the Reviewer regarding the possibility of having a large $t^b_{\text{mix}}$ and  $1/(\eta^b_{\min}\xi)$ in a principled way. Of course, this would require a dedicated sample complexity analysis that accounts for both the samples required to (1) learn such a behavioral policy $\pi^b$ with the algorithms of [1] and (2) the samples required to guarantee $\\|\mathbf{P}-\widehat{\mathbf{P}}\\|_2 \leq 2\varepsilon$, as required in our work.
>
> We will add these considerations in the final version of the work.
>
> [1] Mutti, M. and Restelli, M. An intrinsically-motivated approach for learning highly exploring and fast mixing policies. In Proceedings of the AAAI Conference on Artificial Intelligence, volume 34, pp. 5232–5239, 2020.
>
> > [...] setting of linear MDP or MDP with general function approximation?
> > [...] function approximation or continuous state-action spaces would be more interesting.
>
> We recognize that the regulation problem addressed in this work would naturally fit within a continuous state-action space setting (e.g., linear or general function approximation). We decided to focus this work on the tabular case for two main reasons.
>
> First, the primary contribution of the paper is the introduction of a novel problem, the FMSS, as acknowledged by the Reviewer, and, as customary in the literature, we begin by studying it in the tabular MDP setting. Second, the tabular MDP setting allows for a convenient mathematical treatment that highlights the fundamental insights and concepts that will need to be generalized to the continuous case.
>
> We expect that most of the challenges arising from this generalization are not conceptual but rather technical, primarily related to the mathematical tools required. In particular, one would need to address the mixing properties and spectral analysis over of the densities associated with the continuous Markov chain. While the core ideas remain intuitively similar, the mathematical treatment becomes significantly more involved, moving from finite-dimensional to Hilbert spaces.
>
> For these reasons, we believe that the continuous setting is beyond the scope of the present paper and is left for future work.

---

> > ### Author Rebuttal · Reviewer_u3xK · 2026-04-03
> >
> > The concerns about the dependence on t_{mix} and 1/\eta_{min} remain unsolved. Therefore, I tend to keep my evaluation.

---

> > > ### Author Response · Authors · 2026-04-07
> > >
> > > We understand the Reviewer's push for a tighter dependence of the sample complexity bound w.r.t. $t_{\text{mix}}^{b}$ and $1/\eta^b_{\min}$ as it would indeed be desirable. We realized that our previous response might have prioritized a rather intuitive explanation rather than providing a rigorous justification for the unavoidable dependence on $t_{\text{mix}}^{b}$ and $1/\eta^b_{\min}$. In the following we will make the argument formal.
> > >
> > > **Offline Setting: A Minimax Lower Bound**
> > >
> > > We begin with the following definition,
> > >
> > > **Definition** [Pseudo Spectral Gap, Equation 2.5 in [1]]
> > > The *pseudo-spectral gap* of a geometrically ergodic MC $\mathbf{T}$ over $\mathcal{S}\times\mathcal{A}$ is defined as,
> > > $$\\gamma\_{ps}(\\mathbf{T}) := \\max\_{k \\geq 1} \\left\\{\frac{1-\lambda\_{2}\\left(\\tilde{\\mathbf{T}}^{k} \\mathbf{T}^{k}\\right)}{k}\\right\\}.$$
> > >
> > > Also, define the mixing time of the chain as $t_{\text{mix}}(1/4;\mathbf{T}) = \inf\{t\geq 0\mid d(t;\mathbf{T})\leq 1/4\}$
> > > and finally, denote with $x_{\min}:=\min_{s,a}x(s,a)$ the minimum entry of the stationary distribution of the chain.
> > >
> > > Then, from Theorem 3.2 in [1], we have that for all $\varepsilon \in (0,1/32)$ and forall $\gamma_{ps}\in(0,1/8)$ and $|\mathcal{S}||\mathcal{A}|= 6k \geq 12$ and for every estimator $\hat{\mathbf{T}}$ it is possible to find a Markov chain $\mathbf{T}$ such that given a sample $\\{(S_0,A_0),\ldots,(S_{T-1},A_{T-1})\\}$ there exists a universal constant $c>0$ such that if,
> > >
> > > $$T \geq c \max\left\\{\frac{|\mathcal S||\mathcal {A}|}{\varepsilon^2 x_{\min}},\frac{|\mathcal S||\mathcal {A}|\log(|\mathcal S||\mathcal {A}|)}{\gamma_{ps}(\mathbf{T})}\right\\}$$
> > >
> > > then, $\mathbb{P}(\\|\hat{\mathbf{T}}-\mathbf{T}\\|_{\infty}<\varepsilon)\geq 9/10$.
> > >
> > > We can apply the above lower bound to our setting by first observing that, under Assumption 2.1 and $\pi^b$ as in Assumption 5.1, let $\mathbf{T}^b={\mathbf{P}\mathbf{\Pi}^b}$ and observe that for any estimator $\hat{\mathbf{P}}$ it holds that,
> > >
> > > $$\\|\hat{\mathbf{T}}^b-\mathbf{T}^b\\|\_{\infty}  = \\|\hat{\mathbf{P}}\mathbf{\Pi}^b-\mathbf{P}\mathbf{\Pi}^b\\|\_{\infty} = \\|\hat{\mathbf{P}}-\mathbf{P}\\|\_{\infty}.$$
> > >
> > > Also, we can conveniently relate $\gamma_{ps}$ with the mixing time via Proposition 3.4 from [2],
> > >
> > > $$\frac{1}{2t_{\text{mix}}(1/4;\mathbf{T})}\leq \gamma_{ps}(\mathbf{T})\leq \frac{1+2\log(2)+\log(1/x_{\min})}{t_{\text{mix}}(1/4;\mathbf{T})}$$
> > >
> > > Then observing that $t_{\text{mix}}(\zeta;\mathbf{P}^{\pi^b}) \leq t_{\text{mix}}(\zeta;\mathbf{P}\mathbf{\Pi}^b)$ and renaming $x^b_{\min} = \eta^b_{\min}\xi$, we can rewrite the sample complexity lower bound as,
> > >
> > > $$T \geq c \max\left\\{\frac{|\mathcal S||\mathcal {A}|}{\varepsilon^2 \eta^b_{\min}\xi},2|\mathcal S||\mathcal {A}|\log(|\mathcal S||\mathcal {A}|)t^b_{\text{mix}}(1/4;\mathbf{P}^{\pi^b})\right\\}$$
> > >
> > > Finally, recall that, to have guarantees in terms of $\\|\hat{\mathbf{P}}-\mathbf{P}\\|\_2$ the following inequality can be leveraged,
> > > $$ \\|\hat{\mathbf{P}}-\mathbf{P}\\|\_{\infty} \leq \frac{1}{\sqrt{|\mathcal{S}|}} \\|\hat{\mathbf{P}}-\mathbf{P}\\|\_2.$$
> > >
> > > This lower bound shows that the **dependence on either $t^b_{\text{mix}}$ or $1/\eta^b_{\min}$ is unavoidable**. Specifically, the sample complexity is monothonically increasing w.r.t. these structural properties of the MDP under $\pi^b$.
> > >
> > >
> > > [1] Wolfer, G. and Kontorovich, A., 2021. Statistical estimation of ergodic Markov chain kernel over discrete state space.
> > > [2] Paulin, D., 2015. Concentration inequalities for Markov chains by Marton couplings and spectral methods
> > >
> > > **Online Setting**
> > >
> > > Beside the approach that we have exposed in the previous response, we stress that the problem of learning the MDP transition kernel **when the horizon is infinite in an online setting** (i.e., when we can adapt the baseline policy $\pi^b$ during learning) is under-explored in the literature. Indeed, to our knowledge, no sample complexity lower bounds exist for this case. Additionally, as the online setting differs fundamentally from the offline case, a more exhaustive analysis would require significant further study and despite being an interesting research direction, we believe, it is out of the scope of this work.
> > >
> > > To conclude, we agree with the Reviewer that the dependence highlighted is an important point, thus, we will add a specific remark in the final version of the work.

---

### Decision · Program_Chairs · 2026-04-30

**Decision:**

Accept (regular)

**Comment:**

While reviewers raised several concerns regarding modeling assumptions, surrogate quality, and limited empirical evaluation, they agreed that the paper introduces a novel and well-formulated problem (fast-mixing steady-state control (FMSS) in MDPs) and provides a technically meaningful approach. In particular, the paper develops a convex surrogate based on the second-largest singular value (SLSV) to address an otherwise intractable spectral optimization problem, and complements this with theoretical analysis and a sample-based offline algorithm.

The reviewer opinions are somewhat mixed, but overall on the positive side. Importantly, concerns raised during the review process (such as the characterization of the SLSV–SLEM gap and aspects of the sample complexity analysis) were partially or fully addressed in the rebuttal.
The main remaining limitations concern the restriction to tabular MDPs, dependence on strong coverage assumptions, and the surrogate objective (SLSV), which may deviate from the true mixing rate in non-normal settings. In addition, the empirical validation is limited, leaving open questions about scalability and broader applicability.

Nevertheless, the paper offers a conceptually novel bridge between control-theoretic stability notions and MDP policy design, and provides a tractable and analyzable framework for a previously underexplored objective. Overall, I believe the contribution meets the bar for acceptance.